# Multifunctional roles of Brl1-Brr6 in nuclear envelope fusion during nuclear pore complex biogenesis

Sayan Mondal[1,2,4], Annett Neuner[1,4], Azqa Ajmal Khan [1,3,4✉], Jlenia Vitale[1,3] & Elmar Schiebel [1✉]

## Abstract

**Brl1 and Brr6 are integral membrane proteins of the yeast nuclear envelope (NE) that transiently associate with nuclear pore complexes (NPCs). The exact roles of Brl1 and Brr6 during NPC assembly are unclear. Here, we demonstrate that Brr6 operates at both early and late stages of NPC assembly. Its early function is supported by amphipathic α-helix mutants, which impact nucleoporin recruitment without nuclear envelope deformation, whereas mutations in conserved cysteine residues result in NE deformation accompanied by defective NE fusion. The N-terminus of Brl1 interacts with the nucleoporin Nic96, promoting Nic96 recruitment to early assembly sites. AlphaFold predictions, the essential role of the conserved PAL motif, and the inhibition of NE fusion upon overexpression of PAL mutants together suggest that the perinuclear domains of Brl1 and Brr6 interact across the perinuclear space. Extending the length of the perinuclear-space regions of Brl1 and Brr6 causes uncontrolled NE fusion, as indicated by nuclear envelope disintegration dependent on the conserved cysteine residues. Together, Brl1 and Brr6 promote NE fusion by bridging the perinuclear space through PAL motif interactions, followed by nuclear envelope fusion and NPC insertion.**

**Keywords** Brl1; Brr6; Nuclear Pore Complex; Nuclear Pore Complex Assembly; Nuclear Envelope Fusion
**Subject Categories** Membranes & Trafficking; Organelles; Structural Biology

## Introduction

In eukaryotic cells, the genome is encapsulated by the nuclear envelope (NE), a spherical double membrane (Lin and Hoelz, 2019; Strambio-De-Castillia et al, 2010). Transport of macromolecules, RNA, and proteins, in and out of the nucleus, is bidirectional and facilitated by nuclear pore complexes (NPCs) that are embedded into fusion sites of the outer (ONM) and inner nuclear membrane (INM) (Beck and Hurt, 2017). NPCs consist of multiple copies of 30 different nucleoporins

(Nups) that can be divided into four categories. Transmembrane Nups form a ring structure that anchors the NPC to the nuclear envelope (NE), while the inner and outer ring Nups constitute the core scaffold of the NPC, which also contributes to the formation of the central transport channel (Strambio-De-Castillia et al, 2010). Phe-Gly (FG) Nups are anchored to the core scaffold by linker Nups and are responsible for the selective permeability barrier of NPCs (Beck and Hurt, 2017; Frey and Gorlich, 2007).

Despite the essential functions of the NPC, the mechanisms underlying its assembly remain poorly understood. In higher eukaryotes, NPCs assemble via two distinct pathways. During open mitosis, the NE breaks down in prophase, leading to NPC disassembly. Following chromosome segregation, the NE and NPCs must be reassembled during mitotic exit, marking the transition from mitosis to G1 phase. In addition to this post-mitotic assembly, NPCs also form during interphase through an inside-out mechanism. In this pathway, Nups first accumulate on the nuclear side of the INM followed by local INM deformation toward the ONM, fusion of the INM and ONM, and eventual insertion of the assembled NPC into the NE (Otsuka et al, 2016; Otsuka and Ellenberg, 2018; Otsuka et al, 2018). Some cytoplasmic Nups attach to the NPC after its embedding into the NE.

Because the budding yeast *Saccharomyces cerevisiae* undergoes closed mitosis, where the NE remains intact throughout cell division, NPCs assemble exclusively via the interphase pathway (Winey et al, 1997). Yeast NPC assembly also follows an inside-out mechanism, as evidenced by mutations in some genes encoding Nups such as *NUP116* or involved in NPC biogenesis such as *BRL1* and *BRR6* (de Bruyn Kops and Guthrie, 2001; Onischenko et al, 2017; Wente and Blobel, 1993; Zhang et al, 2018). These mutations lead to herniations, deformations of the INM that protrude into the perinuclear space and contain Nups on the nuclear-facing side of the INM. Herniations probably arise because NPC assembly is successfully initiated at the INM but fusion between the two nuclear membranes fails (Onischenko et al, 2017; Thaller and Lusk, 2018; Zhang et al, 2018).

In *Saccharomyces cerevisiae*, the interacting, integral membrane proteins Brl1, Brr6, and Apq12 are important for the assembly of new NPCs without being components of fully assembled NPCs. While *BRL1* and *BRR6* are essential for yeast cell viability, *Δapq12* cells exhibit a cold-sensitive growth defect (Hodge et al, 2010; Lone et al, 2015; Scarcelli et al, 2007; Zhang et al, 2021; Zhang et al, 2018). Apq12, Brl1, and Brr6 each contain two transmembrane

[1]Zentrum für Molekulare Biologie der Universität Heidelberg (ZMBH), Deutsches Krebsforschungszentrum (DKFZ)-ZMBH Allianz, Universität Heidelberg, Heidelberg 69120, Germany. [2]Heidelberg Biosciences International Graduate School (HBIGS), Universität Heidelberg, Heidelberg 69120, Germany. [3]Present address: EMBL, Meyerhofstraße 1, Heidelberg 69177, Germany. [4]These authors contributed equally: Sayan Mondal, Annett Neuner, Azqa Ajmal Khan. ✉E-mail: azqa.khan@embl.de; e.schiebel@zmbh.uni-heidelberg.de

regions (TM), which are connected by a stretch of amino acids that localize in the perinuclear space (Vitale et al, 2022; Zhang et al, 2021; Zhang et al, 2018). The perinuclear amino acids of Apq12 fold into a short amphipathic α-helix (AαH) that was shown to coordinate Brl1-Brr6 interaction and regulate lipid composition (Zhang et al, 2021).

The paralogues Brl1 and Brr6 also contain an AαH within the perinuclear region, along with an additional disulfide-stabilized antiparallel helix bundle (DAH) that carries the conserved PAL motif at its tip (Gardner et al, 2021). Several observations suggest that Brl1 plays a late role in NPC assembly, specifically, in mediating the fusion of the INM and ONM, thereby facilitating the integration of the emerging NPC into the NE. First, depletion of Brl1 results in the formation of herniations (Zhang et al, 2018). Second, overexpression of *BRL1* mutants that impair AαH function triggers the formation of INM petal-like structures with attached Nups on its base, suggesting continuous expansion of the INM without fusion to the ONM (Kralt et al, 2022; Vitale et al, 2022). Finally, mild overexpression of *BRL1* suppresses the herniation defect in *nup116Δ* cells, meaning that it enables fusion between the INM and ONM (Vitale et al, 2022; Wente and Blobel, 1993; Zhang et al, 2018).

It has been suggested that Brr6 carries out similar functions as Brl1, but at different sides of the NE, as Brr6 localizes to the INM and ONM, while Brl1 is predominately detected at the INM (Kralt et al, 2022; Vitale et al, 2022; Zhang et al, 2018). Brl1 and Brr6 may connect the INM and ONM across the perinuclear space via their perinuclear domain, which comprises the DAH/PAL and the adjacent AαH, once the INM bends toward the perinuclear space as a result of Nup deposition (Kralt et al, 2022; Vitale et al, 2022). However, the molecular mechanism by which Brl1 and Brr6 promote fusion of the INM and ONM during NPC assembly, as well as the specific domains involved in this process, remains unclear. Furthermore, the Brr6–Brl1 interaction across the perinuclear space does not account for Brr6's localization to the INM, suggesting that Brr6 may have a NE fusion–independent function during NPC assembly (Zhang et al, 2018).

In this study, we investigate the functions of Brl1 and Brr6 in NPC biogenesis. Surprisingly, the AαH of Brr6 plays an early role in NPC assembly, as indicated by the *brr6$^{L145E}$* and *brr6$^{F152E}$* mutant phenotypes, which affect Nup accumulation at the INM without INM deformation. Brl1 may also play an early role at the INM during NPC assembly, as suggested by the interaction of its essential N-terminus with Nic96. We further show that INM–ONM fusion during NPC assembly requires the conserved PAL motif in Brl1 and Brr6 at the tip of the DAH, likely mediating interactions across the NE. Elongation of the DAH domain in either Brl1 or Brr6 brings the PAL motifs in the perinuclear space into close proximity, thereby enabling formation of the fusion complex without INM deformation by Nups normally required for NPC assembly (Otsuka and Ellenberg, 2018). These findings indicate that Brl1 and Brr6 perform multiple essential roles in NPC biogenesis and support a model in which the length of the DAH is a critical determinant for proper NE fusion during NPC assembly.

# Results

## Brl1 and Brr6 interactions

Previous studies have shown that Brl1 and Brr6 interact physically (Lone et al, 2015; Zhang et al, 2021; Zhang et al, 2018). Brl1 is predominantly localized to the INM, whereas Brr6 exhibits dual localization at both the ONM and INM (Zhang et al, 2018). These distribution patterns suggest the potential for multiple modes of interaction between Brl1 and Brr6. To investigate this, AlphaFold modeling of Brl1 and Brr6 (Jumper et al, 2021) was performed. As previously reported (Kralt et al, 2022; Vitale et al, 2022), Brr6 and Brl1 are predicted to contain a perinuclear space domain that includes the DAH, stabilized by two disulfide bridges and an AαH, and is flanked by two transmembrane (TM) regions. Their N- and C-termini are exposed to either the nucleoplasm or the cytoplasm, depending on whether the proteins are located in the INM or ONM (Fig. 1A,B).

Using full-length Brl1 and Brr6 for structural predictions of Brl1-Brr6 interactions, AlphaFold suggests that the hydrophobic amino acids Ala388 and Phe391 of the AαH domain of Brl1 interact with the hydrophobic surface (Leu145, Ile149, and Phe152) of the AαH domain of Brr6 due to the hydrophobic effect. This interaction may serve as a shielding mechanism for the AαH domains, preventing them from remaining constantly bound to the NE (Figs. 1C and EV1A). When only the perinuclear space (DAH and AαH) regions of Brl1 and Brr6 are used, AlphaFold predicts head-to-head interactions between Brl1 and Brr6, mediated by the tip region of their DAHs, including the conserved PAL motif (Fig. 1D,E) (Gardner et al, 2021).

We also analyzed by AlphaFold the impact of mutations that disrupt the integrity of the AαH domain in either Brr6 or Brl1 on the predicted Brl1–Brr6 interaction. In contrast to the wild-type proteins that are predicted to interact via their AαH (Fig. 1C), mutations affecting the hydrophobic face of the AαH in either protein (brl1$^{F391E}$ or brr6$^{L145E}$) combined with the wild-type versions of Brr6 or Brl1, respectively, lead to predicted head-to-head Brl1–Brr6 interactions when full-length proteins are modeled (Fig. EV1C,D). Remarkably, introducing mutations in both AαH domains (brl1$^{F391E}$-brr6$^{L145E}$) restores the prediction of lateral AαH interactions observed in the wild-type Brl1–Brr6 complex (Fig. EV1A,E). Evidence supporting these predicated lateral interactions came from immunoprecipitation experiments. The interaction between Brl1 and Brr6 was reduced when Brl1 was tested with brr6$^{L145E}$ or when brl1$^{F391E}$ was tested with Brr6. Interestingly, combining brl1$^{F391E}$ and brr6$^{L145E}$ restored the Brl1–Brr6 interaction (Fig. EV1F).

When AlphaFold was run with multiple copies of Brl1 or Brr6, it predicted that the proteins can oligomerize into both homo-oligomeric (Brl1–Brl1 or Brr6–Brr6) and hetero-oligomeric (Brl1–Brr6) structures (Appendix Fig. S1A,B). Octamers of Brl1 and Brr6 were not predicated to form rings. However, homo-octamers of Brr6 and Brl1 are predicted to form barrel-shaped structures, consisting of two octameric rings of Brr6 and Brl1 that interact at the tips of their DAHs through the PAL motifs. (Appendix Fig. S1C). Notably, the confidence score of the Brr6 and Brl1 oligomer predictions was relatively low.

Evidence for larger Brl1 assemblies was obtained from blue-native gel analysis of cells expressing pGal1-*BRL1* (*BRL1* expressed from the galactose-inducible pGal1 promoter). Brl1-containing complexes of ~130 kDa, along with higher molecular weight complexes, were detected using Brl1-specific antibodies (Appendix Fig. S1D). Notably, mutations in the AαH of Brl1 or Brr6 abolished the formation of high molecular weight complexes (Appendix Fig. S1E), consistent with AlphaFold predictions, indicating that these mutations impair lateral Brr6-Brl1 interactions (Fig. EV1C,D).

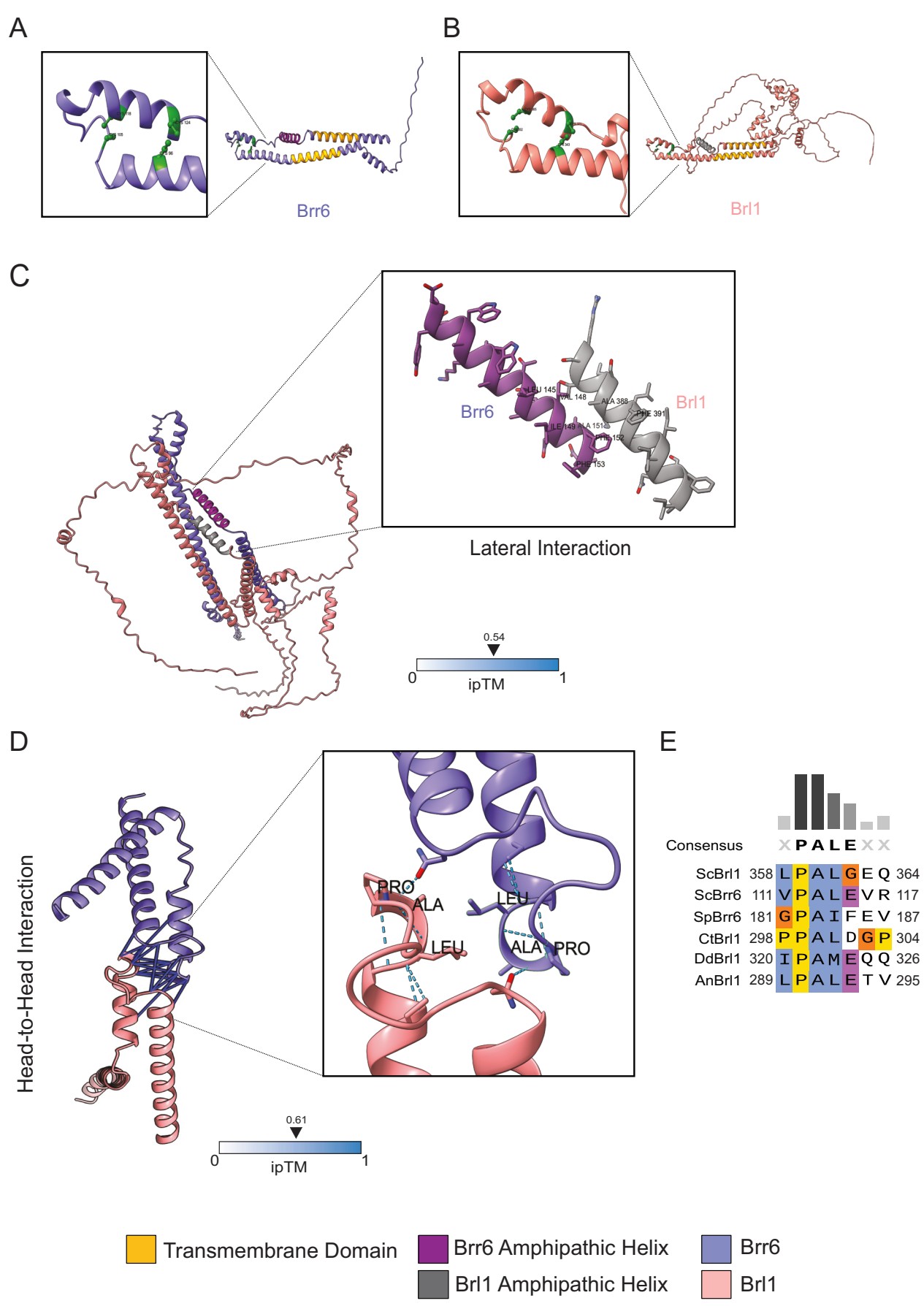

Lateral Interaction

Head-to-Head Interaction

Transmembrane Domain     Brr6 Amphipathic Helix     Brr6

Brl1 Amphipathic Helix     Brl1

**Figure 1.  Brl1 and Brr6 interactions.**

(A) Domain organization of Brr6. An AαH (magenta) is flanked by two transmembrane (TM) domains (yellow). Two disulfide bonds are highlighted in green in the enlargement. The long N-terminal domain localizes to either the nucleoplasm or the cytoplasm, depending on whether Brr6 is associated with the INM or the ONM, respectively. (B) Domain organization of Brl1. The AαH is in dark gray, and the four cysteine residues are shown in light green (enlargement). The two TMs are shown in yellow. (C) AlphaFold2 prediction of the Brl1–Brr6 interaction of full-length Brr6 and Brl1. It predicts interactions between the hydrophobic surfaces of the two AαHs. (D) The interaction surface between the head-to-head domains of both Brr6 and Brl1 predicated using AlphaFold2. The interactions at the tip of the DAH, which contains the conserved PAL sequence, are shown in the enlargement. (E) Sequence alignment of the conserved PAL motif across the indicated Brl1 and Brr6 homologs. ScBrl1 and ScBrr6: Brl1 and Brr6 from *S. cerevisiae*; SpBrr6: *S. pombe* Brr6: CtBrl1: *Chaetomium thermophilum* Brl1; DbBrl1: *Dictyostelium discoideum* Brl1; AnBrl1: *Aspergillus nidulans* Brl1. Sequence alignment was performed using the Clustal Omega program (Madeira et al, 2024).

Interestingly, co-expression of *brl1^{F391E}* and *brr6^{L145E}* mutants restored high molecular weight complex formation (Appendix Fig. S1E; asterisk), in agreement with AlphaFold predictions (Fig. EV1E).

Together, the lateral AαH interactions may mediate Brl1–Brr6 contacts at INM, where both proteins are localized (Zhang et al, 2018). The DAH tip-to-tip interactions could facilitate contacts across the NE, bridging Brl1 on the INM and Brr6 on the ONM through the perinuclear space.

## AαH of Brr6 performs an essential function required for yeast viability

The functional importance of the AαH in Brl1 for INM–ONM fusion during NPC assembly was demonstrated by the defective attempts of fusion of the deformed INM with the ONM in overexpressed *BRL1* AαH mutants (Kralt et al, 2022; Vitale et al, 2022). We addressed the function of the AαH in Brr6 following a similar strategy by introducing the AαH-disrupting mutations L145E and F152E in *BRR6* (Fig. 2A). We first tested whether the L145E mutation affects the ability of the AαH to associate with membranes. To do this, we expressed GFP-tagged fragments corresponding to the wild-type AαH (*BRR6^{AαH}-GFP*) and the mutant AαH (*brr6^{L145E-AαH}-GFP*) in yeast. Brr6^{AαH-WT}-GFP localized predominantly to the NE and the plasma membrane, both of which were marked by dsRED-HDEL (His-Asp-Glu-Leu endoplasmic reticulum (ER) retention signal fused to dsRED) (Munro and Pelham, 1987) (Fig. 2B). In contrast, brr6^{L145E-AαH}-GFP showed only weak NE localization (Fig. 2B). These findings indicate that membrane interactions by the AαH of Brr6 depend on its amphipathic character.

To test the functional importance of the AαH in Brr6, we examined the effects of the *brr6^{L145E}* and *brr6^{F152E}* mutations using a plasmid shuffle approach. Cells with *brr6^{F152E}* failed to grow, indicating that the integrity of the AαH is essential for Brr6 function (Fig. 2C). Cells expressing *brr6^{L145E}* formed colonies with reduced efficiency on 5-FOA plates (Fig. 2C); and *brr6^{L145E}* cells exhibited cold sensitivity (Fig. 4A).

We next assessed the impact of overexpressing *brr6^{L145E}* and *brr6^{F152E}* on yeast growth. Cells overexpressing these mutant alleles displayed severe growth defects on the inducing galactose plates (Gal/Raf) while control cells (pGal1-*BRR6* and pGal1 vector) showed robust growth (Fig. 2D). In summary, the AαH of Brr6 performs an essential function required for yeast viability.

## The AαH of Brr6 is needed in early NPC assembly

We tested if overexpression of *brr6^{L145E}* and *brr6^{F152E}* affects NPC biogenesis. GFP-tagged Nups associated with different NPC

substructures (Fig. EV2A) were analyzed in this experiment (Figs. 3A and EV2). The controls, pGal1 plasmid or overexpression of *BRR6*, did not affect the smooth localization of Nups along the NE (Fig. 3A). In contrast, *brr6^{L145E}* and *brr6^{F152E}* overexpression impaired NE localization of most Nups, which accumulated in dots that localized close to the NE (Figs. 3A and EV2B). Peaks in the line scan analysis of affected Nup–GFP signals along the NE further support the presence of dot-like Nup signals that were not present in the control or the unaffected Nups (Fig. EV2C). Quantification of Nup distortion following overexpression of *brr6^{L145E}* and *brr6^{F152E}* enabled us to classify the Nups into three distinct groups (Fig. 3B). Those most heavily affected, with ~50–70% mis-localization (Group 1); intermediately affected Nups (Group 2; 20–50%); and Nups that retained NE localization (Group 3; >20%; Fig. 3B). Interestingly, Nic96 localization was intermediately affected by ~40–45% upon overexpression of *brr6^{L145E}* and *brr6^{F152E}* (Fig. 3B). Transmembrane Nups like Ndc1 and Pom152 and the nuclear FG repeat Nup2 were not affected (Fig. 3B).

We next analyzed the phenotypes of *brr6^{L145E}* and *brr6^{F152E}* overexpression by EM. Surprisingly, most cells overexpressing pGal1-*brr6^{L145E}* or pGal1-*brr6^{F152E}* for 3 h did not accumulate petal-like structures or herniations, as observed in pGal1-*brl1^{F391E}* or other *brl1* mutant cells (Kralt et al, 2022; Vitale et al, 2022; Zhang et al, 2018) (Fig. 3C). Instead, the NE appeared smooth. Analysis of NPC number per electron microscopy (EM) section revealed a ~50% reduction in cells overexpressing pGal1-*brr6^{L145E}* compared to those expressing pGal1 or pGal1-*BRR6* (Fig. 3D).

Immuno-EM analysis of the nucleoporin Nsp1 localization (Zhang et al, 2018) in cells overexpressing pGal1-*brr6^{L145E}* and pGal1-*brr6^{F152E}* revealed an increase in Nsp1 signals on the nuclear side of the NE that were not associated with NPCs compared to cells with pGal1 or pGal1-*BRR6* (Fig. 3C,E). In 2–5% of pGal1-*brr6^{L145E}* and pGal1-*brr6^{F152E}* cell sections, the Nsp1 signal was clustered on the cytoplasmic side of the NE as indicated by >10 gold particles (Fig. 3C,E). These gold patches probably reflect the stronger Nsp1-tdTomato or Nsp1-GFP dots that accumulate in cells overexpressing pGal1-*brr6^{F152E}* (Figs. 3A and EV2C). Nsp1 clusters were not observed in pGal1 and pGal1-*BRR6* cell sections (Fig. 3E, arrowhead).

Taken together, these findings suggest that overexpression of *brr6* mutants with a defective AαH impairs NPC assembly at an early stage prior to INM deformation, leading to the mis-localization of most Nups.

## Cold-sensitive *brr6^{L145E}* cells are devoid of NPCs

The *brr6^{L145E}* mutant cells exhibited cold sensitivity for growth (Fig. 4A). At 16 °C, analysis using Pom152 and Nup159 as NPC

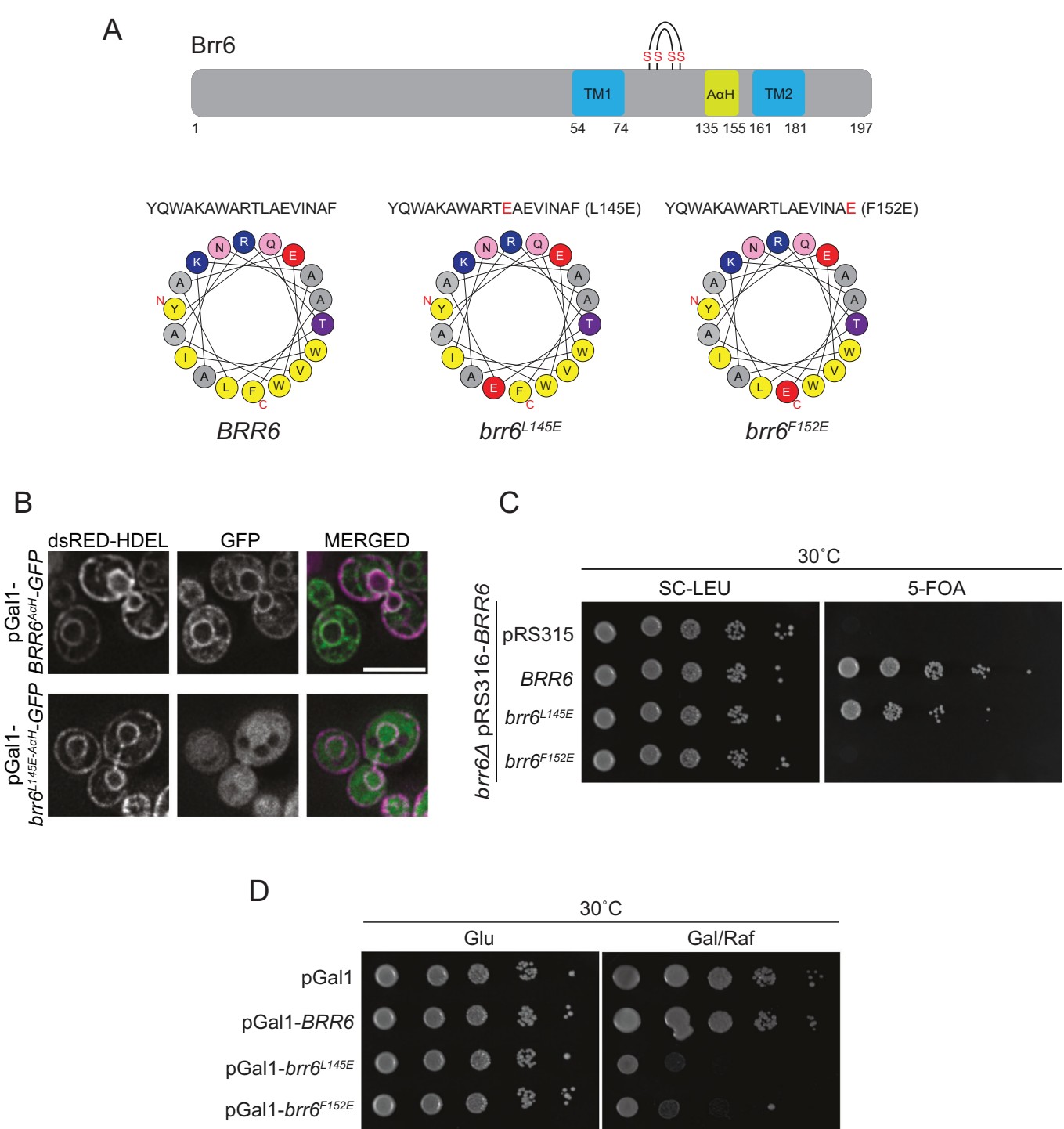

**Figure 2. AαH of Brr6 performs an essential function required for yeast viability.**

(**A**) AαH predictions for the wild-type and mutant sequences of Brr6 were generated using HeliQuest (Gautier et al, 2008). The upper cartoon illustrates the domain organization of Brr6, highlighting the two transmembrane (TM) domains, the AαH, and the four conserved cysteine residues. Amino acid positions are indicated. (**B**) Subcellular localization of GFP-tagged Brr6 wild-type and brr6^L145E AαH region. The pGal1 promoter was expressed for 3 h by the addition of galactose. Fluorescence microscopy was performed to assess localization. dsRed-HDEL was used as NE and ER marker (Madrid et al, 2006). Size bar: 5 μm. Representative image from three independent repeats. (**C**) Growth assay of the *BRR6* shuffle strain carrying the empty *LEU2*-based plasmid pRS315 or pRS315 containing the indicated *BRR6* alleles (*BRR6*, *brr6^L145E*, and *brr6^F152E*). Tenfold serial dilutions were spotted onto SC–LEU and 5-FOA plates and incubated at 30 °C for 2 days. Growth on 5-FOA plates selects for cells that have lost the *URA3* plasmid, thereby testing whether the *LEU2*-based plasmid alone can support viability in the absence of wild-type *BRR6*. One representative of three independent experiments is shown. (**D**) Wild-type yeast cells carrying the indicated pGal1 plasmids were spotted in tenfold serial dilutions onto glucose (Glu) or galactose/raffinose (Gal/Raf) plates and incubated at 30 °C. One representative of three independent experiments is shown.

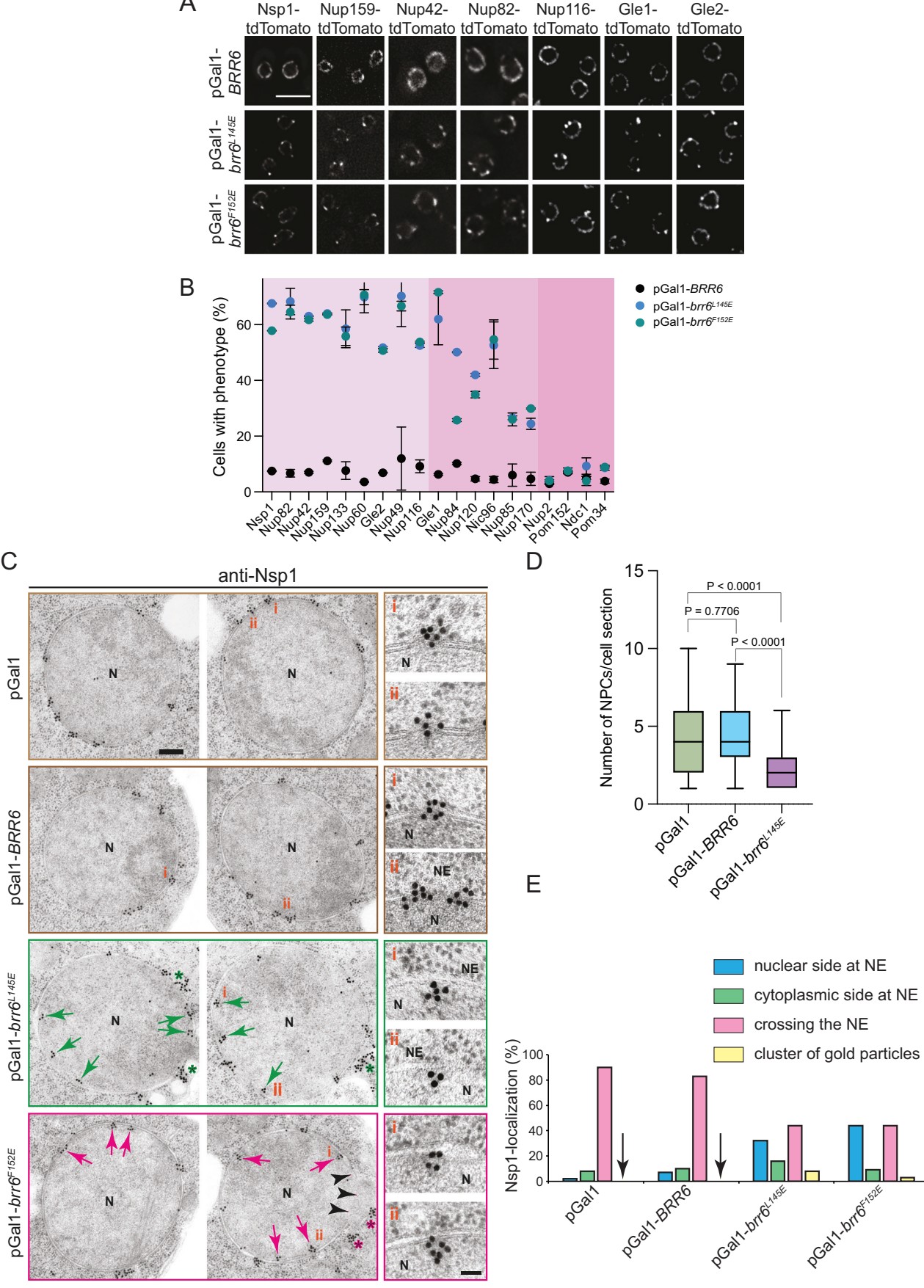

**Figure 3. The AαH of Brr6 is needed in early NPC assembly.**

(A) Overexpression of *brr6*^L145E^ alters NE localization of Nups. *BRR6*, *brr6*^L145E^, and *brr6*^F152E^ were overexpressed for 3 h in yeast cells expressing fluorescently tagged (tdTomato) components of the NPC. Three independent experiments. Note, the *NSP1-tdTomato* pGal1-*brr6*^F152E^ cells (first column) are identical to the cells used in the line scan in Fig. EV2C (top). Size bar: 5 µm. (B) Quantification of the mis-localization phenotype across different tagged Nups. The graph shows the percentage of cells exhibiting mis-localization upon overexpression of pGal1-*BRR6*, pGal1-*brr6*^L145E^, and pGal1-*brr6*^F152E^. 150 cells were analyzed per Nup-tdTomato/Nup–GFP strain and for each expression plasmid and experiment. Three independent experiments. Nup-tdTomato and Nup–GFP constructs (also see Fig. EV2) showing defects in 50–70% of cells were classified as strongly affected (light pink), those in 20–50% as intermediate (pink), and those in less than 20% as not affected (darker pink). *n* = 3; data show mean (indicated by dots) with standard deviation (SD). (C) Immuno-electron micrographs of wild-type cells with pGal1 (control plasmid), pGal1-*BRR6*, pGal1-*brr6*^L145E^, or pGal1-*brr6*^F152E^ for 3 h, stained with anti-Nsp1 antibodies and protein A–gold conjugated with 15 nm gold. Each panel shows two representative cells for each plasmid. Red and green arrows indicate the Nsp1 signal below the INM. Green and red asterisks highlight clusters (10 and more) of gold particles. N nucleus, NE nuclear envelope. Size bar: 200 nm. Enlarged views of selected regions marked with i and ii are shown to the right. Size bar: 50 nm. (D) Quantification of NPCs per EM section. The quantification of NPCs was done based on Nsp1 signals at the NPCs. The number of NPCs per section was analyzed in the cells from (C). *n* = 68 for pGal1; *n* = 70 for pGal1-*BRR6*; *n* = 67 for pGal1-*brr6*^L145E^. Statistical test: unpaired *t* test; Box and whisker plots represent the median with interquartile range. *P* = 7.3E-8 for the statistical significance between pGal1 and pGal1-*brr6*^L145E^ condition; *P* = 2.5E-10 for the statistical significance between pGal1-*BRR6* and pGal1-*brr6*^L145E^ condition. (E) Quantification of the localization of gold particles from (C). Only clusters of more than 2 gold particles were counted. Clusters containing ≥10 gold particles were counted as clusters of gold particles, as these likely correspond to the intense Nsp1 fluorescence foci observed in Fig. EV2. *n* = 35 cell sections per plasmid type. Arrowheads indicate the absence of gold particle clusters in pGal1 and pGal1–*BRR6* cells. Source data are available online for this figure.

markers revealed a localization defect for Nup159, but not for Pom152 that still showed smooth NE localization (Fig. 4B,C), similar to pGal1-*brr6*^L145E^ and pGal1-*brr6*^F152E^ cells (Fig. 3B). EM analysis revealed that the number of NPCs per NE section decreased from four in control cells to basically zero in *brr6*^L145E^ cells when incubated at 16 °C (Fig. 4D,E). Notably, the NE in *brr6*^L145E^ cells incubated at 16 °C showed only a very small number of herniations (Appendix Fig. S2A,B). The NE of most *brr6*^L145E^ cells was relatively smooth. However, we observed an accumulation of ER-like structures in the cytoplasm of *brr6*^L145E^ cells, containing elongated, electron-dense material. These structures were labeled by the Nsp1 antibodies, suggesting the presence of Nups (Fig. 4D, red asterisk). This phenotype suggests that because *brr6*^L145E^ cells cannot assemble NPCs in the NE, Nup-containing assemblies accumulate in the ER.

Taken together, these results support conclusions drawn from overexpression studies of *brr6* AαH mutants and indicate that Brr6 has a function in early NPC assembly, which requires its AαH.

### *BRL1* and *BRR6* interdependencies indicate functional cooperation between early steps of NPC assembly

Brr6 likely performs its early function in NPC assembly at the INM, where the process is initiated (Otsuka et al, 2016). AlphaFold predicts lateral interactions between the AαH domains of Brl1 and Brr6 (Fig. 1), most likely at the INM, where both proteins co-localize (Zhang et al, 2018). If the lateral Brl1–Brr6 interaction is important during early NPC assembly, one would expect that the cold-sensitive growth defect of *brr6*^L145E^ cells could be suppressed by mild overexpression of *BRL1* (pAdh1-*BRL1*, *BRL1* constitutively expressed from the pAdh1 promoter). This was indeed the case (Fig. 5A). In contrast, the growth defect of the DAH mutant *brr6-721* (C124R) (Zhang et al, 2018) that is defective in INM/ONM fusion was not suppressed by pAdh1-*BRL1* overexpression (Fig. 5B). Along these lines, co-overexpression of pGal1-*brl1*^F391E^ and pGal1-*brr6*^L145E^, which most likely restores lateral Brl1–Brr6 interactions (Fig. EV1F), rescued the toxic effects caused by expression of the single mutants (Fig. 5C). Additionally, we observed that overexpression of pGal1-*BRR6* suppressed the growth defect caused by pGal1-*brl1*^F391E^ (Fig. 5C).

Consistent with the rescue of the growth defect, the petal-like structures were detected in pGal1-*brl1*^F391E^ pGal1 cells, while co-

expression of pGal1-*brl1*^F391E^ and pGal1-*brr6*^L145E^ prevented the formation of these structures (Fig. 5D,E). Surprisingly, pGal1-*brl1*^F391E^ pGal1-*BRR6* cells, although able to grow on Raf/Gal (Fig. 5C), developed petals when grown on Raf/Gal plates (Fig. 5D,E). We propose that viability is maintained because some functional NPCs can still form in these cells, whereas this is not the case for *brl1*^F391E^ overexpression. The suppression of petals by pGal1-*brl1*^F391E^ pGal1-*brr6*^L145E^ co-overexpression compared to pGal1-*brl1*^F391E^ pGal1 was confirmed by EM analysis (Fig. 5F,G). We established that expression of Brl1 was similar in all cell types (Fig. 5H).

These genetic data indicate cooperativity between Brl1 and Brr6 during early steps in NPC assembly. Additional support for this idea came from the observation that expression of pGal1-*brr6*^F152E^, which disrupts NPC assembly at an early stage, impaired the NE localization of Brl1-GFP (Fig. 5I).

Taken together, these genetic data indicate functional cooperation between Brl1 and Brr6 at the INM during the early stages of NPC assembly.

### The N-terminus of Brl1 fulfils an essential function and interacts with the Nic96 complex

Brl1 possesses a relatively long N-terminal extension of 299 amino acids, which based on its topology is exposed to the cytoplasm or nucleoplasm (Zhang et al, 2018). In contrast, its C-terminal extension comprises only 42 amino acids. Interestingly, homology is not limited to the perinuclear space regions of Brl1 and Brr6 (Zhang et al, 2018); the N-terminus of *S. cerevisiae* Brl1 also shares sequence similarity with, for example, the N-terminus of *Schizosaccharomyces pombe* Brr6 (Appendix Fig. S3A).

To assess the functional relevance of these regions in Brl1, we replaced them with the corresponding segments from Brr6 (53 aa N-terminal/15 aa C-terminal) (Fig. 6A). While the *N-BRR6–BRL1* fusion (*brr6*^1-53^-*brl1*^300-471^) was non-functional (Fig. 6B, row 4—5-FOA), the *BRL1–C-BRR6* chimera (*brl1*^1-428^-*brr6*^182-197^) retained the essential function of *BRL1* (Fig. 6B, row 5—5-FOA), indicating that the N-terminus of Brl1 carries a critical function. Similarly, analysis of *BRR6* function revealed that the N-terminal Brl1–Brr6 fusion (*brl1*^1-299^-*brr6*^54-197^) was non-functional (Fig. 6C, row 6—5-FOA), whereas the C-terminal Brr6–Brl1 fusion (*brr6*^1-181^-*brr6*^429-471^) retained functionality (Fig. 6C, row 7—5-FOA).

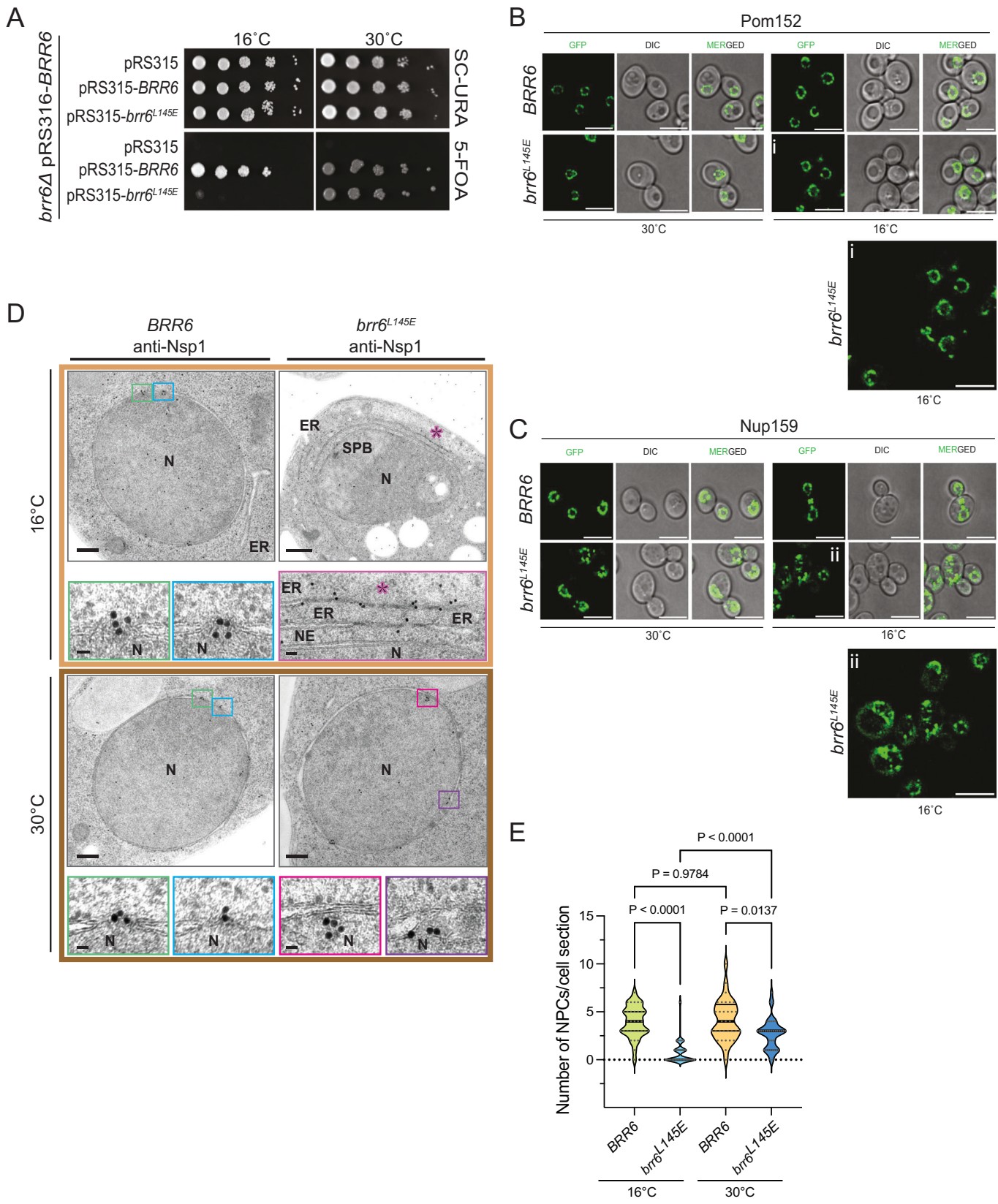

Figure 4. Cold-sensitive *brr6*<sup>L145E</sup> cells are devoid of NPCs.

**Figure 4. Cold-sensitive *brr6*$^{L145E}$ cells are devoid of NPCs.**

(A) The *brr6*$^{L145E}$ mutant exhibits cold-sensitive growth. To assess functionality at 16 °C, Δ*brr6* cells carrying the *URA3*-based plasmid pRS316-*BRR6* were transformed with the indicated *LEU2*-based plasmids. Transformants were grown at both 16 °C and 30 °C. One representative of three independent experiments is shown. (B, C) Phenotypic analysis of *brr6*$^{L145E}$ cells. The *brr6*$^{L145E}$ allele was genomically integrated into Δ*brr6* pRS316-*BRR6* cells, followed by counterselection on 5-FOA. *POM152* (B) and *NUP159* (C) were tagged with GFP using genomic integration (Janke et al, 2004). Cold-sensitive *brr6*$^{L145E}$ cells were incubated at 16 °C overnight and imaged by fluorescence microscopy. The enlargements (i) and (ii) at the bottom of (B, C) show *brr6*$^{L145E}$ cells grown at 16 °C. Three independent experiments. Size bar: 5 µm. (D) *brr6*$^{L145E}$ cells analyzed by immuno-electron microscopy using anti-Nsp1 antibodies. In wild-type cells, Nsp1 was predominantly detected at the NPCs. In *brr6*$^{L145E}$ mutant cells, the Nsp1 signal was observed below the INM. Additionally, *brr6*$^{L145E}$ cells accumulated electron-dense regions within the ER that were specifically labeled by Nsp1 antibodies. The purple asterisk indicates expanded ER structures containing electron-dense material labeled by anti-Nsp1 antibodies. N nucleus, NE nuclear envelope, ER endoplasmic reticulum, SPB spindle pole body. Size bar: 200 nm; enlargements: 25 nm (NPC or NE) and 50 nm (ER structures). (E) Quantification of NPCs per EM section from (D). The quantification of NPCs was done based on Nsp1 signals at the NPCs. *BRR6* (16 °C): n = 31; *brr6*$^{L145E}$ (16 °C): n = 32; *BRR6* (30 °C): n = 31; *brr6*$^{L145E}$ (30 °C): n = 35 sections were analyzed per strain and condition. Statistical test: One-way ANOVA. Plot shows the median with quartiles. The single dots indicate the individual values of the measurements. $P = 3.8E{-}10$ for the statistical significance between *BRR6* (16 °C) and *brr6*$^{L145E}$ (16 °C) condition; $P = 4.4E{-}7$ for the statistical significance between *brr6*$^{L145E}$ (16 °C) and *brr6*$^{L145E}$ (30 °C) condition. Source data are available online for this figure.

AlphaFold predictions revealed a potential interaction between the N-terminal region of Brl1 and Nic96, which forms a complex with Nsp1-Nup57-Nup49 (Fig. 6D) (Schrader et al, 2008). Although Nsp1 is not shown in Fig. 6D for clarity, its binding to the complex does not affect the predicted interaction with N-Brl1 (Appendix Fig. S3B). The N-Brl1–Nic96 interface is no longer accessible once the Nic96 complex is incorporated into mature NPCs, as it becomes occupied by Nup59 (Appendix Fig. S3C) (Akey et al, 2023). The predicted Brl1–Nic96 interaction, therefore, likely functions specifically during NPC assembly. This is consistent with the observation that Brl1 associates only with NPC assembly intermediates, but not with fully assembled NPCs (Zhang et al, 2018).

We validated the interaction between Brl1 and Nic96 by demonstrating co-immunoprecipitation between Brl1, Nic96, and Nup57 (Fig. 6E). We next asked whether the interaction between Nic96 and Brl1 requires the N-terminal region of Brl1, as predicted by AlphaFold (Fig. 6D). To test this, we replaced the N-terminus of Brl1 with the N-terminus of Brr6 (Brr6$^{1\text{-}53}$–Brl1$^{300\text{-}471}$; Fig. 6A). Both pGal1-*BRL1*-HA and pGal1-*brr6*$^{1\text{-}53}$-*brl1*$^{300\text{-}471}$-HA were expressed in yeast strains with or without chromosomally tagged *NIC96-GFP*. Immunoprecipitation of Nic96-GFP using GFP-binder beads co-immunoprecipitated Brl1, but not the brr6$^{1\text{-}53}$–brl1$^{30\text{-}471}$ chimera (Fig. 6F). Anti-Brl1 antibodies directed against N-Brl1 detected endogenous Brl1 co-immunoprecipitated with Nic96-GFP (Fig. 6F, bottom lane); however, the Brr6$^{1\text{-}53}$–Brl1$^{300\text{-}471}$ construct was not detected, as it lacks Brl1's N-terminal region. This interaction was specific, as no signal was detected in cells lacking *NIC96-GFP*. Finally, we showed that recombinant GST-Brl1$^{1\text{-}200}$ interacted with 6His-Nic96, both expressed in *E. coli*, whereas 6His-Nic96 did not bind to GST-nSec1 (Parisotto et al, 2012) that was used as negative control (Fig. 6G).

These findings suggest that the interaction between Nic96 and the N-terminus of Brl1 is likely critical for Brl1's role in NPC assembly.

## The PAL motif of Brl1 and Brr6 is essential for INM and ONM fusion

The PAL motif, located at the tip of the DAH, is a highly conserved feature of Brl1/Brr6-like proteins (Fig. 1E). AlphaFold predictions suggest that Brl1 and Brr6 may interact via their PAL motifs

(Fig. 1D), allowing the proteins, localized to the INM and ONM, respectively, to contact each other in the perinuclear space. If so, mutations in the PAL motif would not only impair Brl1 and Brr6 function but also block NE fusion during NPC biogenesis.

To investigate the function of the PAL motifs in Brl1 and Brr6, we replaced the PAL amino acids using site-directed mutagenesis as indicated in Fig. 7A,B. A plasmid shuffle approach revealed that the PAL motif is essential for viability (Fig. 7C,D). Overexpression of the *brl1* and brr6 PAL mutants (*brl1*$^{PAL}$ and *brr6*$^{PAL}$) caused severe toxicity (Fig. 7E,F). In *NUP82-GFP* cells expressing *BRL1*, *brl1*$^{PAL}$, *BRR6*, or *brr6*$^{PAL}$ from the pGal1 promoter, overexpression of the PAL mutants disrupted the smooth NE localization of Nup82-GFP (Fig. 7G–I). EM revealed large NE extrusions ("superherniations") that were morphologically distinct from the petal-like INM extensions in pGal1-*brl1*$^{F391E}$ cells (Vitale et al, 2022), as they lacked multiple continuous membrane layers (Figs. 7J and EV3). Notably, these superherniations frequently contained electron-dense material, possibly membrane fragments, and often pinched off from the NE to become released into the cytoplasm (Fig. 7J).

The PAL mutant phenotype indicates that NPC assembly was initiated but that INM–ONM fusion failed, allowing the INM to continue expanding and subsequently deform the ONM. Evidence for this notion came from immuno-EM analysis of the PAL mutants using anti-Nsp1 antibodies. The Nsp1 gold signal was concentrated on the nuclear INM side at the base of the superherniations (Appendix Fig. S4). Taken together, these findings demonstrate that the PAL motifs in Brl1 and Brr6 are essential for INM–ONM fusion during NPC assembly.

Since brr6 mutants affecting the AαH display an early role in NPC assembly, whereas the *brr6*$^{PAL}$ mutant indicates a late role in NE fusion, we next asked which additional region of Brr6 contributes to NE fusion. To investigate this, we analyzed six randomly generated conditional-lethal *brr6(ts)* mutants that had previously been examined only in the context of benzyl alcohol sensitivity (Zhang et al, 2018). Sequence analysis revealed that all *brr6(ts)* mutants harbored mutations in the perinuclear space region, including the conserved cysteine residues (Fig. EV4A). All *brr6(ts)* mutants accumulated herniations that distorted the INM when incubated at 37 °C (Fig. EV4B). These herniations, measuring approximately 40–140 nm (Fig. EV4C,D), were close to the ONM, but unlike the pGal1-*brr6*$^{PAL}$ phenotype, the ONM was only moderately deformed, and the herniations were completely filled

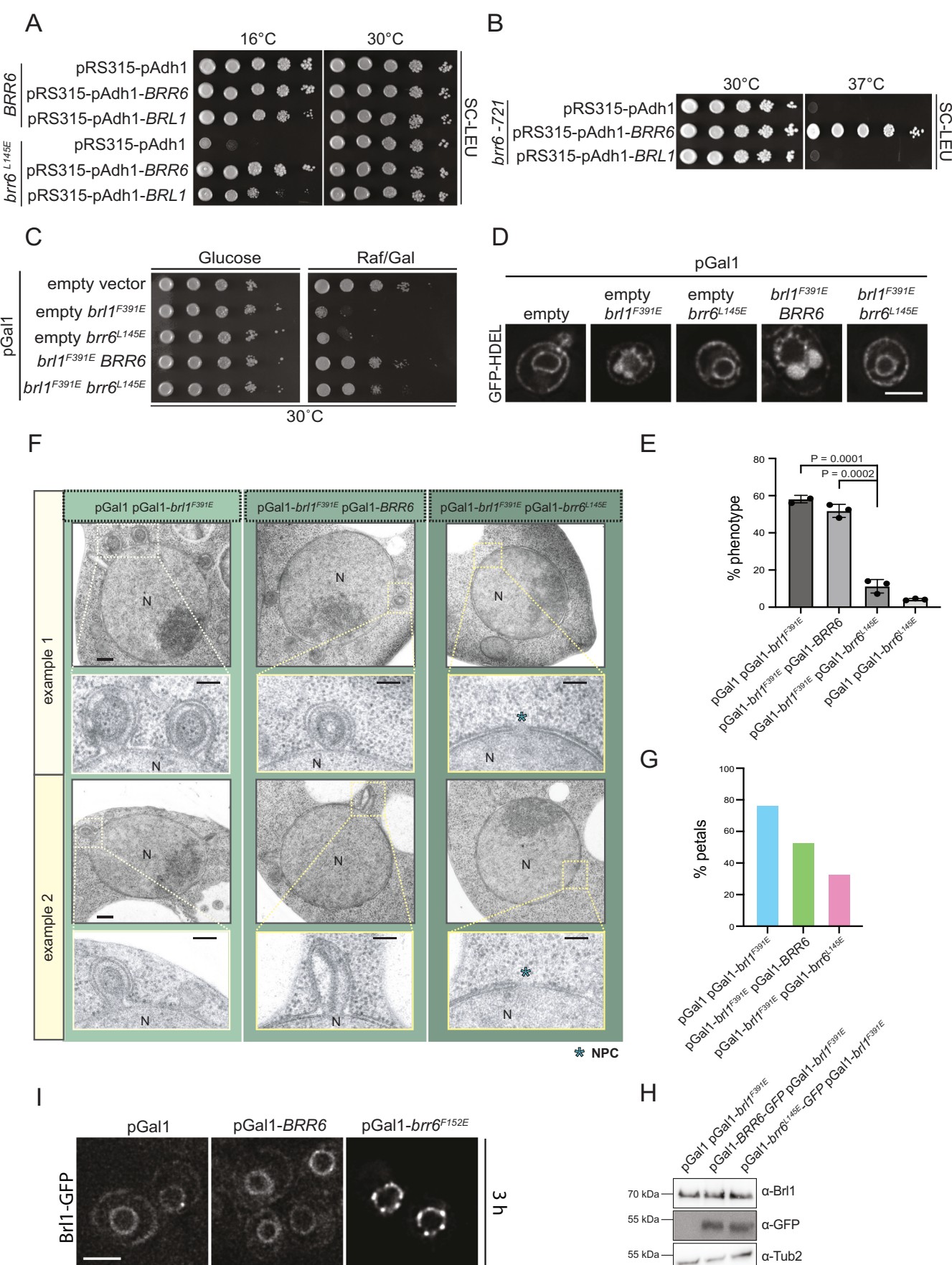

* NPC

**Figure 5. BRL1 and BRR6 interdependencies indicate functional cooperation between early steps of NPC assembly.**

(A) The growth defect of the cold-sensitive *brr6*[L14SE] cells was suppressed by pAdh1-*BRL1*, consistent with the role of the mutated helix in lateral Brl1–Brr6 interactions. (B) The growth defect of the DAH mutant *brr6-721* (C124R) was not suppressed by pAdh1-*BRL1* overexpression. (A,B) Three independent experiments. (C) Overexpression of the *brr6*[L14SE] AαH mutant reduces the toxicity of the *brl1*[F391E] AαH mutant. Tenfold serial dilutions of wild-type yeast cells carrying the indicated plasmids, along with the empty pGal1 plasmid, were grown on glucose or galactose/raffinose plates at 30 °C for 2 days. One representative of three independent experiments is shown. (D) Cells expressing the ER marker GFP-HDEL and the indicated plasmids were imaged by fluorescence microscopy after a 3 h induction with galactose. Petal-like structures are indicated as larger GFP-HDEL blobs at the NE. Three independent experiments. "Empty" indicates the pGal1 plasmid without an insert. Size bar: 3 μm. (E) Quantification of cells with NE petal-like structures upon overexpression, based on data from (D). Data represent the mean ± SD from three independent experiments ($n = 3$, 50 cells each condition). Statistical test: One-way ANOVA. Data show mean with SD. (F) Electron micrographs of cells overexpressing the indicated plasmids for 3 h at 30 °C. Size bar: 200 nm in overviews and 100 nm in enlargements. Blue asterisks indicate NPCs. N nucleus. (G) Quantification of the petal-like phenotype observed in (F). $n = 40$ sections were analyzed per plasmid combination. (H) Immunoblot showing expression levels of *brl1*[F391E]-GFP, *BRR6*-GFP, and *brr6*[L14SE]-GFP using the indicated antibodies. Tub2 was used as a loading control. One representative of three independent experiments is shown. (I) *BRL1*-GFP cells carrying the indicated pGal1 plasmids were incubated for 3 h at 30 °C after the addition of galactose and imaged under a fluorescence microscope. Three independent experiments. Size bar: 3 μm. Source data are available online for this figure.

with electron-dense material. Random measurements of the distance between the INM and ONM away from herniations or NPCs revealed no differences between wild-type and mutant *brr6(ts)* strains except for *brr6-732* cells (Fig. EV4E). A herniation phenotype similar to that observed in *brr6(ts)* mutants was also detected in the conditional-lethal *brl1*[C343Y] mutant, which had previously been examined only in pGal1–*brl1*[C343Y] overexpression experiments (Vitale et al, 2022). On the nuclear side, these herniations of *brl1*[C343Y] cells were labeled by the Nsp1 antibody (Appendix Fig. S5). The distinct herniation phenotypes suggest that the *brr6*[PAL] mutant is specifically defective in NE fusion, whereas the *brr6(ts)* and *brl1*[C343Y] mutants (Vitale et al, 2022) impair both NE fusion and overextension of the INM.

We also measured in a random manner the distance between the INM and ONM in wild-type cells incubated at 30 °C, which was on average approximately 22 nm (Fig. EV4F). In summary, mutations in Brr6 that affect the DAH domain impair INM–ONM fusion during NPC assembly.

## Brl1 and Brr6 promote NE fusion

NE fusion defects in *brl1*[PAL] and *brr6*[PAL] mutants indicate that Brl1 and Brr6 promote INM–ONM fusion during NPC assembly via head-to-head interactions of the PAL motifs at the tips of their DAH domains (Fig. 1D). AlphaFold analysis of Brl1 and Brr6 suggests that the lengths of their DAH domains, even in head-to-head configurations, are within 16 nm, insufficient to span the ~22 nm width of the perinuclear space in yeast (Figs. 8A and EV4F). Notably, as the INM becomes deformed by the deposition of Nups from the nuclear side, the local narrowing of the perinuclear space may bring the DAH/PAL domains of Brl1 at the INM and Brr6 at the ONM into close proximity (Fig. 8A), enabling their interaction. Based on this model, we hypothesized that artificially extending the DAH domains beyond a critical distance could allow Brl1 and Brr6 to interact and promote membrane fusion independently of INM deformation.

To test this, we extended the DAH domains of Brl1 and Brr6 by inserting 42 amino acids into Brl1 (*brl1*[ePNS]) and 30 into Brr6 (*brr6*[ePNS]), increasing their predicted lengths from 9 nm to 16 nm and from 7 nm to 12 nm, respectively, without affecting the tip of the DAH region (Fig. 8B). This indicates that the length of the interacting PNS regions is approximately 23 nm for *brl1*[ePNS]–Brr6

and 21 nm for Brl1–*brr6*[ePNS], which is very close to the average perinuclear space width observed in wild-type yeast cells (Fig. EV4F). Both the *brl1*[ePNS] and *brr6*[ePNS] constructs were non-functional in yeast (Fig. 8C,D). Strikingly, overexpression of these extended variants *brl1*[ePNS] and *brr6*[ePNS] resulted in a pronounced toxic phenotype, which was not observed upon overexpression of the wild-type *BRL1* or *BRR6* (Fig. 8E,F).

To understand the cause of the lethality associated with *brl1*[ePNS] and *brr6*[ePNS], we analyzed *NUP82*-GFP–tagged cells expressing either the wild-type or extended versions of *BRL1* and *BRR6* under the control of the pGal1 promoter. Fluorescence microscopy of cells 1 h after pGal1 induction revealed a pronounced defect in NPC organization (Figs. 8G,H and EV5A,B). Some cells exhibited a uniform Nup82-GFP signal distributed throughout the cytoplasm, indicating severe disruption of the NE and NPC structures (Fig. 8G, arrow). These findings indicate that expression of the extended versions of *BRL1* and *BRR6* compromises NE integrity.

To gain deeper insight into the defects observed by fluorescence microscopy data, we performed EM analysis. Thin-serial section EM of *brl1*[ePNS]- and *brr6*[ePNS]-expressing cells revealed NE abnormalities, including cells with partially disintegrated NE visible across serial sections (Figs. 8I and EV5C,D). In anaphase cells, the defective NE was most often found in the daughter cell (the bud), likely because transport of the NE through the bud neck during anaphase imposes mechanical stress on the membrane (Figs. 8I and EV5C). NE disintegration was also observed in interphase cells (Fig EV5D). Notably, such NE defects were absent in pGal1-*BRL1* and pGal1-*BRR6* control cells (Figs. 8I and EV5C,D).

To confirm the EM data showing NE defects upon over-expression of *brl1*[ePNS] and *brr6*[ePNS], we analyzed the localization of GFP–NLS (GFP fused to a nuclear localization signal), which in pGal1-*BRL1* and pGal1-*BRR6* control cells is restricted to the nucleus (Figs. 8J,K and EV5E,F), and of Rpl25–GFP, a ribosomal protein that is cytoplasmic in control cells (Figs. 8L,M and EV5G,H). Overexpression of *brl1*[ePNS] and *brr6*[ePNS] disrupted nuclear accumulation of GFP–NLS, which became distributed throughout the cell (Figs. 8J,K and EV5E,F). A substantial fraction of NLS-GFP was now observed in the cytoplasm. Additionally, Rpl25–GFP accumulated in the dsRED–HDEL–labeled nucleus upon overexpression of *brl1*[ePNS] and *brr6*[ePNS], indicating a loss of NE integrity (Figs. 8L,M and EV5G,H).

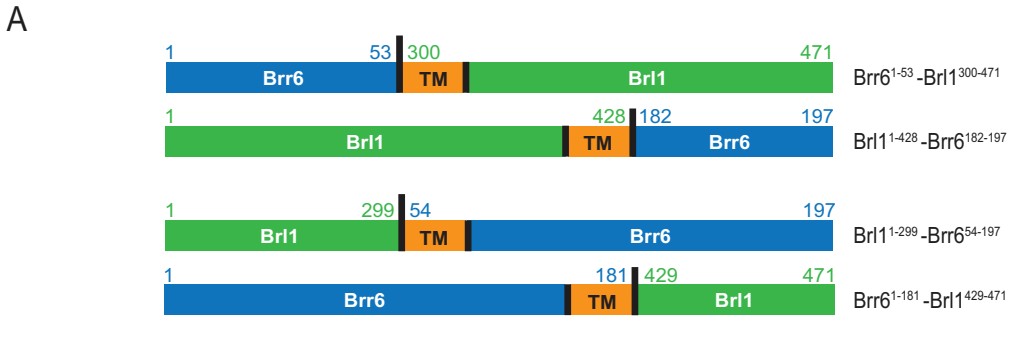

◄ **Figure 6. The N-terminus of Brl1 fulfills an essential function and interacts with the Nic96 complex.**

(A) Schematic representation of Brl1-Brr6 fusion proteins and their nomenclature. The amino acid numbers at the fusion junctions are indicated. The cartoon also depicts the TM domains of Brl1 and Brr6. (B, C) *BRL1*, *BRR6*, and *brl1-brr6* fusion constructs were expressed from their respective endogenous promoters using the *LEU2*-based pRS315 plasmid and transformed into *BRL1* (B) or *BRR6* (C) shuffle strains. Tenfold serial dilutions of transformants were grown at 30 °C on SC–URA and 5-FOA plates. The *brl1*$^{1-428}$-*brr6*$^{182-197}$ construct was functional for *BRL1*, while the fusion containing *BRL1* with the N-terminus of *BRR6* (*brr6*$^{1-53}$-*brl1*$^{300-471}$) was non-functional (B, rows 5 and 4, respectively), indicating an essential role of the N-terminus of Brl1. Similarly, *brl1*$^{1-299}$-*brr6*$^{54-197}$ failed to complement *BRR6* function, while *brr6*$^{1-181}$-*brl1*$^{429-471}$ was functional (C, rows 6 and 7). One representative of three independent experiments is shown. (D) AlphaFold prediction showing a potential interaction between the N-terminus of Brl1 and nucleoporins Nic96, Nup57, and Nup49. Nsp1 is not shown in this model to reduce complexity. The prediction of the Brl1-Nic96-Nsp1-Nup57-Nup49 interaction is shown in Appendix Fig. S3B. (E) Co-immunoprecipitation of Brl1 with Nic96 and Nup57. Cells expressing either *NIC96* or *NIC96-GFP*, both with pAdh1-*BRL1*, were lysed, and Nic96-GFP was immunoprecipitated using GFP nanobody beads. Eluted proteins were analyzed by immunoblotting with the indicated antibodies. Brl1 and Nup57 were specifically enriched in the Nic96-GFP pulldown. Three independent experiments. (F) Co-immunoprecipitation of Brl1-HA, but not Brr6$^{1-53}$-Brl1$^{300-471}$-HA, with Nic96-GFP. Yeast cells expressing either untagged *NIC96* or *NIC96-GFP* were transformed with pGal1-*BRL1-HA* or pGal1-*Brr6*$^{1-53}$-*Brl1*$^{300-471}$-*HA*. Expression from the pGal1 promoter was induced by the addition of galactose for 3 h. Shown is a GFP-binder immunoprecipitation analyzed by immunoblotting with the indicated antibodies. Note that Brr6$^{1-53}$-Brl1$^{300-471}$-HA is not recognized by anti-Brl1 antibodies, as these antibodies were raised against the N-terminus of Brl1, which is absent in this construct. Data are representative of three independent experiments. (G) Binding of recombinant 6His-Nic96 to GST-Brl1$^{1-200}$. GST-nSec1 was used as a negative control. GST-nSec1, GST-Brl1$^{1-200}$, and 6His-Nic96 were expressed in *E.coli* (Input). GST-nSec1 and GST-Brl1$^{1-200}$ were bound to glutathione beads. Cell lysates from *E. coli* expressing 6His-Nic96 were incubated with the glutathione beads. Bound proteins were analyzed by immunoblotting using GST and 6His antibodies. Results are from three independent experiments, and a representative image is shown. Source data are available online for this figure.

To assess the specificity of the pGal1-*brl1*$^{ePNS}$ and pGal1-*brr6*$^{ePNS}$ NE disintegration, we mutated the four cysteine residues in the extended DAH to alanine (CA), generating the *brl1*$^{ePNS-CA}$ and *brr6*$^{ePNS-CA}$ variants. The 4C-4A substitution inactivates the function of the wild-type Brl1 protein (Zhang et al, 2018). *brl1*$^{ePNS-CA}$ and *brr6*$^{ePNS-CA}$ were non-functional in a plasmid shuffle assay (Appendix Fig. S6A,B); their overexpression from pGal1 was toxic to cells (Appendix Fig. S6C,D). EM of pGal1-*brl1*$^{ePNS-CA}$ and pGal1-*brr6*$^{ePNS-CA}$ revealed a phenotype distinct from that caused by pGal1-*brl1*$^{ePNS}$ and pGal1-*brr6*$^{ePNS}$ (Figs. 8I and EV5C). Overexpression of the *brl1*$^{ePNS-CA}$ and *brr6*$^{ePNS-CA}$ mutants led to the accumulation of superherniations, vacuoles attached to the NE, ER fragments in the nucleus, and swollen NE (Appendix Fig. S6E,F). These phenotypes were not observed in control cells in which the pGal1 promoter was not induced (Appendix Fig. S6E). These observations indicate that the extended brl1$^{ePNS}$ and brr6$^{ePNS}$ variants require an intact DAH motif to induce NE disintegration.

Together, these findings show that the extended Brl1 and Brr6 variants compromise NE integrity, most likely by inducing uncontrolled fusion events between the INM and ONM.

## Discussion

With respect to the NE, the inside-out NPC assembly mechanism in yeast and mammalian cells involves fusion of the INM and ONM (Otsuka and Ellenberg, 2018), allowing the NPCs to become embedded in the nascent openings of the NE. Despite its importance, the mechanism underlying this membrane fusion has remained elusive. However, the discovery of Brl1 and Brr6 in yeast, two interacting paralogous integral membrane proteins of the NE, has shed light on this process, although the molecular details of their function remain unclear (de Bruyn Kops and Guthrie, 2001; Hodge et al, 2010; Kralt et al, 2022; Lo Presti et al, 2007; Lone et al, 2015; Saitoh et al, 2005; Vitale et al, 2022; Zhang et al, 2018).

Previous studies have shown that overexpression of *brl1* mutants with a defective AαH results in excessive proliferation of the INM, forming large petal-like structures that deform the ONM but do not lead to membrane fusion (Kralt et al, 2022; Vitale et al, 2022). *brl1* mutations such as *brl1*$^{C343Y}$ (Appendix Fig. S5) that disrupt the integrity of the DAH cause a temperature-sensitive phenotype, characterized by the accumulation of herniations, which is an indication of failed INM/ONM fusion (de Bruyn Kops and Guthrie, 2001; Saitoh et al, 2005; Zhang et al, 2018). Together, these findings suggest that DAH and AαH of Brl1 both have functions late in NPC assembly, in the fusion of the INM/ ONM.

In contrast to *BRL1*, *brr6* mutants affecting the AαH or the DAH indicate early and late NPC assembly defects, respectively, suggesting a dual role for Brr6 in NPC assembly. Overexpression of *brr6*$^{L145E}$, which affects the AαH, disrupts the NE localization of most nucleoporins and reduces the number of NPCs without causing NE deformations (Fig. 3). Similar phenotypes were observed in the cold-sensitive *brr6*$^{L145E}$ mutant as well (Fig. 4). In contrast, *brr6* mutants affecting the DAH exhibited small, compact herniations similar to those observed in *brl1*$^{C343Y}$ cells (Fig. EV4; Appendix Fig. S5), indicating a role of the DAH in NE fusion during NPC assembly. The occurrence of the early *brr6* phenotype raises the question of why it was not reported in earlier studies, such as those using Brr6 degron or Brl1/Brr6 double degron systems, which described the accumulation of NE herniations following protein degradation (Zhang et al, 2018). One possibility is that the early function of Brr6 in NPC assembly requires only low protein levels, and thus is less sensitive to depletion than its later roles.

The cross-complementation observed between the *brl1* and *brr6* AαH mutants (Fig. 5), together with the mis-localization of Brl1 in *brr6*$^{L145E}$ cells (Fig. 5), suggests that Brr6's early role in Nup deposition may be carried out in coordination with its interacting partner Brl1. This interaction likely occurs at the INM, where both proteins co-localize and probably engage through their AαH domains (Fig. 1) (Zhang et al, 2018). Consistent with this notion, Brl1 or Brr6 mutants affecting the AαH exhibit reduced interactions with their wild-type partner protein (Fig. EV1). Screening *BRL1* for cold-sensitive alleles, particularly within the region encoding the N-terminus (see below), may identify mutants that disrupt its early role in NPC assembly.

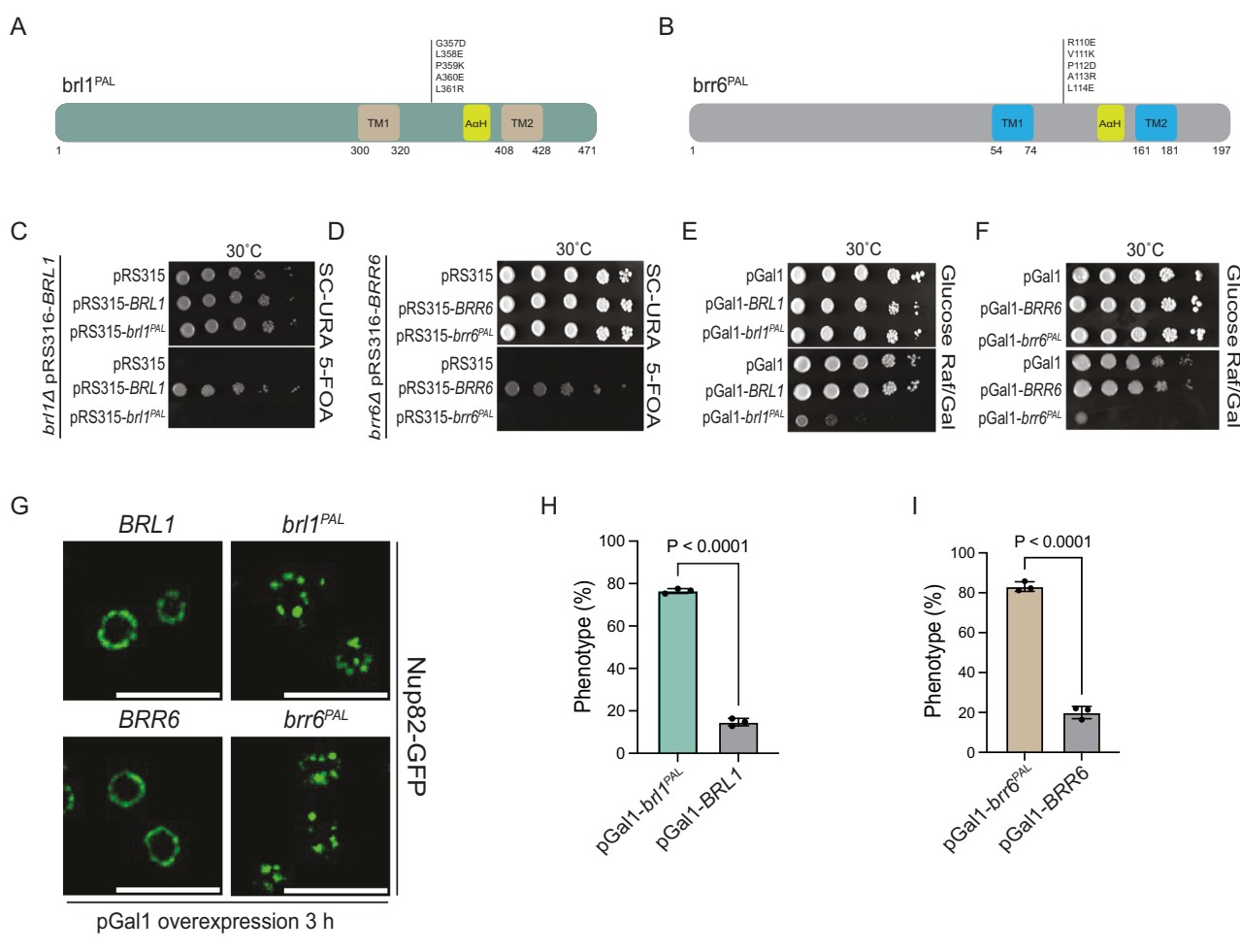

**Figure 7. The PAL motif of Brl1 and Brr6 is essential for INM and ONM fusion.**

(A, B) Summary of the mutations that were introduced in the Brl1 and Brr6 PAL mutant proteins. (C, D) The *brl1^PAL^* and *brr6^PAL^* mutants are non-functional. *BRL1* and *BRR6* shuffle strains were transformed with the indicated *LEU2*-based pRS315 plasmids. Tenfold serial dilutions of transformants were tested for growth on SC–URA and 5-FOA plates. Three independent experiments; one representative image is shown. (E, F) The indicated pGal1 plasmids were transformed into wild-type yeast cells. Tenfold serial dilutions of transformants were grown with the repressing glucose or the inducing galactose at 30 °C, respectively. One representative of three independent experiments is shown. (G) Expression of pGal1-*BRL1*, pGal1-*brl1^PAL^*, pGal1-*BRR6*, or pGal1-*brr6^PAL^* were induced in yeast strains carrying *NUP82-GFP* for 3 h at 30 °C by the addition of galactose and analyzed by fluorescence microscopy. Images are representative of three independent experiments. Size bar: 5 μm. (H, I) Quantification of the fluorescence data shown in (G). Data are pooled from three independent experiments; the total number of analyzed cells are 177 and 182 for pGal1-*BRL1* and pGal1-*brl1^PAL^*, respectively (H); the total number of analyzed cells are 195 and 224 for pGal1-*BRR6* and pGal1-*brr6^PAL^*, respectively (I). Statistical test: unpaired two-tailed *t* test. Data shows mean with SD. $P = 8.4E-7$ for the statistical significance between pGal1-*brl1^PAL^* and pGal1-*BRL1* condition (H); $P = 9.6E-6$ for the statistical significance between pGal1-*brr6^PAL^* and pGal1-*BRR6* condition (I). (J) EM analysis of cells expressing pGal1-*BRL1* or pGal1-*brl1^PAL^* using the same conditions as in (G) revealed accumulation of superherniations at the NE. The pGal1-*BRR6* and pGal1-*brr6^PAL^* data are shown in Fig. EV3. Representative EM images are shown. Size Bars: 200 nm in the row showing nuclei; 50 nm in the enlargements with the NPCs; 200 nm in the enlargements showing the superherniations (SH). The pink asterisks highlight NPCs in pGal1-*BRL1* cells. N nucleus, V vacuole, NPC nuclear pore complex, SH superherniation. Source data are available online for this figure.

The ability of the Brl1 N-terminus to bind the Nic96 complex (Fig. 6), an essential component of the early stages of NPC assembly (Onischenko et al, 2020), together with the similar NPC assembly defects observed in *brr6^L145E^* and *nic96(ts)* mutant cells and the genetic interaction between *brr6-1* (Arg110Lys) and the *Δnic96(532–839)* allele (de Bruyn Kops and Guthrie, 2001; Grandi et al, 1995; Zabel et al, 1996), raises the possibility that the early function of Brr6 is carried out in complex with Brl1 and involves interaction with Nic96. Because Nic96 is mis-localized in cells expressing pGal1-*brr6^L145E^* and pGal1-*brr6^F152E^* (Fig. 3B), we propose that the interaction of Nic96 with Brl1-Brr6 through the N-terminus of Brl1 is essential for the proper recruitment of the Nic96 complex to early NPC assembly sites. Interestingly, however, the N-Brl1–Nic96 interface is occluded by the interaction of Nic96 with Nup53–Nup59 in fully assembled NPCs, as suggested by AlphaFold predictions (Akey et al, 2023) (Appendix Fig. S3). This observation indicates that Brl1 interacts with Nic96 only during the initial steps of NPC assembly and not after Nic96 has been incorporated into the mature NPC.

As more Nups are deposited at these early NPC assembly sites, the INM begins to deform inward into the perinuclear space region. This membrane deformation reduces the local distance between the INM and ONM from approximately 22 nm to around 15 nm (Figs. EV4 and 8). We propose that this decrease in distance is sufficient to allow head-to-head interactions between the PAL regions of Brl1 on the INM and Brr6 on the ONM. This model is supported by the INM/ONM fusion defect of *brl1^PAL^* and *brr6^PAL^* mutants, which induce a distinctive form of superherniation characterized by pronounced NE deformations (Fig. 7). The nucleoporin Nsp1 resides on the nuclear side of these super-herniations, indicating that they originate from NPC assembly sites (Appendix Fig. S4).

Further support for this spatially regulated interaction model comes from the phenotypes of *brl1* and *brr6* mutants, in which the DAH was artificially extended from 9 nm to 16 nm in Brl1 and from 7 nm to 12 nm in Brr6. In these extended Brl1 and Brr6 variants, the head-to-head DAH interactions between Brl1^ePNS^–Brr6 and Brl1–Brr6^ePNS^ approach or exceed the critical ~22 nm distance between the INM and ONM (Fig. 8). The extended *brl1^ePNS^* and *brr6^ePNS^* mutants were non-functional in a shuffle strain, but their overexpression caused a severe toxic growth defect (Figs. 8

and EV5) with defective NPC distribution along the NE, evidence of NE disintegration by EM and the mis-localization of the normally nuclear GFP–NLS and the cytoplasmic Rpl25–GFP. Interestingly, destabilizing the DAH by mutating the conserved cysteines to alanine in brl1^ePNS^ and brr6^ePNS^ resulted in a markedly different overexpression phenotype, characterized by superhernia-tions, swollen NE, and ER sheets within the nucleus (Appendix Fig. S6). This phenotypic change is consistent with the idea that an extended and stabilized DAH is required to span the intermem-brane space, enabling Brl1 and Brr6 to interact via their PAL motifs. While the DAH/PAL domains likely mediate the bridging function between Brl1 and Brr6 across the intermembrane space, the AαH of Brl1 may promote INM–ONM fusion once the two membranes are in close proximity. An additional possibility is that membrane remodeling is facilitated by the formation of Brl1–Brr6 oligomers (see below), which deform the membrane and ultimately promote fusion of the INM and ONM, as observed in other systems (Beiter and Voth, 2024).

Whether oligomerization of Brl1 and Brr6 is required to promote INM and ONM fusion during NPC assembly remains unclear. AlphaFold predicts an assembly in which two octameric rings, Brl1 at the INM and Brr6 at the ONM, interact via their PAL motif containing DAH tips (Appendix Fig. S1C), a model that is particularly compelling given the eightfold symmetry of NPCs. Oligomerization of Brl1 and Brr6 would enhance their interactions across the INM and ONM. In addition, the octameric Brl1-Brr6 configuration could facilitate the recruitment of eight Nic96 complexes to the INM via N-Brl1 interactions during NPC assembly. However, due to the relatively low confidence scores of the AlphaFold predictions and the unknown composition of Brl1-containing complexes in the native gels (Appendix Fig. S1D,E), further experimental analysis is required to confirm the existence and functional relevance of such oligomers for INM/ONM fusion.

Homologs of Brl1 and Brr6 are found in organisms that retain the NE during mitosis, suggesting that the Brl1/Brr6 NE fusion machinery is conserved (Tamm et al, 2011). Interestingly, a remote-homology search identified CLCC1 as a putative Brl1/Brr6 homolog with potential roles in NPC assembly (preprint: Mathiowetz et al, 2024). It will be intriguing to determine which functional domains within CLCC1 mediate Nup recruitment and INM/ONM fusion.

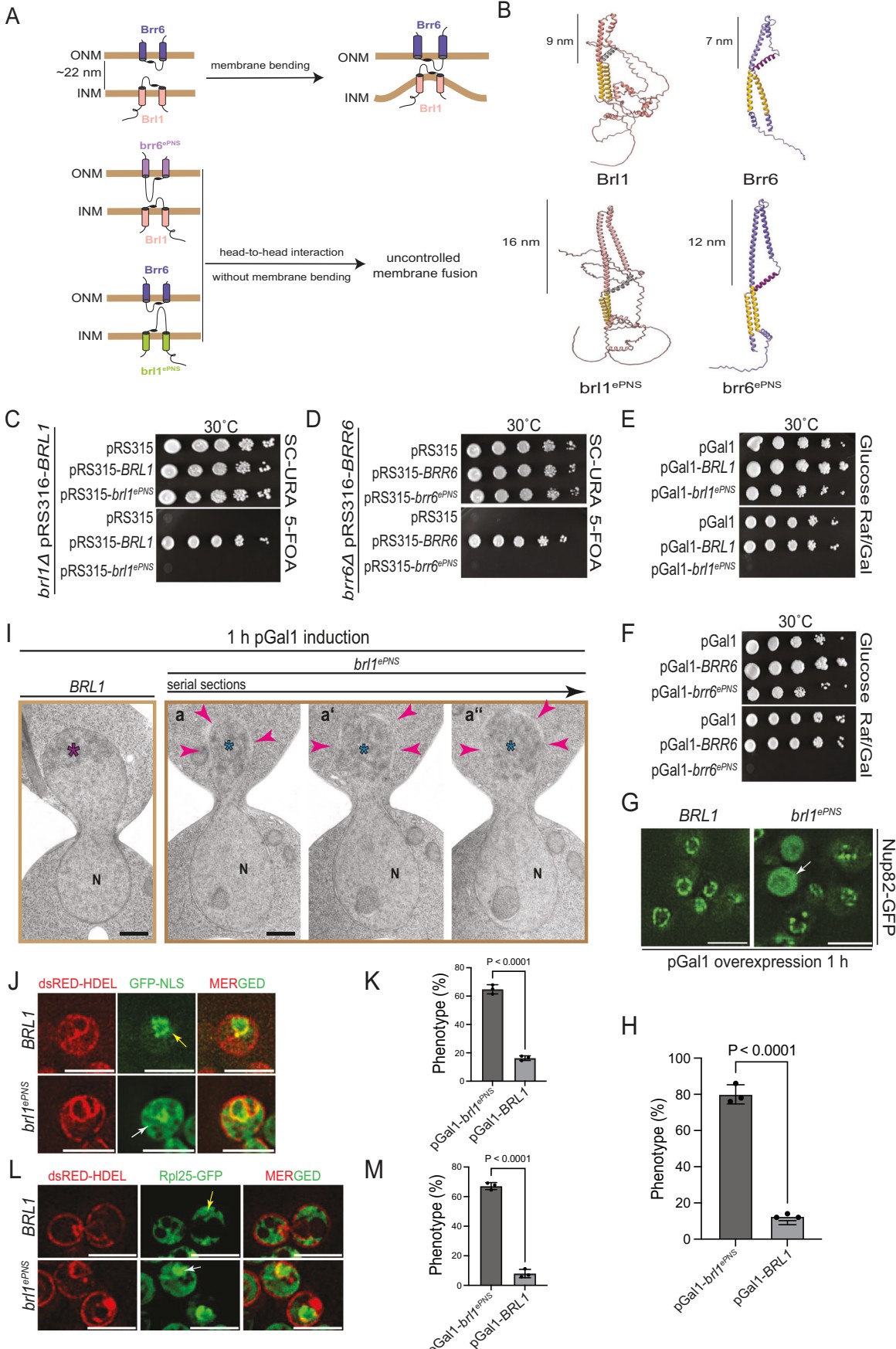

◀

**Figure 8.  Brl1 and Brr6 promote NE fusion.**

(A) Model illustrating the length-dependent interaction of the DAHs of Brl1 and Brr6 bridging the perinuclear space region. (B) AlphaFold structural predictions of the extended regions of Brl1 (brl1$^{ePNS}$) and Brr6 (brr6$^{ePNS}$) in comparison to wild-type Brl1 and Brr6. The lengths of the PNS domains were determined using AlphaFold predictions. (C, D) brl1$^{ePNS}$ and brr6$^{ePNS}$ constructs are non-functional in a plasmid shuffle assay at 30 °C, whereas wild-type *BRL1* and *BRR6* are functional. Tenfold serial dilutions; three independent experiments. (E, F) Overexpression of pGal1-brl1$^{ePNS}$ and pGal1-brr6$^{ePNS}$ is toxic to cells, while overexpression of wild-type *BRL1* and *BRR6* does not affect cell growth. Tenfold serial dilutions; three independent experiments. (G) pGal1-*BRL1* and pGal1-brl1$^{ePNS}$ were expressed in *NUP82-GFP* cells following 1 h galactose induction. Localization of Nup82-GFP was analyzed by fluorescence microscopy. The arrow highlights a cell with uniform Nup82-GFP staining. Three independent experiments. Size bars: 5 µm. Note, the corresponding experiment with pGal1-brr6$^{ePNS}$ and pGal1-*BRR6* is shown in Fig. EV5. (H) Quantification of (G). Total number of analyzed cells: 150 each for both pGal1-*BRL1* and pGal1-brl1$^{ePNS}$; statistical test: unpaired two-tailed *t* test. Data show mean with SD; ($P = 2.7E{-}5$). (I) EM analysis of cells expressing pGal1-*BRL1* or pGal1-brl1$^{ePNS}$ after 1 h of galactose induction. (a–a″) show 80 nm serial sections of the same anaphase cell. Arrowheads indicate regions of the NE undergoing disintegration. Purple asterisk marks compact chromatin; blue asterisks denote decondensed chromatin adjacent to disintegrating NE regions. N nucleus. Size bar: 500 nm. The corresponding experiment with pGal1-*BRR6* and pGal1-brr6$^{ePNS}$ is shown in Fig. EV5C. Fig. EV5D shows EM analysis of pGal1-*BRL1*, pGal1-brl1$^{ePNS}$, pGal1-*BRR6* and pGal1-brr6$^{ePNS}$ cells in interphase. (J) pGal1-brl1$^{ePNS}$ overexpression affects localization of GFP–NLS in the nucleus. *dsRED-HDEL GFP-NLS* cells with pGal1-*BRL1* or pGal1-brl1$^{ePNS}$ were incubated for 1 h with galactose. Localization of GFP–NLS was analyzed by fluorescence microscopy. The yellow arrow indicates nuclear localization of GFP–NLS; the white arrowhead indicates cytoplasmic localization. (K) Cells from (J) were quantified for the localization of GFP–NLS. Three independent experiments with total number of analyzed cells 224 and 221 for pGal1-*BRL1* and pGal1-brl1$^{ePNS}$, respectively; Statistical test: unpaired two-tailed *t* test. Data shows mean with SD ($P = 2.1E{-}5$). (L) As (J) but with *dsRED-HDEL RPL25-GFP* cells. The yellow arrow indicates cytoplasmic Rpl25–GFP; the white arrow indicates nuclear Rpl25–GFP. (M) Cells from (L) were quantified for the localization of Rpl25–GFP. Three independent experiments with total no. of analyzed cells 238 and 272 for pGal1-*BRL1* and pGal1-brl1$^{ePNS}$, respectively; Statistical test: unpaired two-tailed *t* test. Data shows mean with SD ($P = 1.0E{-}5$). Source data are available online for this figure.

# Methods

## Reagents and tools table

| Experimental models—yeast strains (genotype) | Reference or source | Identifier or catalog number |
| --- | --- | --- |
| *MATa ura3-52 trp1Δ63 his3Δ200 leu2Δ1* | Schiebel Lab | ESM356-1 |
| *MATa ura3-52 leu2Δ1 his3Δ200 trp1Δ63::YIPlac-dsRED-HDEL-NatMX* | This study | YZW292 |
| *MATa ura3-52 leu2Δ1 his3Δ200 trp1Δ63 NUP82- tdTomato-HpHMX6* | This study | AAK0042 |
| *MATa ura3-52 leu2Δ1 his3Δ200 trp1Δ63 NSP1- tdTomato-HphMX6* | This study | YZW815 |
| *MATa ura3-52 leu2Δ1 his3Δ200 trp1Δ63 NUP42- tdTomato-HphMX6* | This study | YJV075 |
| *MATa ura3-52 leu2Δ1 his3Δ200 trp1Δ63 NUP116-tdTomato-HphMX6* | This study | YJV081 |
| *MATa ura3-52 leu2Δ1 his3Δ200 trp1Δ63::YIPlac-dsRED-HDEL-NatMX RPL25-yeGFP-klTRP1* | This study | AAK0031 |
| *MATa ura3-52 leu2Δ1 brr6-751* | Dr. I. Hagan | YZW14 |
| *MATa ura3-52 leu2Δ1 brr6-19* | Dr. I. Hagan | YZW13 |
| *MATa ura3-52 leu2Δ1 brr6-721* | Dr. I. Hagan | YZW15 |
| *MATa ura3-52 leu2Δ1 brr6-732* | Dr. I. Hagan | YZW16 |
| *MATa ura3-52 leu2Δ1 brr6-733* | Dr. I. Hagan | YZW17 |
| *MATa ura3-52 leu2Δ1 brr6-69* | Dr. I. Hagan | YZW19 |
| *MATa ura3-52 leu2Δ1 brr6Δ::KANMX pRS316-BRR6* | Dr. I. Hagan | YZW05 |
| *MATa ura3-52 leu2Δ1 trp1Δ63 brl1Δ::KANMX pRS316-BRL1* | Dr. I. Hagan | YZW07 |
| *MATa ura3-52 leu2Δ1 his3Δ200 trp1Δ63 GLE1-tdTomato-HphMX6* | This study | YZW079 |
| *MATa ura3-52 leu2Δ1 his3Δ200 trp1Δ63 GLE2-tdTomato-HphMX6* | This study | YZW078 |
| *MATa ura3-52 leu2Δ1 his3Δ200 trp1Δ63 POM152-tdTomato-HphMX6* | This study | AAK0046 |
| *MATa ura3-52 leu2Δ1 his3Δ200 trp1Δ63 NUP170-yeGFP-klTRP1* | This study | AAK0047 |
| *MATa ura3-52 leu2Δ1 his3Δ200 trp1Δ63 NUP60-yeGFP-klTRP1* | This study | AAK0048 |
| *MATa ura3-52 leu2Δ1 his3Δ200 trp1Δ63 NUP84-yeGFP-klTRP1* | This study | AAK0050 |
| *MATa ura3-52 leu2Δ1 his3Δ200 trp1Δ63 NUP82-yeGFP-klTRP1* | This study | AAK0052 |
| *MATa ura3-52 leu2Δ1 brr6Δ::KANMX, pRS303N-brr6$^{L145E}$* | This study | SMBK001 |
| *MATa ura3-52 leu2Δ1 brr6Δ::KANMX, pRS303N-brr6$^{L145E}$ NUP159-yeGFP-HpHMX6* | This study | SMBK002 |
| *MATa ura3-52 leu2Δ1 brr6Δ::KANMX, pRS303N-brr6$^{L145E}$ POM152-yeGFP-HpHMX6* | This study | SMBK003 |
| *MATa ura3-52 leu2Δ1 his3Δ200 trp1Δ63 NUP159-yeGFP-klTRP1* | Schiebel lab | AK821 |
| *MATa ura3-52 leu2Δ1 his3Δ200 trp1Δ63 NDC1-yeGFP-klTRP1* | Schiebel lab | AK819 |
| *MATa ura3-52 leu2Δ1 his3Δ200 trp1Δ63 POM34-yeGFP-klTRP1* | Schiebel lab | AK721 |
| *MATa ura3-52 leu2Δ1 his3Δ200 trp1Δ63 NUP49-yeGFP-klTRP1* | Schiebel lab | LD046 |
| *MATa ura3-52 leu2Δ1 his3Δ200 trp1Δ63 NUP2-mCherry-KanMX6* | Schiebel lab | AK949 |
| *MATa ura3-52 leu2Δ1 his3Δ200 trp1Δ63 NIC96-yeGFP-klTRP1* | Schiebel lab | AK820 |
| *MATa ura3-52 leu2Δ1 his3Δ200 trp1Δ63 NUP120-yeGFP-klTRP1* | Schiebel lab | AK969 |
| *MATa ura3-52 leu2Δ1 his3Δ200 trp1Δ63 NUP133-yeGFP-klTRP1* | Schiebel lab | AK720 |
| **Recombinant DNA—plasmids** | **Reference or source** | **Identifier or catalog number** |
| *p426GAL1-BRL1* | This study | pZW78 |
| *p425GAL1-brl1$^{F391E}$* | This study | pAAK0014 |
| *pRS315-BRL1* | Zhang et al, 2018 | pZW24 |
| *p426GAL1-brr6$^{L145E}$* | This study | pAAK0047 |

| Experimental models—yeast strains (genotype) | Reference or source | Identifier or catalog number |
|---|---|---|
| p426GAL1-brr6$^{F152E}$ | This study | pAAK0048 |
| pRS315-BRR6 | Zhang et al, 2018 | pZW63 |
| pRS315-brr6$^{L14SE}$ | This study | pAAK0053 |
| pRS315-brr6$^{F152E}$ | This study | pAAK0054 |
| p426GAL1-BRR6-10HIS | Zhang et al, 2018 | pZW69 |
| p426GAL1-BRR6$^{AαH}$-yeGFP-8HIS | This study | pAAK0051 |
| p426GAL1-brr6$^{L14SE\ AαH}$-yeGFP-8HIS | This study | pAAK0052 |
| pRS315-pADH1-BRR6 | Zhang et al, 2018 | pZW83 |
| pRS315-pADH1-BRL1 | Zhang et al, 2018 | pZW59 |
| p425-GAL1-BRL1 | Zhang et al, 2018 | pZW77 |
| pRS315-brr6$^{1-53}$-brl1$^{300-471}$ | This study | pJV056 |
| pRS315-brr6$^{1-181}$-brl1$^{429-471}$ | This study | pJV059 |
| pRS315-brl1$^{1-428}$-brr6$^{182-197}$ | This study | pJV058 |
| pRS315-brl1$^{1-299}$-brr6$^{54-197}$ | This study | pJV057 |
| p426GAL1-BRL1-6HA | This study | pSMBK0019 |
| p426GAL1-brr6$^{1-53}$-brl1$^{300-471}$-6HA | This study | pSMBK0020 |
| pRS315-brr6$^{PAL}$ | This study | pSMBK0015 |
| pRS315-brl1$^{PAL}$ | This study | pSMBK0016 |
| p426GAL1-brr6$^{PAL}$ | This study | pSMBK0017 |
| p426GAL1-brl1$^{PAL}$ | This study | pSMBK0018 |
| pRS315-brr6$^{ePNS}$ | This study | pSMBK0007 |
| pRS315-brl1$^{ePNS}$ | This study | pSMBK0008 |
| p426GAL1-brr6$^{ePNS}$ | This study | pSMBK0009 |
| p426GAL1-brl1$^{ePNS}$ | This study | pSMBK0010 |
| pRS315-brr6$^{ePNS-CA}$ | This study | pSMBK0011 |
| pRS315-brl1$^{ePNS-CA}$ | This study | pSMBK0012 |
| p426GAL1-brr6$^{ePNS-CA}$ | This study | pSMBK0013 |
| p426GAL1-brl1$^{ePNS-CA}$ | This study | pSMBK0014 |
| p425-GAL1-NLS-GFP | This study | pSMBK0021 |
| pEG(KG)-GST-nSEC1 | Parisotto et al, 2012 | pEG(KG) |
| pEG(KG)-BRL1$^{1-200}$ | This study | pSMBK0022 |
| pCA528-6HIS-NIC96 | This study | pSMBK0023 |

## Yeast strains, plasmids, and culture conditions

All strains and plasmids used for this study are listed in the reagent table. Yeast strains are derivatives of ESM356-1 (MATa *ura3-52 trp1Δ63 his3Δ200 leu2Δ1*). Endogenous gene tagging and gene deletions were performed using PCR-based integration methods as described by (Janke et al, 2004). Strains were cultured in YPD (yeast extract, peptone, and glucose), SC (synthetic complete) medium, or SC selective medium lacking specific amino acids or bases, as

indicated. For induction of proteins under control of the pGal1 promoter, cells were grown in SC medium containing 2% raffinose and supplemented with 2% galactose. Growth assays were performed by growing cells overnight in selective medium, adjusting cultures to an $OD_{600}$ of 1.0, and spotting tenfold serial dilutions onto selective agar plates. Plates were incubated at the specified temperatures. For protein analysis by immunoblotting, yeast extracts were prepared using alkaline lysis followed by TCA precipitation (Knop et al, 1999).

## Electron microscopy

High-pressure frozen yeast samples were processed for EM analysis as described in the following: cells were collected onto a 0.45-μm polycarbonate filter (Millipore) using vacuum filtration and subsequently frozen using a high-pressure freezing device (HPM010, Abra-Fluid, Switzerland). Freeze substitution was performed in an EM-AFS2 system (Leica Microsystems, Vienna, Austria) using a solution containing 0.2% uranyl acetate and 1% water in anhydrous acetone. Samples were stepwise infiltrated with Lowicryl HM20 resin (Polysciences, Inc., Warrington, PA) starting at −90 °C. Polymerization was carried out under UV light for 48 h at −45 °C, followed by gradual warming to 20 °C.

Resin-embedded yeast cells were serially sectioned at a thickness of 80 nm using a Reichert Ultracut S microtome (Leica Instruments, Vienna, Austria). The sections were subsequently mounted onto slot grids coated with a thin plastic support film for analysis. Sections were post-stained with 3% uranyl acetate and lead citrate and imaged using a JEOL JEM-1400 transmission EM (JEOL Ltd., Tokyo, Japan), operated at 80 kV and equipped with a 4k × 4k digital camera (F416, TVIPS, Gauting, Germany). Image brightness and contrast were adjusted using Fiji (NIH, Bethesda, MD).

For immunogold labeling, a primary antibody against Nsp1 was used. Serial sections were mounted on slot grids, incubated with blocking buffer (1.5% BSA, 0.1% fish skin gelatin in phosphate-buffered saline [PBS]), followed by incubation with the primary antibody (rabbit anti-Nsp1). After washing in PBS, sections were incubated with a secondary linker antibody (rabbit anti-mouse), and labeled using protein A–gold conjugates (15 nm; Utrecht University, Utrecht, Netherlands). Post-staining was performed as above using 3% uranyl acetate and lead citrate.

## Immunoprecipitation

Cells equivalent to 25 $OD_{600}$ units were harvested and resuspended in lysis buffer (20 mM Tris–Cl, pH 8.0; 150 mM NaCl; 5 mM $MgCl_2$; 10% glycerol) supplemented with 10 mM NaF, 60 mM β-glycerophosphate, 1 mM PMSF, and 1 tablet of EDTA-free protease inhibitor cocktail (Roche) per 50 ml. Cell lysis was performed using a FastPrep homogenizer (MP Biomedicals) with the addition of glass beads (BioSpec Products). After lysis, Triton X-100 was added to a final concentration of 0.5%, and samples were incubated on ice for 10 min.

The lysates were clarified by centrifugation to separate soluble proteins from debris. The supernatant was incubated with GFP-Trap agarose beads (Chromotek) for 2 h at 4 °C with gentle rotation. Beads were then washed three times with lysis buffer containing 0.1% Triton X-100 and twice with wash buffer (20 mM Tris–Cl, pH 8.0; 150 mM NaCl; 5 mM $MgCl_2$). Bound proteins

were eluted by boiling the beads in 50 μl of 2× Laemmli sample buffer for 5 min at 95 °C and analyzed by SDS-PAGE and immunoblotting.

## Fluorescence microscopy

Cell imaging was performed using a DeltaVision RT system (Applied Precision Ltd.) based on an Olympus IX71 microscope and equipped with a Photometrics CoolSnap HQ camera (Roper Scientific), a 100×/1.4 NA Super-Plan Apochromat oil immersion objective (Olympus), and a four-color Standard Insight SSI module light source. The system was controlled via a workstation running the CentOS operating system and softWoRx software (Applied Precision Ltd.).

Images were acquired using either the GFP or mCherry channels under identical exposure times and illumination settings to enable direct comparison across samples. A single optical z-stack was collected per field of view and used for analysis. Image deconvolution was performed using softWoRx, and further image processing was conducted with ImageJ (National Institutes of Health, Bethesda, MD).

Most imaging experiments and quantitative analyses were independently repeated three times. Data analysis and statistical evaluation were performed using GraphPad Prism software.

## Western blot

PAGE-separated proteins were transferred onto a PVDF membrane (Immobilon, Millipore) using a semi-dry method. After transfer, the membrane was blocked in blocking buffer for 50–60 min at room temperature, and incubated overnight in primary antibody, diluted in blocking buffer, at 4 °C. Next, it was incubated in HRP-conjugated secondary antibody diluted in blocking buffer for 1 h at room temperature. The membrane was washed three times, 5 min each, with TBS-T before being imaged on the LAS-4000 camera system (FujiFilm) using freshly mixed ECL 1 and 2 solutions in a 1:1 ratio.

## Blue-native PAGE

Higher-order oligomeric complexes of Brl1 were detected by Blue-native polyacrylamide gel electrophoresis (Schägger et al, 1994). For the sample preparation, cells equivalent to 25 $OD^{600}$ units were harvested and resuspended in lysis buffer (20 mM Tris–Cl, pH 8.0; 150 mM NaCl; 5 mM $MgCl_2$; 10% glycerol) supplemented with 10 mM NaF, 60 mM β-glycerophosphate, 1 mM PMSF, and 1 tablet of EDTA-free protease inhibitor cocktail (Roche) per 50 ml. Cell lysis was performed using a FastPrep homogenizer (MP Biomedicals) with the addition of glass beads (BioSpec Products). After lysis, Triton X-100 was added to a final concentration of 1%, and samples were incubated on ice for 10 min. The lysates were clarified by centrifugation to separate soluble proteins from debris. The supernatant was mixed with 5× sample buffer, run on blue-native page, and further analyzed by western blotting.

## AlphaFold structure predictions

Multimeric structures of Brl1, Brr6, and other proteins were predicted using the AlphaFold2 and AlphaFold3 servers (Abramson et al, 2024). The predicted structures were analyzed using the tool ChimeraX (Meng et al, 2023) based on the predicted error plot with different scores, such as predicted local distance difference test (pLDDT), predicted template modeling score (pTM), and interface predicted template modeling (ipTM).

## Spot assay

Yeast strains were grown overnight in appropriate media and at the appropriate temperature. In all, 1 $OD_{600}$ was collected by centrifugation at $3200 \times g$ for 2 min. Cell pellets were resuspended in 1 ml of sterile PBS and serially diluted 10, 100, 1000, and 10,000 times. In total, 2–3 μl of the cell suspensions were spotted on appropriate plates and incubated at desired temperatures for 2–4 days.

## Statistical analysis

PRISM v.10.6.1 software (GraphPad) was used for the statistical analysis. Unpaired two-tailed $t$ test and one-way ANOVA were used for the statistical significance test with a significance level of $P \leq 0.05$.

## Antibodies

The antibodies of this study were used as follow: mouse anti-Nsp1 (immuno-EM, 1:100; ab4641; Abcam), rabbit anti-Tub2 (immunoblot, 1:1000; made in-house), rabbit anti-N-Brl1 (gift from Dr. I. Hagan, Manchester; 1:500) and rabbit anti-Nup57 antibodies (gift from Ed Hurt, Heidelberg; 1:100). Rabbit anti-HA (immunoblot, 1:500; Protein-Tech), Mouse anti-penta-His (immunoblot, 1:1000; QIAGEN), Goat anti-GST (immunoblot, 1:1000; Cytiva), Rabbit anti-GFP (immunoblot, 1:1000; Protein-Tech), Anti-mouse HRP (1:5000; Jackson ImmunoResearch), Anti-rabbit HRP (1:5000; Jackson ImmunoResearch), Anti-Goat HRP (1:5000; Jackson ImmunoResearch).

# Data availability

This study includes no data deposited in external repositories. An earlier version of this manuscript was deposited in bioRxiv on 23.07.25 (https://doi.org/10.1101/2025.07.22.665954).

The source data of this paper are collected in the following database record: biostudies:S-SCDT-10_1038-S44318-026-00718-y.

# Peer review information

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

## Acknowledgements

This work was funded by the Deutsche Forschungsgemeinschaft (CRC1638). We thank Dr. Iain Hagan, University of Manchester, for kindly providing the conditional lethal *brr6(ts)* cells and N-Brl1 antibodies. We thank Ed Hurt (BZH, Heidelberg University) and Martin Beck (Max Planck Institute of Biophysics, Frankfurt) for the anti-Nup57 antibody and Jörg Malsam (BZH, Heidelberg University) for the GST-nSec1 plasmid. We thank the EM core facility of Heidelberg University (EMCF) for their technical support and Uta Haselmann for assistance.

## Author contributions

**Sayan Mondal**: Conceptualization; Formal analysis; Methodology; Writing—original draft; Writing—review and editing. **Annett Neuner**: Formal analysis; Methodology; Writing—review and editing. **Azqa Ajmal Khan**: Methodology. **Jlenia Vitale**: Methodology. **Elmar Schiebel**: Conceptualization; Resources; Formal analysis; Supervision; Funding acquisition; Validation; Writing—original draft; Project administration; Writing—review and editing.

Source data underlying figure panels in this paper may have individual authorship assigned. Where available, figure panel/source data authorship is listed in the following database record: biostudies:S-SCDT-10_1038-S44318-026-00718-y.

## Funding

## Disclosure and competing interests statement

The authors declare no competing interests.

# Expanded View Figures

**Figure EV1. AlphaFold predictions of Brr6-Brl1 interactions.**

Extension of Fig. 1. (**A**) AlphaFold predicts full-lenh Brr6–Brl1 interactions mediated by the two AαH domains. The pTM and ipTM scores are indicated. (**B**) AlphaFold using the PNS domains predicts DAH/PAL region Brr6–Brl1 interactions. The pTM and ipTM scores are indicated. (**C**) AlphaFold predicts full-length brr6$^{L145E}$–Brl1 interactions via DAH/PAL but not the AαH domains. The pTM and ipTM scores are indicated. (**D**) AlphaFold predicts full-length Brr6–brl1$^{F391E}$ interactions via DAH/PAL but not the AαH domains. The pTM and ipTM scores are indicated. (**E**) AlphaFold predicts full-length brr6$^{L145E}$–brl1$^{F391E}$ interactions mediated by the two AαH domains. The pTM and pTM scores are indicated. (**F**) Immunoprecipitation experiment of brl1 and brr6 mutant proteins. Cells expressing the indicated *BRL1* and *BRR6* constructs were lysed and the Brl1-Brr6 interaction was analyzed by immunoprecipitation using GFP-binder. The immunoprecipitation was analyzed by immunoblotting with the indicated antibodies. Three independent experiments, one representative is shown. Source data are available online for this figure.

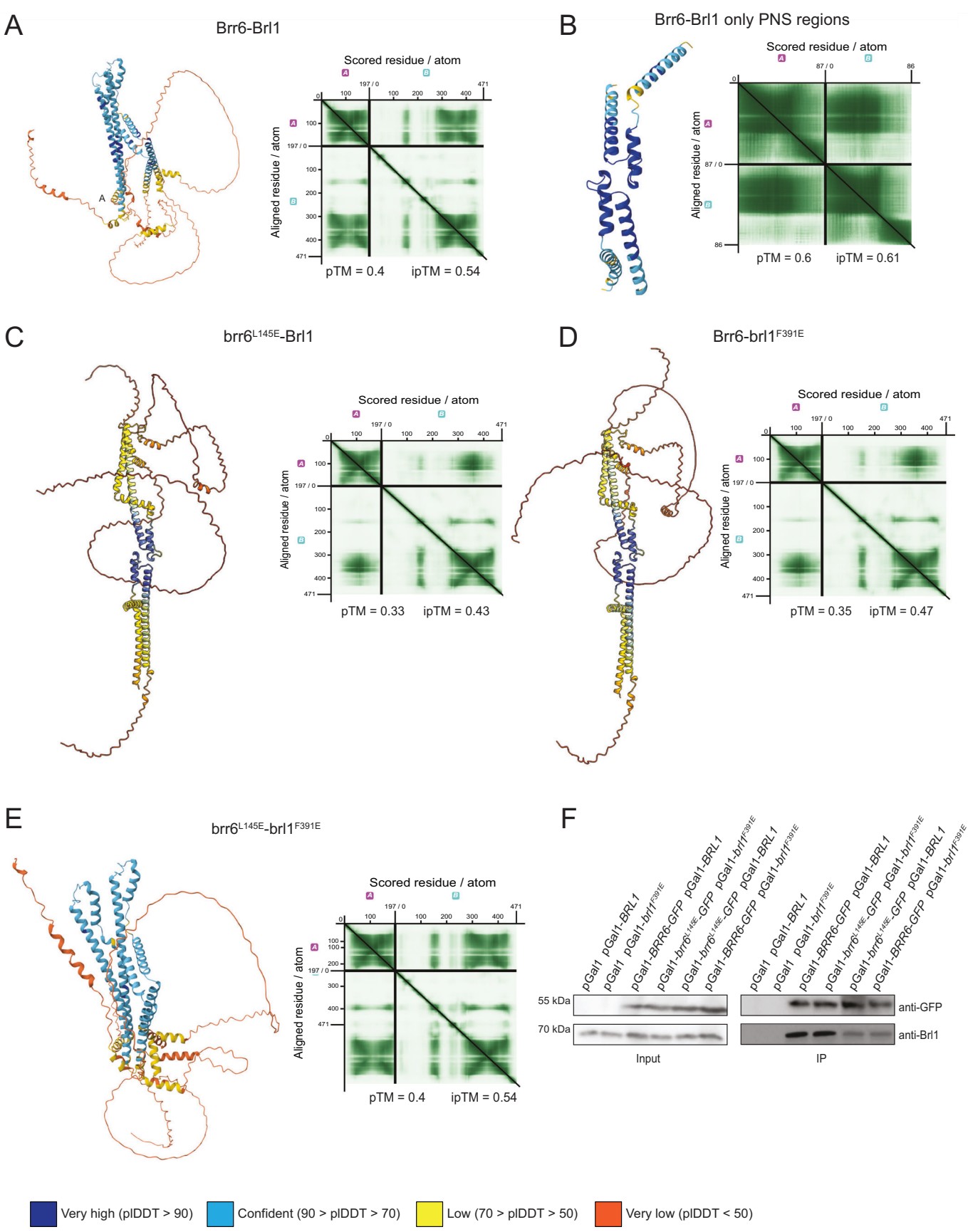

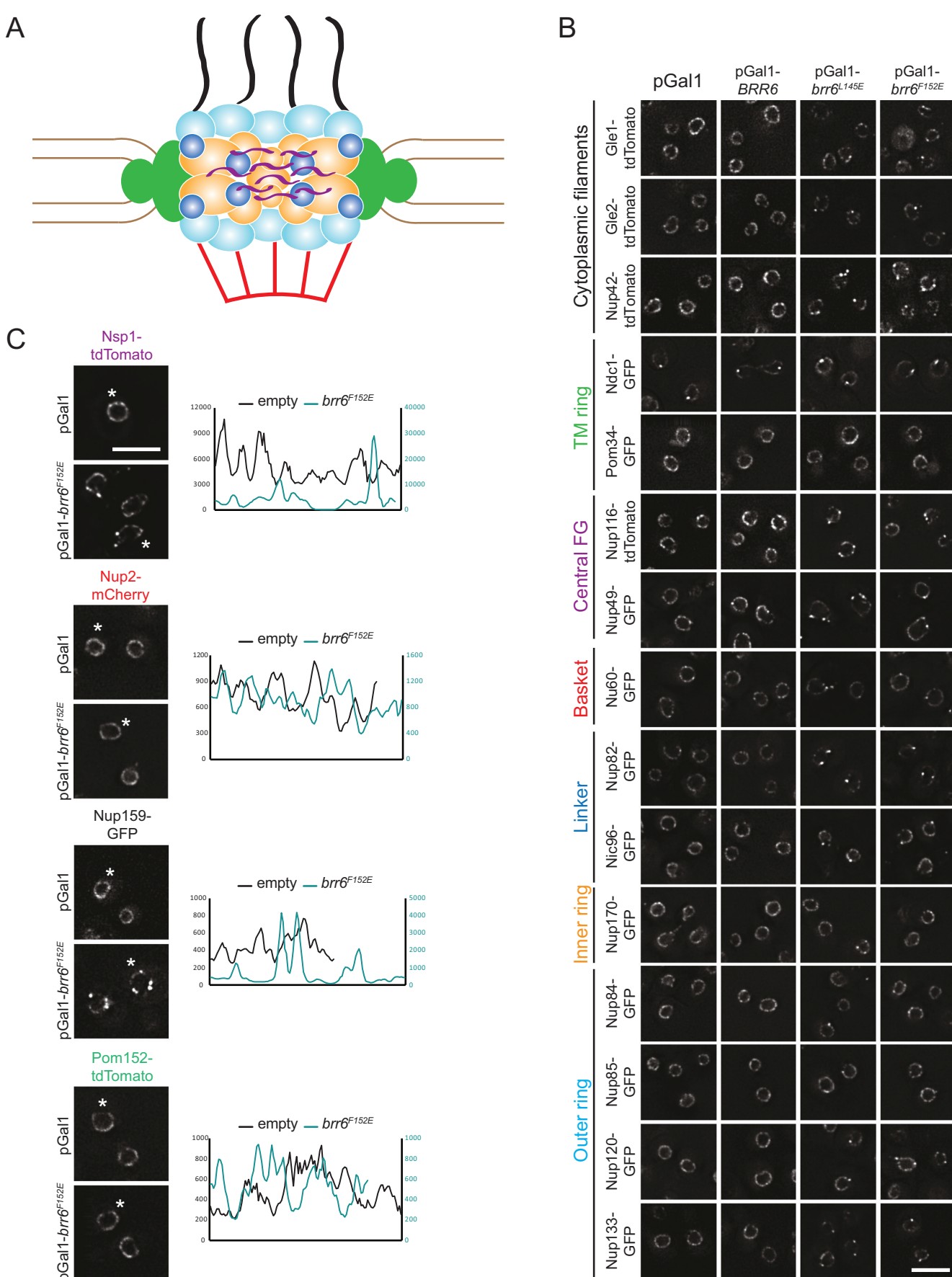

◀ **Figure EV2. Analysis of brr6^{L14SE} and brr6^{F152E} overexpression phenotypes.**

Extension of Fig. 3. (A) Schematic representation of the NPC with color-coded components: transmembrane ring (green), central FG-Nups (purple), nuclear basket Nups (red), linker Nups (blue), inner ring Nups (ochre), and outer ring Nups (light blue). (B) As in Fig. 3A using the indicated yeast strains expressing tdTomato- and GFP-tagged Nup constructs. Cells additionally carried pGal1, pGal1-BRR6, pGal1-brr6^{L14SE}, or pGal1-brr6^{F152E}, which were overexpressed for 3 h. See Fig. 3B for quantification of the phenotypes. Size bar: 5 μm. (C) Overexpression of brr6^{F152E} alters NE localization of Nsp1 and Nup159, whereas Nup2 and Pom152 are only minimally affected. Line scans on the right show NE distribution of the indicated Nups in cells carrying either empty pGal1 or pGal1-brr6^{F152E} plasmids. The cell that was selected for line scan is indicated by an asterisk. Cells were incubated for 3 h in raffinose/galactose medium to induce expression from the pGal1 promoter. Note, the NSP1-tdTomato pGal1-brr6^{F152E} cells (top) are identical to the cells in Fig. 3A (first column). Size bar: 5 μm. Source data are available online for this figure.

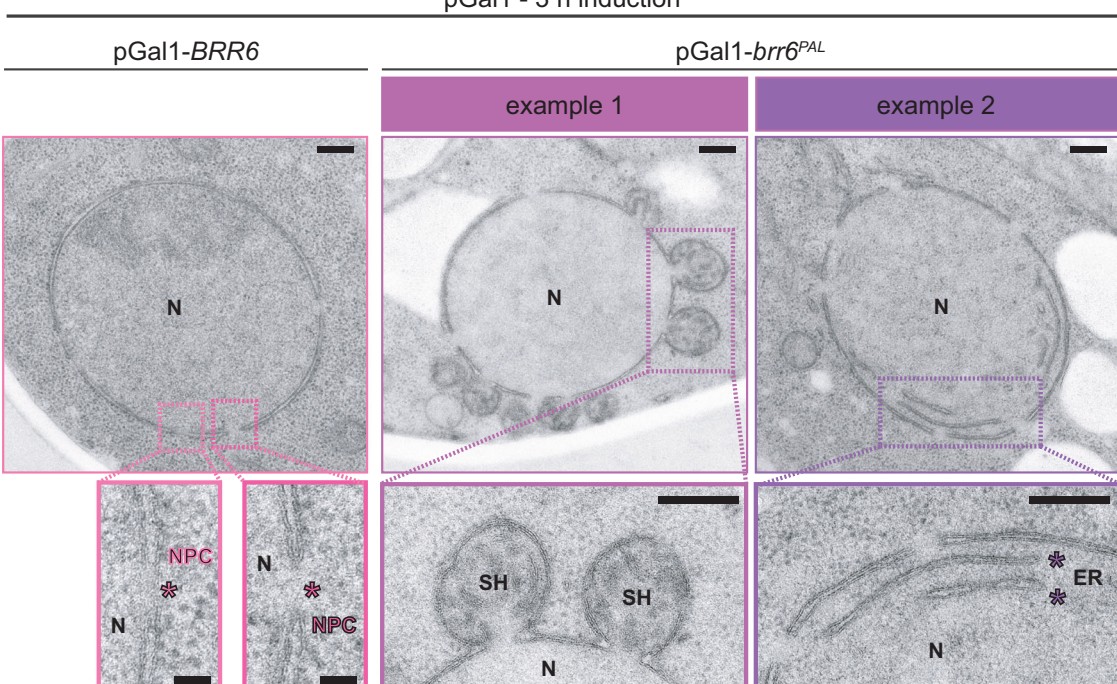

**Figure EV3.    Phenotypes of *brr6^PAL* mutant overexpression.**

Extension of Fig. 7. EM analysis of cells overexpressing pGal1-*BRR6* or pGal1-*brr6^PAL* using the same conditions as in Fig. 7J revealed accumulation of superherniations (SH) at the NE. The pGal1-*BRL1* or pGal1-*brl1^PAL* data are shown in Fig. 7J. Representative EM images are shown. Size Bars: 200 nm in the row showing nuclei; 50 nm in the enlargements with the NPCs; 200 nm in the enlargements showing the superherniations (SH) or ER inside the nucleus (example 2; purple asterisk). The pink asterisks highlight NPCs in pGal1-*BRR6* cells. ER endoplasmic reticulum, N nucleus, NPC nuclear pore complex, SH superherniation. Source data are available online for this figure.

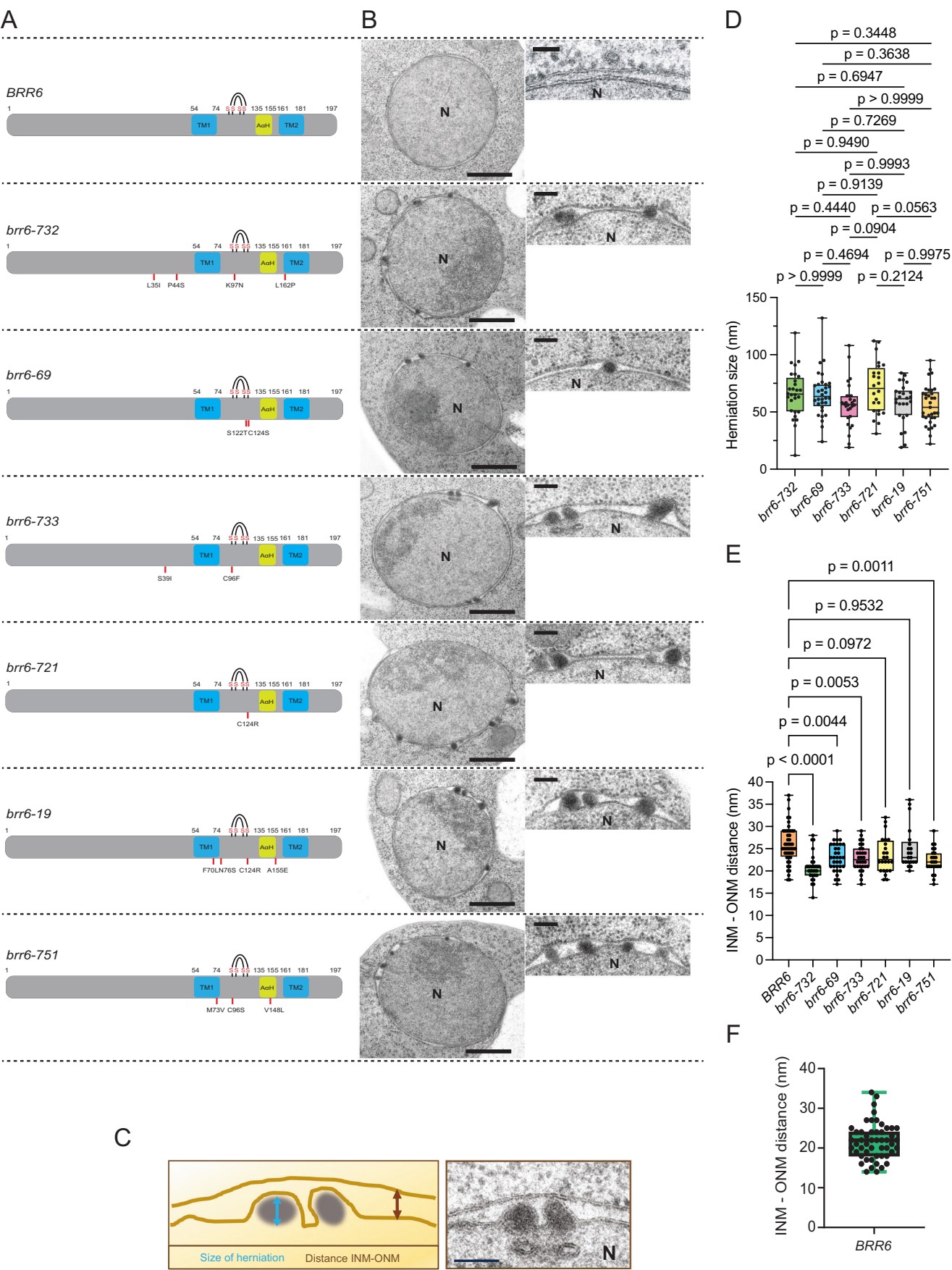

◄ **Figure EV4.   Brr6 also functions in INM/ONM fusion during NPC biogenesis.**

Extension of Fig. 7. (**A**) Overview of conditional lethal *brr6(ts)* alleles generated by mutagenic PCR (Zhang et al, 2018). The resulting *brr6(ts)* alleles were verified by sequencing and integrated into the endogenous *BRR6* locus using a pop-in pop-out strategy (Rothstein, 1991). The amino acid substitutions in the various *brr6(ts)* alleles are indicated. TM, AαH, and the four cysteines involved in disulfide bridge formation are also marked. Note, these mutants were previously only analyzed in the context of benzyl alcohol sensitivity (Zhang et al, 2018). (**B**) The wild-type and *brr6(ts)* mutants were shifted to 37 °C for 3 h, and their phenotype was analyzed by thin-section EM. All *brr6(ts)* mutants accumulated herniations at the NE. N nucleus. Size bar: 500 nm; magnified inset: 50 nm. (**C**) Scheme for the analysis of wild-type and *brr6(ts)* mutant cells. Size bar: 100 nm. Note that the micrograph next to the schematic is a re-display of a section from (**B**) for illustrative purposes. (**D**, **E**) Herniation size (**D**) and random INM–ONM distance (**E**) away from herniations were measured from EM sections in (**B**). In wild-type cells, INM–ONM distances were measured randomly away from NPCs. (**D**) The size of 24, 32, 28, 24, 26 and 33 herniations from 11, 10, 10, 8, 7, 8 cell sections of *brr6-732*, *brr6-69*, *brr6-733*, *brr6-721*, *brr6-19* and *brr6-751* cells was measured. (**E**) The INM–ONM distance from 16, 11, 11, 10, 8, 7 and 8 cell sections with three measurement per section was determined. Statistical test: One-way ANOVA. Box and Whisker plots shows median with interquartile range. $P = 2.2\text{E-8}$ for the statistical significance between *BRR6* and *brr6-732* condition (**E**). (**F**) The distance between the INM and ONM in wild-type cells grown at 30 °C was measured by EM. INM–ONM distances were measured randomly away from NPCs. Ultrathin sections from 16 cells with four measurement per section were analyzed, revealing an average INM–ONM spacing of 22 nm. Data shows median with interquartile range. Source data are available online for this figure.

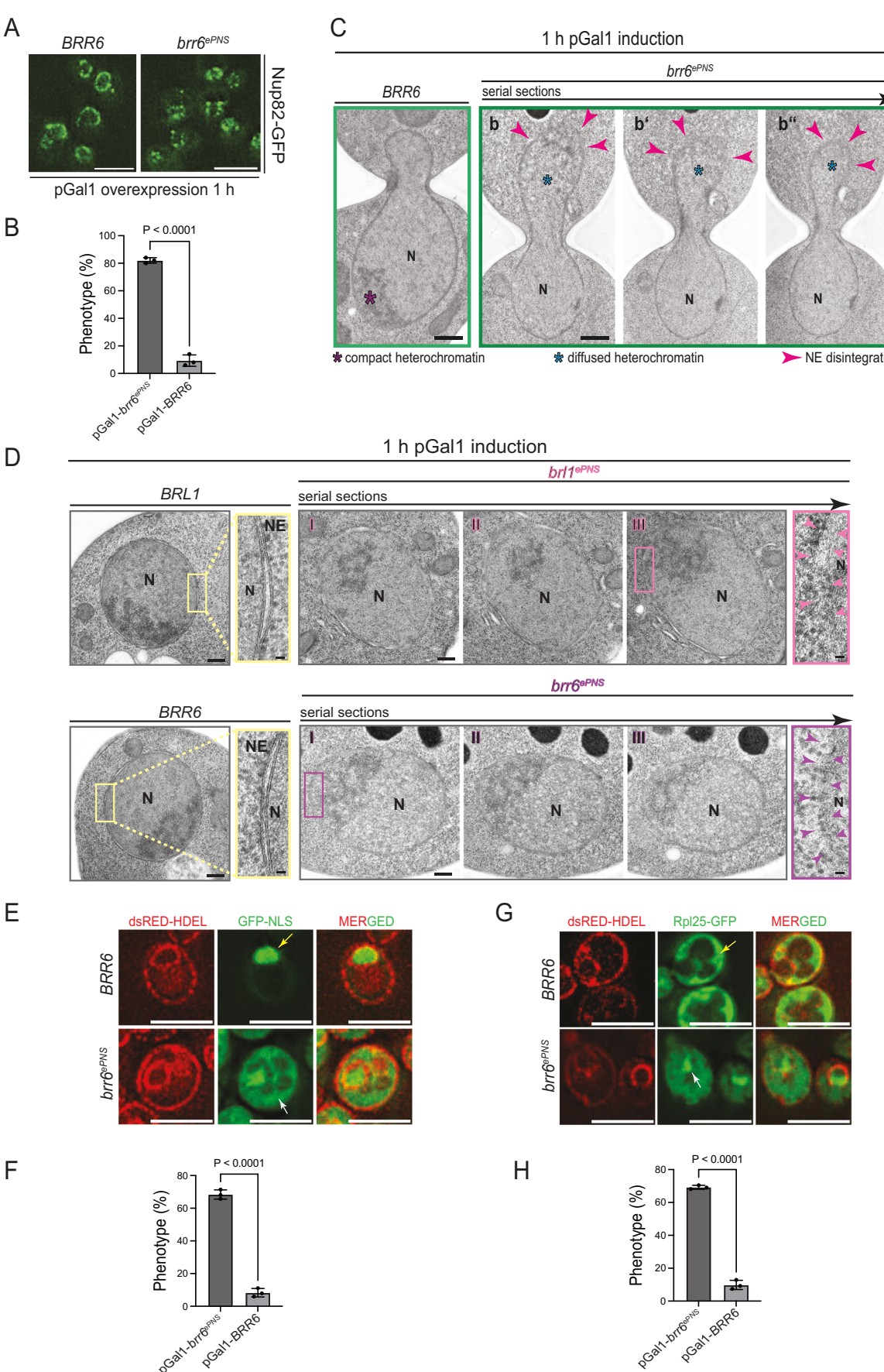

◀  **Figure EV5.   Phenotypes of *brr6^ePNS* overexpression.**

Extension of Fig. 8. (**A**) Analysis of pGal1-*BRR6* and pGal1-*brr6^ePNS* cells carrying *NUP82-GFP*. Localization of Nup82-GFP was analyzed by fluorescence microscopy. Three independent experiments. Size bars: 5 µm. Note, the corresponding experiment with pGal1-*BRL1 and* pGal1-*brl1^ePNS* is shown in Fig. 8G. (**B**) Quantification of (**A**). Total number of analyzed cells: 150 each for both pGal1-*BRR6 and* pGal1-*brr6^ePNS*; Statistical test: unpaired two-tailed *t* test. Data represent the mean ± SD from three independent experiments; ($P = 1.1E-5$). (**C**) EM analysis of cells expressing pGal1-*BRR6* or pGal1-*brr6^ePNS* after 1 h of galactose induction. Panels b–b″ show 80 nm serial sections of the same anaphase cell. Red arrowheads indicate regions of the NE undergoing disintegration. Purple asterisk marks compact chromatin; blue asterisks denote decondensed chromatin adjacent to disintegrating NE regions. N nucleus. Size bars: 500 nm. The corresponding experiment with pGal1-*BRL1* and pGal1-*brl1^ePNS* is shown in Fig. 8I. (**D**) Shows EM analysis of pGal1-*BRL1*, pGal1-*brl1^ePNS*, pGal1-*BRR6* and pGal1-*brr6^ePNS* cells in interphase. I-III indicates serial sections. Arrowheads in enlargements show disintegrated NE. Size bars: 500 nm; enlargements 25 nm. N nucleus, NE nuclear envelope. (**E**) pGal1-*brr6^ePNS* overexpression affects localization of GFP–NLS in the nucleus. *GFP-NLS dsRED-HDEL* cells with pGal1-*BRL1* or pGal1-*brl1^ePNS* were incubated for 1 h with galactose. Localization of GFP–NLS was analyzed by fluorescence microscopy. Size bar: 5 µm. The yellow arrow indicates GFP–NLS in the nucleus; the white arrow NLS-GFP in the cytoplasm. (**F**) Cells from (**E**) were quantified for the localization of GFP–NLS. Three independent experiments with total number of analyzed cells 218 and 216 for pGal1-*BRR6* and pGal1-*brr6^ePNS*, respectively. Statistical test: unpaired two-tailed *t* test. Data shows mean with SD; ($P = 1.13E-5$). (**G**) As (**E**) but with *dsRED-HDEL RPL25-GFP* cells. The yellow arrow indicates Rpl25–GFP in the cytoplasm. The white arrow indicates Rpl25–GFP in the nucleus. Size bars: 5 µm. (**H**) Cells from (**G**) were quantified for the localization of Rpl25–GFP. Three independent experiments with total number of analyzed cells 281 and 311 for pGal1-*BRR6* and pGal1-*brr6^ePNS*, respectively. Statistical test: unpaired two-tailed *t* test. Data shows mean with SD ($P = 4.5E-6$). Source data are available online for this figure.

