## [Peer Review File · The EMBO Journal]

Multifunctional Roles of Brl1-Brr6 in Nuclear Envelope Fusion During Nuclear Pore Complex Biogenesis

Sayan Mondal, Annett Neuner, Azqa Khan, Jlenia Vitale, and Elmar Schiebel

Corresponding author(s): Elmar Schiebel (e.schiebel@zmbh.uni-heidelberg.de) , Azqa Khan (azqa.khan@embl.de)

Review Timeline:

Submission Date:	4th Aug 25
Pre-consultation with Author:	15th Sep 25
Revision Plan Received:	19th Sep 26
Editorial Decision:	26th Sep 25
Revision Received:	23rd Dec 25
Editorial Decision:	30th Jan 26
Revision Received:	3rd Feb 26
Accepted:	4th Feb 26

Editor: Hartmut Vodermaier

Transaction Report:

Dear Elmar,

Thank you for your patience during the evaluation of your recent submission, which was considerably delayed due to limited referee availability combined with slower-than-usual feedback from those experts that eventually agreed to review. We have now finally received a complete set of reports, which I am copying below for your information. As you will see, the referees all appreciate the importance of the overall question, but also raise a number of substantive issues that would need to be satisfactorily addressed before possible publication. While some of these may be relatively straightforward, but nevertheless requiring considerable effort, others appear to be more fundamental, especially some of the well-taken points brought up by referee 1. I will not go through them in detail here, but the overall sentiment seems to be that various key conclusions of the study remain insufficiently supported by the current structural modeling and the complementary experimental analyses.

Since it is not clear if and how the referees' criticisms could be satisfactorily addressed during a regular revision, I would first invite you to go through the reports and to then send me a tentative point-by-point response, detailing how you would attempt to clarify the key concerns in case you should be given the opportunity to revise this work for The EMBO Journal; keeping in mind that our scooping protection policy would in principle allow you sufficient time also for more dedicated follow-up experiments. Based on such a revision proposal (parts of which I may share and discuss with some of the referees), I could then determine whether a major revision for The EMBO Journal would seem realistic; or whether a less substantively revised/toned-down version might alternatively warrant publication in one of our sister journals such as EMBO Reports.

Once more apologies for the substantial delay with this external review, and I look forward to hearing from you within the next two weeks,

Best regards,

Hartmut

Referee #1 (Report for Author)

The manuscript by Mondal and colleagues addresses a fundamental problem, namely how the inner and outer nuclear membranes fuse during nuclear pore complex (NPC) assembly to give rise to the actual pore and thus to a conduit that allows nucleocytoplasmic exchange. The integral membrane proteins Brl1 and Brr6 are the focus of this study. There is ample prior evidence for a role of these two proteins in NPC assembly and membrane fusion, with landmark contributions from the Schiebel lab. The current study extends on this, mainly on the basis of AlphaFold predictions of homotypic Brl1, Brr6, and heterotypic Brl1-Brr6 interactions and potential fusion intermediates. These predictions are, however, highly speculative, and this speculative nature is not at all eased by the fact that the competing Olzmann, Weis, and Kutay labs show similar predictions in their recent preprints. While this paper includes a number of interesting observations, the dodgy modelling leaves the reader with serious doubts.

Specific points:

1. The scores for AlphaFold predictions are clearly below the threshold for a reliable prediction. They are, therefore, not reliable and cannot be taken as facts.
2. Furthermore, the predictions cannot be taken as a test for the hypothesis that such interactions occur. The reason is that the stoichiometry is an input parameter/ boundary condition for the modelling. AlphaFold simply tries to find models that satisfy the boundary condition. It is a circular argument to take the output as a proof for the input, unless the scores indicate a reliable prediction.
3. The sparse mutational analysis does not solve this fundamental problem, because the mutations could affect other types of interactions in the same way.
4. Given the sparse contacts, the modelled subunit-subunit interface for the Brl1-Brr6 interaction looks very weak to me. Hard to believe that such weak contacts could possibly drive membrane fusion, even when assuming that multiple contacts are involved. Here, I would take the very strong interactions between SNAREs as a benchmark for the required interaction strength. Other benchmarks would be the ΔG changes between pre- and post-fusion conformations of viral fusion proteins.
5. The proposed interaction between Nic96 and Brl1 is puzzling. It is well established that Nic96 binds a coiled-coil trimer with three subunits: Nsp1, Nup57, and Nup49. This trimer is an obligate trimer. Instead of the trimer, however, the authors included only a Nup57-Nup49 dimer in the AlphaFold modelling. This calls for artifacts, as documented by a series of crystallographic papers by the Blobel lab on the homologous Nup62-Nup54-Nup58 complex, where the omission of one or two of the partners led to physiologically irrelevant

coiled-coil complexes. Just from looking at the 2D-image in Figure 7D, I would guess that including Nsp1 would cause clashes in the predicted complex.

6. As the authors point out, Br1 would clash with the interaction partners of Nic96 in fully assembled NPCs. At which stage of assembly would membrane fusion and pore formation then occur? Before the Y-complexes assemble with the inner ring components? This would cause a non-selectively permeable nuclear envelope - a pore without a functional barrier. For conceptual reasons, I consider this as unlikely. It also appears to be in conflict with the observation that a loss of the FG Nup Nup116 causes herniations and thus a membrane fusion defect. As linker sequences and the autoproteolytic domain of Nup116 interconnect the inner ring and Y-complexes, it appears more likely that membrane fusion requires a rather advanced stage of NPC scaffold assembly. According the suggestion here, however, fusion would already occur before inner ring and Y-complexes have built the NPC scaffold.

7. I also wonder in how far the assumed self-interactions of Br1 (Figure 1 and EV2) are compatible with the Nic96 interactions. Given that the Nic96 complex is very bulky, it seems impossible then that Br1 homo-oligomers can form as shown in Figure EV2. To me, this looks more like two mutually exclusive models.

8. The authors consider Br1 a transient interaction partner of Nic96 but it co-purifies with Nic96 with a higher efficiency than the stably bound Nup57 ligand. This is puzzling. Can the authors be certain that their membranes have been fully solubilized in their low-salt Triton buffer? Can they authors absolutely exclude that large fragments of residual membranes copurified? I would suggest to repeat the experiment at 150 and 500mM NaCl and to blot for a range of membrane marker proteins (e.g. Sec61, other outer membrane proteins and inner membrane proteins) to establish how selective the co-IP actually is.

9. Anyway, the IP experiment does not provide conclusive evidence for a direct interaction. I would suggest to produce the components/ potentially interacting domains recombinantly and to test interactions by size exclusion chromatography. If the interaction interface is as large as the figure suggests, it should be straightforward to document a homogeneous and stable complex.

10. The authors present new mutations and their phenotypes. From the text, however, it is sometimes difficult to tell which ones had been described before and what is new information. This should be structured in a cleaner way.

11. There are many smaller problems, such as discrepancies between text and figures, between quantifications and images, indications for images of the same dataset being processed in different ways, etc.

Referee #2 (Report for Author)

General summary and opinion about the principle significance of the study, its questions

and findings:

Brl1 and Brr6 are essential, related membrane proteins in the yeast nuclear envelope that temporarily associate with nuclear pore complexes during their assembly, likely to aid in the fusion of the inner and outer nuclear membranes. However, their precise roles in this process are not fully understood. The manuscript "Multifunctional Roles of Brl1-Brr6 in Nuclear Envelope Fusion During Nuclear Pore Complex Biogenesis" by Mondal et al. reveals that Brr6 is involved in both early and late stages of nuclear pore complex assembly. Early functions are demonstrated by AαH mutants that impede nucleoporin recruitment and inner membrane deformation. Notably, the nuclear envelope phenotype ("herniation") differs from the phenotype observed in AαH mutants of Brl1 ("petal-like structures"), suggesting that Brr6 acts at an earlier stage. Mutations in conserved cysteine residues disrupt membrane fusion, while the N-terminus of Brl1 interacts with Nic96 to facilitate its recruitment to new assembly sites. Additionally, artificially lengthening specific regions of Brl1 and Brr6 or overexpressing these protein variants leads to lethality, likely due to compromised nuclear envelope integrity.

Overall, this study provides valuable insights into the mechanisms of nuclear pore complex assembly in yeast, particularly regarding Brr6's function. It suggests that Brr6 may play a role in lateral interactions with Brl1 within the inner membrane and head-to-head interactions connecting the inner and outer membranes. While the data largely support this interpretation, further experiments are needed to strengthen these conclusions.

Specific major concerns essential to be addressed to support the conclusions

1.) 1.) The authors suggest that mutations in the AαH mutants of Brr6 (and Brl1) disrupt the homo/heterodimerization, referred to as "lateral interaction," between Brr6 and Brl1 (specifically brl1F391E and brr6L145E). Interestingly, combining these two mutations appears to rescue the phenotype, which is proposed to restore the interaction. It is surprising that introducing two negatively charged amino acids would reinstate a hydrophobic interaction; one would typically expect an ionic interaction between a negatively and positively charged group. Since the model is primarily supported by AlphaFold predictions, which in the figures does not clearly visualize the compensatory interaction, and hard to interpret blue native gels (Figure EV2F) conducting immunoprecipitation experiments similar to those performed by Zhang et al. (2018) or Lone et al. (2015) would enhance clarity. Additionally, while blue native gel electrophoresis is presented in Figure EV2F, it remains unclear why no 130 kDa signal is detected for the coexpression condition of brl1F391E-brr6L145E.

2.) One of the most intriguing findings in this manuscript is the toxic effect observed with constructs of brl1 and brr6 featuring an extended perinuclear space domain. This

phenomenon is interpreted as indicating that these proteins could induce uncontrolled fusion between the inner and outer nuclear membranes independent of nuclear pore complex assembly. However, the data supporting this interpretation, particularly EM figure 8h, are difficult to interpret. It raises questions as to why the authors present a nucleus from a dividing cell in this context. Additionally, quantification of this significant experiment is absent. If there were indeed a loss of nuclear envelope integrity, one would expect an efflux of nuclear proteins and an influx of cytosolic proteins; testing this with appropriate marker proteins (e.g., immunofluorescence or fluorescent protein fusions) should be considered.

3.) The study suggests that besides lateral interactions, head-to-head interactions between Brr6 and Brl1 are likely crucial for facilitating membrane fusion. The authors propose that a conserved PALXE motif is necessary for this interaction. This hypothesis should be validated by introducing mutation(s) that disrupt the head to head interactions into Brr6 and Brl1; such mutations would predictably block the fusion process, resulting in both proteins exhibiting a petal-like phenotype. While cysteine mutations may affect conserved disulfide bridges and potentially influence overall folding within the perinuclear space region of Brr6 and Brl1, they do not definitively prove the proposed head-to-head interaction.

Minor concerns that should be addressed:

- 1.) Figure 5A demonstrates that the combination of brl1F391E with BRR6 overexpression is viable; however, it also exhibits the petal phenotype shown in Figure 5B. This raises a question about how this is possible, especially considering that the AlphaFold predictions do not indicate a restoration of the Brl1-Brr6 interaction as depicted in Figure EV1. The authors should clarify how these seemingly contradictory observations can coexist and what mechanisms might allow for viability.
- 2.) Figures 4D and 5E present quantification of nuclear pore complexes, but it is unclear how these are defined in these instances. Are they identified through Nsp1 staining or by membrane strain? The authors should provide this information.

Referee #3 (Report for Author)

In the current manuscript, Mondal et al. set out to explore the role(s) of the two integral membrane proteins Brl1 and Brr6 during NPC assembly. They provide evidence that both proteins perform multiple roles in NPC biogenesis, and that their loss of function impairs nucleoporin recruitment and inner nuclear membrane deformation on the one hand and defective fusion of the inner and outer nuclear membranes on the other hand. The authors

show that Brl1 interacts with the nucleoporin Nic96 and that the interaction of Brl1-Brr6-Nic96 is essential to recruit Nic96 and its partner nucleoporins during early NPC assembly. Potential oligomerization of Brl1-Brr6 on the other hand is critical for INM-ONM fusion.

I found this a very interesting paper that provides exiting new insights into the NPC assembly mechanism. The paper is well written and the data are clearly presented. However, some points should be addressed prior to publication.

1. Most of the experiments were done only twice, common sense is that experiments should be done at least three times.
2. The statistics are not always carried out to a sufficient degree. For example, error bars are missing in Figs. 3E and 5E. Fig. 4E, here it is not specified what the data present: mean, median, standard deviation, interquartile range? In cases in which ANOVA test were performed, please specify which.
3. On page 13, the authors state that "..., contacted the ONM without any evidence of membrane fusion (Fig. 6C)." Fig. 6C is a graph, not a micrograph. Please refer to the correct panel.
4. Next, the authors state that they measured the distance between the INM and ONM: where exactly? Next to NPCs, away from NPCs, random? This would be important to know as the spacing in the mutant strains appears, apart from the herniations, similar as in the wild type strain. It would be also useful to include measurements/statistics from the mutant strains into Fig. 6D.
5. Page 14, "It is likely that relatively mild INM deformations were repaired upon BRR6 expression, whereas extended herniations may have reached a point of no return." What exactly is meant by "mild"? How would this herniations look like? Please specify and ideally include EM images.
6. Fig. 6B: it would be beneficial to also show a high magnification image of the NE in wild-type cells.
7. Fig. 7: the pull-down experiment is insufficient. Negative controls are missing to proof the "specific" enrichment of Brl1 and Nup57 with Nic96. Test for other Nups, other membrane proteins, use Nic96 truncations to show that this leads to a loss of interaction.

Minor comments:

1. Fig. 1 and one page 14: the mention of the *S. bombe* Brr6 comes out of the blue. Please remove or elaborate.
2. On page 21: Apq12 is mentioned out of context and was not mentioned before. Please specify or remove.
3. The introduction and the discussion are a bit too long, please get more to the point.
4. Please check carefully that all abbreviations are sufficiently introduced.

19. September 2025

Dear Hartmut,

We thank all three reviewers for their helpful and constructive comments. All reviewers responded positively and emphasized the importance of our work. Overall, the comments from Reviewers 1–3 are straightforward to address, as outlined below. I would like to begin with a few general remarks:

1. Reviewer 1's comments on AlphaFold predictions mainly concern the oligomeric structures. We can refine this section by focusing on the most relevant predictions. However, we strongly believe that showing one or two of these predictions is important (in the Supplement). From discussions at the Potsdam meeting, we know that K. Weis has submitted his manuscript to Nature largely based on AlphaFold predictions and MD simulations and to my knowledge it is presently reviewed. Therefore, we feel it is important to present oligomerization as a possible mechanism for how Brl1 and Brr6 might function (in Supplement). It would be unfortunate if similar approaches were accepted in K. Weis work but not in ours.
2. In the meantime, we have repeated most of the experiments presented in the manuscript three times.
3. One to two months ago, we have initiated most of the additional experiments requested by reviewers 1–3. We are confident that we can complete these experiments in the given revision time, as they are technically straightforward.
4. Reviewer 1's point 5 has already been addressed in the Supplementary Information but may have been overlooked. This point is critical, as it shows that Brl1 is predicted by AlphaFold to interact with the Nic96–Nsp1–Nup57–Nup49 complex.
5. We believe there may also be a misunderstanding by Reviewer 1. Our model proposes that Brr6–Brl1 have both an early and a late function. The early function is not in nuclear envelope fusion, but rather in the initiation of NPC assembly. Reviewer 1's comments focus only on fusion of the inner and outer nuclear membranes, overlooking this earlier role. We can clarify our model further in the revision, which should resolve this issue.
6. There is also overlap among the reviewers' comments. Ultimately, we estimate that only about five additional experiments are needed to fully address all concerns of the three reviewers, and we have already begun these.
7. We should be done with revision latest November-December 2025.

As these points illustrate, we are well prepared for the revision.

Elmar Schiebel

Referee #1 (Report for Author)

The manuscript by Mondal and colleagues addresses a fundamental problem, namely how the inner and outer nuclear membranes fuse during nuclear pore complex (NPC) assembly to give rise to the actual pore and thus to a conduit that allows nucleocytoplasmic exchange. The integral membrane proteins Brl1 and Brr6 are the focus of this study. There is ample prior evidence for a role of these two proteins in NPC assembly and membrane fusion, with landmark contributions from the Schiebel lab. The current study extends on this, mainly on the basis of AlphaFold predictions of homotypic Brl1, Brr6, and heterotypic Brl1-Brr6 interactions and potential fusion intermediates. These predictions are, however, highly speculative, and this speculative nature is not at all eased by the fact that the competing Olzmann, Weis, and Kutay labs show similar predictions in their recent preprints. While this paper includes a number of interesting observations, the dodgy modelling leaves the reader with serious doubts.

Specific points:

1. The scores for AlphaFold predictions are clearly below the threshold for a reliable prediction. They are, therefore, not reliable and cannot be taken as facts.

Comment: We are aware that the scores are relatively low, and we have already pointed this out in the manuscript. To address this concern of reviewer 1, we can scale down the AlphaFold predictions of multimers and present only one representative prediction in the Supplementary Material. This prediction would then be discussed as a potential model in the Discussion section. As you know from K. Weis BioRxiv manuscript, his work on Brl1 and Brr6 relies heavily on AlphaFold predictions, Brl1/Brr6 multimer formation, and MD simulations. His manuscript has been submitted to *Nature* and is most likely under review there (he indicated that he would inform me if *Nature* declined to consider it).

2. Furthermore, the predictions cannot be taken as a test for the hypothesis that such interactions occur. The reason is that the stoichiometry is an input parameter/ boundary condition for the modelling. AlphaFold simply tries to find models that satisfy the boundary condition. It is a circular argument to take the output as a proof for the input, unless the scores indicate a reliable prediction.

Comment: We agree with Reviewer 1's comment. In our manuscript, we have discussed Brl1 and Brr6 oligomerization as one possible model among several. In the revised version, we will clarify this point and provide a more detailed description of alternative models based on Brl1 and Brr6 monomers.

3. The sparse mutational analysis does not solve this fundamental problem, because the mutations could affect other types of interactions in the same way.

Comment: We are currently adding additional data on mutation analysis, particularly focusing on the conserved PAL region and conserved cysteine residues. These data will provide strong support for our model. The reviewer should also take into account the technical challenges of this analysis and the significance of the work. Step by step, our results are building a mechanistic understanding of how nuclear envelope fusion is driven by Brl1 and Brr6 during NPC biogenesis.

4. Given the sparse contacts, the modelled subunit-subunit interface for the Brl1-Brr6

interaction looks very weak to me. Hard to believe that such weak contacts could possibly drive membrane fusion, even when assuming that multiple contacts are involved. Here, I would take the very strong interactions between SNAREs as a benchmark for the required interaction strength. Other benchmarks would be the ΔG changes between pre- and post-fusion conformations of viral fusion proteins.

Comment: Thank you very much for this comment and concern. In fact, this point supports the idea that oligomerization of Br11 and Brr6 may be a prerequisite for nuclear envelope fusion. We will take Reviewer 1's comments into account in the revised manuscript and will include it to the Discussion.

5. The proposed interaction between Nic96 and Br11 is puzzling. It is well established that Nic96 binds a coiled-coil trimer with three subunits: Nsp1, Nup57, and Nup49. This trimer is an obligate trimer. Instead of the trimer, however, the authors included only a Nup57-Nup49 dimer in the AlphaFold modelling. This calls for artifacts, as documented by a series of crystallographic papers by the Blobel lab on the homologous Nup62-Nup54-Nup58 complex, where the omission of one or two of the partners led to physiologically irrelevant coiled-coil complexes. Just from looking at the 2D-image in Figure 7D, I would guess that including Nsp1 would cause clashes in the predicted complex.

Comment: We agree that the interaction of Nic96 with the trimeric complex: Nsp1, Nup57, and Nup49 is well established. Keeping this in mind, we have done the AlphaFold prediction with Br11, Nic96, Nsp1, Nup57, and Nup49. We did not show Nsp1 in the main figure (Figure 7D) but we have shown the whole complex (Br11 with Nic96, Nsp1, Nup57, Nup49) in the supplementary figure (Supplementary Figure 3B). This was probably overlooked by reviewer 1. In the revised version of the manuscript, we can include the whole predicted complex in the main figure.

6. As the authors point out, Br11 would clash with the interaction partners of Nic96 in fully assembled NPCs. At which stage of assembly would membrane fusion and pore formation then occur? Before the Y-complexes assemble with the inner ring components? This would cause a non-selectively permeable nuclear envelope - a pore without a functional barrier. For conceptual reasons, I consider this as unlikely. It also appears to be in conflict with the observation that a loss of the FG Nup Nup116 causes herniations and thus a membrane fusion defect. As linker sequences and the autoproteolytic domain of Nup116 interconnect the inner ring and Y-complexes, it appears more likely that membrane fusion requires a rather advanced stage of NPC scaffold assembly. According the suggestion here, however, fusion would already occur before inner ring and Y-complexes have built the NPC scaffold.

Comment: We think that the N-terminal domain of Br11 transiently interacts with Nic96 for the recruitment. Then the Nup53-Nup59 complex replaces the Br11. As per the KARMA study to show the order of nuclear pore complex maturation events (Onischenko *et al.*, 2020), Nic96, Nup53, Nup59 belong to the early tier complex of the Nups recruitment. Previous studies have also shown that Br11 preferentially binds with the central scaffold Nups (early tier) rather than the intermediate and late tier complexes (the peripheral nucleoplasmic and cytoplasmic) (Kraft *et al.*, 2022). So, we are interpreting that Br11 probably in complex with Brr6 helps in the recruitment of

Nic96 by interacting transiently and this interaction does not have a direct role in the membrane fusion process.

7. I also wonder in how far the assumed self-interactions of Brl1 (Figure 1 and EV2) are compatible with the Nic96 interactions. Given that the Nic96 complex is very bulky, it seems impossible then that Brl1 homo-oligomers can form as shown in Figure EV2. To me, this looks more like two mutually exclusive models.

Comment: Keeping in mind the eight-fold rotational symmetry of nuclear pore complex, there is a possibility that homo-octameric complex of Brl1 at the INM (Figure EV2C) helps in the recruitment of Nic96 through its N-terminal domain. Our interpretation is that this transient interaction helps only in the recruitment and does not contribute directly in the membrane fusion process.

8. The authors consider Brl1 a transient interaction partner of Nic96 but it co-purifies with Nic96 with a higher efficiency than the stably bound Nup57 ligand. This is puzzling. Can the authors be certain that their membranes have been fully solubilized in their low-salt Triton buffer? Can they authors absolutely exclude that large fragments of residual membranes copurified? I would suggest to repeat the experiment at 150 and 500mM NaCl and to blot for a range of membrane marker proteins (e.g. Sec61, other outer membrane proteins and inner membrane proteins) to establish how selective the co-IP actually is.

Comment: The IP conditions were as published before (Zhang et al. 2018) and in this analysis we showed that Brl1 only interacts with a small number of Nups. In the pull-down experiment shown by Mondal et al., we did the immunoprecipitation in the constitutive expression condition of *BRL1* under *Adh1* promoter. Our interpretation is that we get the co-IP of Brl1 with a higher efficiency because of the constitutive expression of *BRL1* although there is a transient interaction between Brl1 and Nic96.

As the referee suggested, we have started repeating the experiment at 150 mM and 500 mM NaCl concentration with varying Triton concentrations, and also blotting for different membrane marker proteins. In addition, we will test the N-Brr6-Brl1 hybrid as a negative control.

9. Anyway, the IP experiment does not provide conclusive evidence for a direct interaction. I would suggest to produce the components/ potentially interacting domains recombinantly and to test interactions by size exclusion chromatography. If the interaction interface is as large as the figure suggests, it should be straightforward to document a homogeneous and stable complex.

Comment: Thank you for the suggestion. We will start producing the recombinant versions of Brl1-N terminal and the corresponding interacting domain of Nic96 and testing the interactions by the size exclusion chromatography. We will include the results in the revised version.

10. The authors present new mutations and their phenotypes. From the text, however, it is sometimes difficult to tell which ones had been described before and what is new information. This should be structured in a cleaner way.

Comment: Thank you for pointing this out. We are currently editing the manuscript as suggested by the referee.

11. There are many smaller problems, such as discrepancies between text and figures, between quantifications and images, indications for images of the same dataset being processed in different ways, etc.

Comment: Thank you for pointing this out. We are currently editing the manuscript as suggested by the referee.

Referee #2 (Report for Author)

General summary and opinion about the principle significance of the study, its questions and findings:

Brl1 and Brr6 are essential, related membrane proteins in the yeast nuclear envelope that temporarily associate with nuclear pore complexes during their assembly, likely to aid in the fusion of the inner and outer nuclear membranes. However, their precise roles in this process are not fully understood. The manuscript "Multifunctional Roles of Brl1-Brr6 in Nuclear Envelope Fusion During Nuclear Pore Complex Biogenesis" by Mondal et al. reveals that Brr6 is involved in both early and late stages of nuclear pore complex assembly. Early functions are demonstrated by AαH mutants that impede nucleoporin recruitment and inner membrane deformation. Notably, the nuclear envelope phenotype ("herniation") differs from the phenotype observed in AαH mutants of Brl1 ("petal-like structures"), suggesting that Brr6 acts at an earlier stage. Mutations in conserved cysteine residues disrupt membrane fusion, while the N-terminus of Brl1 interacts with Nic96 to facilitate its recruitment to new assembly sites. Additionally, artificially lengthening specific regions of Brl1 and Brr6 or overexpressing these protein variants leads to lethality, likely due to compromised nuclear envelope integrity.

Overall, this study provides valuable insights into the mechanisms of nuclear pore complex assembly in yeast, particularly regarding Brr6's function. It suggests that Brr6 may play a role in lateral interactions with Brl1 within the inner membrane and head-to-head interactions connecting the inner and outer membranes. While the data largely support this interpretation, further experiments are needed to strengthen these conclusions.

Specific major concerns essential to be addressed to support the conclusions

- 1.) The authors suggest that mutations in the AαH mutants of Brr6 (and Brl1) disrupt the homo/heterodimerization, referred to as "lateral interaction," between Brr6 and Brl1 (specifically brl1F391E and brr6L145E). Interestingly, combining these two mutations appears to rescue the phenotype, which is proposed to restore the interaction. It is surprising that introducing two negatively charged amino acids would reinstate a hydrophobic interaction; one would typically expect an ionic interaction between a negatively and positively charged group. Since the model is primarily supported by AlphaFold predictions, which in the figures does not clearly visualize the compensatory interaction, and hard to interpret blue native gels (Figure EV2F)

conducting immunoprecipitation experiments similar to those performed by Zhang et al. (2018) or Lone et al. (2015) would enhance clarity. Additionally, while blue native gel electrophoresis is presented in Figure EV2F, it remains unclear why no 130 kDa signal is detected for the coexpression condition of brl1F391E-brr6L145E.

Comment: Thank you for the suggestion to perform immunoprecipitation experiments with wild-type and mutant versions of Brl1 and Brr6 for more stronger validation. We have started the co-IP experiment and can include in the revised version of the manuscript. This is indeed interesting how the replacement of two hydrophobic residues by two negatively charged residues reinstate the interaction. Our interpretation is that the negatively charged residues make anion- π interactions to stabilize the interaction surface (for example, Glu391 (Brl1) with Phe152 (Brr6)). Thus, co-expression of the double mutants results in cross-complementation behaving similar like wild-type.

Regarding blue native gel electrophoresis, there is a very faint signal (130 kDa) in co-expression condition of brl1F391E-brr6L145E. But it's hard to interpret and the blue native gel electrophoresis result does not give any conclusive evidence, just a hint that there is similar interaction patterns like wild-type when both the mutants are co-expressed.

2.) One of the most intriguing findings in this manuscript is the toxic effect observed with constructs of brl1 and brr6 featuring an extended perinuclear space domain. This phenomenon is interpreted as indicating that these proteins could induce uncontrolled fusion between the inner and outer nuclear membranes independent of nuclear pore complex assembly. However, the data supporting this interpretation, particularly EM figure 8h, are difficult to interpret. It raises questions as to why the authors present a nucleus from a dividing cell in this context. Additionally, quantification of this significant experiment is absent. If there were indeed a loss of nuclear envelope integrity, one would expect an efflux of nuclear proteins and an influx of cytosolic proteins; testing this with appropriate marker proteins (e.g., immunofluorescence or fluorescent protein fusions) should be considered.

Comment: We believe the phenotype is more pronounced in anaphase cells, as the nuclear envelope passes through the bud neck and is subjected to additional mechanical stress. We will mention this point in the revised version of our manuscript.

We already have the EM images of interphase cells with disintegrated nuclear envelope as a result of the toxic effect. We will include the EM image in the revised version of the manuscript. We have already done the quantifications for the fluorescence microscopy and we are currently performing the experiments (such as coexpressing either *brr6^{ePNS}* or *brl1^{ePNS}* with *GFP-NLS* in *dsRED-HDEL* cells) to show the loss of the nuclear envelope integrity, as suggested by the referee. We will include the results in the revised version.

3.) The study suggests that besides lateral interactions, head-to-head interactions between Brr6 and Brl1 are likely crucial for facilitating membrane fusion. The authors propose that a conserved PALXE motif is necessary for this interaction. This hypothesis should be validated by introducing mutation(s) that disrupt the head to head interactions into Brr6 and Brl1; such mutations would predictably block the

fusion process, resulting in both proteins exhibiting a petal-like phenotype. While cysteine mutations may affect conserved disulfide bridges and potentially influence overall folding within the perinuclear space region of Brr6 and Brl1, they do not definitively prove the proposed head-to-head interaction.

Comment: Thank you for the suggestion. We have already started doing experiments with the mutations in the conserved PALXE motif. We will include this in the revised version.

Minor concerns that should be addressed:

1.) Figure 5A demonstrates that the combination of *brl1F391E* with *BRR6* overexpression is viable; however, it also exhibits the petal phenotype shown in Figure 5B. This raises a question about how this is possible, especially considering that the AlphaFold predictions do not indicate a restoration of the Brl1-Brr6 interaction as depicted in Figure EV1. The authors should clarify how these seemingly contradictory observations can coexist and what mechanisms might allow for viability.

Comment: In this experiment, co-overexpression of *brl1F391E* and *BRR6* were done on the top of the wild-type versions. As the wild-type copies are present with an additional copies of *BRR6*, this leads to the viability of cells and the *brl1F391E* overexpression gives the petal-like phenotype at the same time.

2.) Figures 4D and 5E present quantification of nuclear pore complexes, but it is unclear how these are defined in these instances. Are they identified through Nsp1 staining or by membrane strain? The authors should provide this information.

Comment: The quantification of nuclear pore complexes was done based on Nsp1 signals at the nuclear pore complexes.

Referee #3 (Report for Author)

In the current manuscript, Mondal et al. set out to explore the role(s) of the two integral membrane proteins Brl1 and Brr6 during NPC assembly. They provide evidence that both proteins perform multiple roles in NPC biogenesis, and that their loss of function impairs nucleoporin recruitment and inner nuclear membrane deformation on the one hand and defective fusion of the inner and outer nuclear membranes on the other hand. The authors show that Brl1 interacts with the nucleoporin Nic96 and that the interaction of Brl1-Brr6-Nic96 is essential to recruit Nic96 and its partner nucleoporins during early NPC assembly. Potential oligomerization of Brl1-Brr6 on the other hand is critical for INM-ONM fusion. I found this a very interesting paper that provides exiting new insights into the NPC assembly mechanism. The paper is well written and the data are clearly presented. However, some points should be addressed prior to publication.

1. Most of the experiments were done only twice, common sense is that experiments should be done at least three times.

Comment: Thank you for pointing this out. We are almost done with three replicates for per experiments. Thus, the revised version will show three independent repetitions.

2. The statistics are not always carried out to a sufficient degree. For example, error bars are missing in Figs. 3E and 5E. Fig. 4E, here it is not specified what the data present: mean, median, standard deviation, interquartile range? In cases in which ANOVA test were performed, please specify which.

Comment: Thank you for pointing this out. We are currently editing the figures with proper statistics, wherever needed.

3. On page 13, the authors state that "..., contacted the ONM without any evidence of membrane fusion (Fig. 6C)." Fig. 6C is a graph, not a micrograph. Please refer to the correct panel.

Comment: Thank you for pointing this out. We have already edited the text.

4. Next, the authors state that they measured the distance between the INM and ONM: where exactly? Next to NPCs, away from NPCs, random? This would be important to know as the spacing in the mutant strains appears, apart from the herniations, similar as in the wild type strain. It would be also useful to include measurements/statistics from the mutant strains into Fig. 6D.

Comment: Thank you for the suggestion. We have a model schematics showing how we have measured the distance between the INM and ONM. We are currently performing the measurements in the mutants as the referee suggested. We will include this in the revised version.

5. Page 14, "It is likely that relatively mild INM deformations were repaired upon BRR6 expression, whereas extended herniations may have reached a point of no return." What exactly is meant by "mild"? How would this herniations look like? Please specify and ideally include EM images.

Comment: Thank you for the suggestion to include the EM images with mild herniations. We already have the EM images and we will include in the revised version.

6. Fig. 6B: it would be beneficial to also show a high magnification image of the NE in wild-type cells.

Comment: Thank you for pointing this out. We will include the higher magnification image of the NE in wild-type cells in the revised version of the manuscript.

7. Fig. 7: the pull-down experiment is insufficient. Negative controls are missing to proof the "specific" enrichment of Br1 and Nup57 with Nic96. Test for other Nups, other membrane proteins, use Nic96 truncations to show that this leads to a loss of interaction.

Comment: Thank you for the suggestion. We have already started to do the pull-

down experiment with proper controls and also in the way suggested by the referee
1. Please refer to our response to the comments of point 8 and 9 from referee 1 for more details.

Minor comments:

1. Fig. 1 and one page 14: the mention of the S. bombe Brr6 comes out of the blue. Please remove or elaborate.

Comment: Thank you for pointing this out. We are currently editing the manuscript as suggested by the referee.

2. On page 21: Apq12 is mentioned out of context and was not mentioned before. Please specify or remove.

Comment: Thank you for pointing this out. We are currently editing the manuscript as suggested by the referee.

3. The introduction and the discussion are a bit too long, please get more to the point.

Comment: Thank you for the suggestion. We are currently editing the manuscript as suggested by the referee.

4. Please check carefully that all abbreviations are sufficiently introduced.

Comment: Thank you for pointing this out. We are currently editing the manuscript as suggested by the referee.

Prof. Elmar Schiebel
Universität Heidelberg
Zentrum für Molekulare Biologie
Im Neuenheimer Feld 282
Heidelberg 69120
Germany

26th Sep 2025

Re: EMBOJ-2025-122067
Multifunctional Roles of Brl1-Brr6 in Nuclear Envelope Fusion During Nuclear Pore Complex Biogenesis

Dear Elmar,

Thank you for your letter outlining how you may address the referee comments on your recent EMBO Journal submission. I have now had a chance to go through your tentative point-by-point responses, and I appreciate that your plans should in principle be suitable for responding to the key issues of referees 1 and 2 in a satisfactory manner. Since a number of the requested clarification experiments are ongoing and have the potential to alter key conclusions of the work, I am obviously not able to definitively predict whether their eventual results will decisively address the most salient concerns; however, I would in any case give you the opportunity to revise the study for The EMBO Journal, along the lines proposed in your revision plan.

It is our policy to allow only a single round of (major) revision, so please keep me updated should there be any unexpected problems with the revisions, or should you require an extension beyond the default 3-months deadline. As always, competing manuscript published elsewhere during the course of this revision will not affect our eventual decision on your study. Finally, please note the detailed information and guidelines on how to prepare a revision below (and in our online Guide to Authors) - closely adhering to them shall greatly facilitate the editorial process at the time of resubmission.

Thank you again for the opportunity to consider this work, and I look forward to receiving your revision in due time.

Yours sincerely,

Hartmut

4) Each main and each Expanded View (EV) figure should be uploaded as individual production-quality files (preferably in .eps,

.tif, .jpg formats). For suggestions on figure preparation/layout, please refer to our Figure Preparation Guidelines: <http://bit.ly/EMBOPressFigurePreparationGuideline>

9) To facilitate reproducibility and cross-laboratory adoption of methodologies, please structure the Materials & Methods section as outlined in our guide to authors, including a completed Reagents and Tools Table that can be downloaded from our author guidelines as well (<https://www.embopress.org/page/journal/14602075/authorguide#structuredmethods>).

10) Digital image enhancement is acceptable practice, as long as it accurately represents the original data and conforms to community standards. If a figure has been subjected to significant electronic manipulation, this must be clearly noted in the figure legend and/or the 'Materials and Methods' section. The editors reserve the right to request original versions of figures and the original images that were used to assemble the figure. Finally, we generally encourage uploading of numerical as well as gel/blot image source data; for details see: embopress.org/page/journal/14602075/authorguide#sourcedata

Revision to The EMBO Journal should be submitted online within 90 days, unless an extension has been requested and approved by the editor; please click on the link below to submit the revision online before 25th Dec 2025:
Link Not Available

If you choose to alternatively have this study further considered by another EMBO Press publication, please use the following hyperlink to directly transfer the manuscript, optionally with inclusion of referee reports and identities:
Link Not Available

We thank all the reviewers for their comments, considering, “this study provides valuable insights into the mechanisms of nuclear pore complex assembly in yeast”, “a very interesting paper that provides exiting new insights into the NPC assembly mechanism“. Below we address all the comments and discuss point-by-point response for all the points raised by the Reviewer’s and the corresponding changes in the revised version of the manuscript.

Referee #1 (Report for Author)

The manuscript by Mondal and colleagues addresses a fundamental problem, namely how the inner and outer nuclear membranes fuse during nuclear pore complex (NPC) assembly to give rise to the actual pore and thus to a conduit that allows nucleocytoplasmic exchange. The integral membrane proteins Brl1 and Brr6 are the focus of this study. There is ample prior evidence for a role of these two proteins in NPC assembly and membrane fusion, with landmark contributions from the Schiebel lab. The current study extends on this, mainly on the basis of AlphaFold predictions of homotypic Brl1, Brr6, and heterotypic Brl1-Brr6 interactions and potential fusion intermediates. These predictions are, however, highly speculative, and this speculative nature is not at all eased by the fact that the competing Olzmann, Weis, and Kutay labs show similar predictions in their recent preprints. While this paper includes a number of interesting observations, the dodgy modelling leaves the reader with serious doubts.

Specific points:

1. The scores for AlphaFold predictions are clearly below the threshold for a reliable prediction. They are, therefore, not reliable and cannot be taken as facts.

Comment: We are aware that the scores are relatively low, and we have already pointed this out in the manuscript. To address this concern of reviewer 1, we have scaled down the AlphaFold predictions of multimers and present only one representative prediction in the Supplemental Fig. 1. This prediction is discussed as a potential model in the Discussion section. The work of K. Weis (BioRxiv manuscript) on Brl1 and Brr6 relies heavily on AlphaFold predictions, Brl1/Brr6 multimer formation, and MD simulations. His manuscript has been submitted to *Nature* and is under review.

2. Furthermore, the predictions cannot be taken as a test for the hypothesis that such interactions occur. The reason is that the stoichiometry is an input parameter/ boundary condition for the modelling. AlphaFold simply tries to find models that satisfy the boundary condition. It is a circular argument to take the output as a proof for the input, unless the scores indicate a reliable prediction.

Comment: We agree with Reviewer 1’s comment. In our initial manuscript, we have discussed different modes of Brl1 and Brr6 oligomerization (for example homo- and hetero-oligomerization). In the revised version of Supplemental Fig. 1, we focus on Brl1 and Brr6 homo-octamers in light of the eightfold symmetry of the NPC. This rationale is discussed in greater detail in the Discussion: “AlphaFold predicts an assembly in which two octameric rings, Brl1 at the INM and Brr6 at the ONM, interact via their PAL motif containing DAH tips (Supplemental Fig. 1C), a model that is particularly compelling given the eightfold symmetry of NPCs. Oligomerization of

Brl1 and Brr6 would enhance their interactions across the INM and ONM. In addition, the octameric Brl1-Brr6 configuration could facilitate the recruitment of eight Nic96 complexes to the INM via N-Brl1 interactions during NPC assembly.“

3. The sparse mutational analysis does not solve this fundamental problem, because the mutations could affect other types of interactions in the same way.

Comment: We have added additional data on mutation analysis, particularly focusing on the conserved PAL region (Figure 7 and Figure EV3). These data provide strong support for our model. In addition, we have added the analysis on the cysteine-alanine mutations in the extended versions of Brl1 and Brr6 (Supplemental Fig. 6). The reviewer should also take into account the technical challenges of this analysis and the significance of the work. Our results provide a mechanistic concept supported by strong data how nuclear envelope fusion is driven by Brl1 and Brr6 during NPC biogenesis.

4. Given the sparse contacts, the modelled subunit-subunit interface for the Brl1-Brr6 interaction looks very weak to me. Hard to believe that such weak contacts could possibly drive membrane fusion, even when assuming that multiple contacts are involved. Here, I would take the very strong interactions between SNAREs as a benchmark for the required interaction strength. Other benchmarks would be the ΔG changes between pre- and post-fusion conformations of viral fusion proteins.

Comment: Thank you very much for this comment and concern. In fact, this point supports the idea that oligomerization of Brl1 and Brr6 may be a prerequisite for nuclear envelope fusion. We have added this idea to the Discussion: „Oligomerization of Brl1 and Brr6 would enhance their interactions across the INM and ONM.“ The phenotype analysis of the PAL mutants (new Fig. 7) supports this interaction model.

5. The proposed interaction between Nic96 and Brl1 is puzzling. It is well established that Nic96 binds a coiled-coil trimer with three subunits: Nsp1, Nup57, and Nup49. This trimer is an obligate trimer. Instead of the trimer, however, the authors included only a Nup57-Nup49 dimer in the AlphaFold modelling. This calls for artifacts, as documented by a series of crystallographic papers by the Blobel lab on the homologous Nup62-Nup54-Nup58 complex, where the omission of one or two of the partners led to physiologically irrelevant coiled-coil complexes. Just from looking at the 2D-image in Figure 7D, I would guess that including Nsp1 would cause clashes in the predicted complex.

Comment: We agree that the interaction of Nic96 with the trimeric complex: Nsp1, Nup57, and Nup49 is well established. Keeping this in mind, we have done the AlphaFold prediction with Brl1, Nic96, Nsp1, Nup57, and Nup49. We did not show Nsp1 in the main figure (Figure 6D) but we have shown the whole complex (Brl1 with Nic96, Nsp1, Nup57, Nup49) in Supplemental Fig. 3B.

6. As the authors point out, Brl1 would clash with the interaction partners of Nic96 in fully assembled NPCs. At which stage of assembly would membrane fusion and pore formation then occur? Before the Y-complexes assemble with the inner ring components? This would cause a non-selectively permeable nuclear envelope - a

pore without a functional barrier. For conceptual reasons, I consider this as unlikely. It also appears to be in conflict with the observation that a loss of the FG Nup Nup116 causes herniations and thus a membrane fusion defect. As linker sequences and the autoproteolytic domain of Nup116 interconnect the inner ring and Y-complexes, it appears more likely that membrane fusion requires a rather advanced stage of NPC scaffold assembly. According the suggestion here, however, fusion would already occur before inner ring and Y-complexes have built the NPC scaffold.

Comment: We think that the N-terminal domain of Brl1 transiently interacts with Nic96 for the recruitment. Then the Nup53-Nup59 complex replaces the Brl1. As per the KARMA study to show the order of nuclear pore complex maturation events (Onischenko *et al.*, 2020), Nic96, Nup53, Nup59 belong to the early tier complex of the Nups recruitment. Previous studies have also shown that Brl1 preferentially binds with the central scaffold Nups (early tier) rather than the intermediate and late tier complexes (the peripheral nucleoplasmic and cytoplasmic) (Kraft *et al.*, 2022). So, we are interpreting that Brl1 probably in complex with Brr6 helps in the recruitment of Nic96 by interacting transiently and this interaction does not have a direct role in the membrane fusion process.

7. I also wonder in how far the assumed self-interactions of Brl1 (Figure 1 and EV2) are compatible with the Nic96 interactions. Given that the Nic96 complex is very bulky, it seems impossible then that Brl1 homo-oligomers can form as shown in Figure EV2. To me, this looks more like two mutually exclusive models.

Comment: Keeping in mind the eight-fold rotational symmetry of NPCs, there is a possibility that homo-octameric complex of Brl1 at the INM (Supplemental Fig. 1) helps in the recruitment of Nic96 through the long and flexible N-terminal domain. Our interpretation is that this transient interaction helps only in the recruitment and does not contribute directly in the membrane fusion process.

8. The authors consider Brl1 a transient interaction partner of Nic96 but it co-purifies with Nic96 with a higher efficiency than the stably bound Nup57 ligand. This is puzzling. Can the authors be certain that their membranes have been fully solubilized in their low-salt Triton buffer? Can they authors absolutely exclude that large fragments of residual membranes copurified? I would suggest to repeat the experiment at 150 and 500mM NaCl and to blot for a range of membrane marker proteins (e.g. Sec61, other outer membrane proteins and inner membrane proteins) to establish how selective the co-IP actually is.

Comment: The IP conditions were as published before (Zhang *et al.* 2018) and in this analysis we showed that Brl1 only interacts with a small number of Nups. In the pull-down experiment shown in this manuscript by Mondal *et al.*, we did the immunoprecipitation in the constitutive expression condition of *BRL1* under Adh1 promoter. Our interpretation is that we get the co-IP of Brl1 with a higher efficiency because of the constitutive expression of *BRL1* although there is a transient interaction between Brl1 and Nic96.

We have performed the experiment with 150 mM and 500 mM NaCl. In the manuscript (Fig. 6E,F), we show the 150 mM data. For the reviewer, we have added to the rebuttal letter the 500 mM data that are identical to the 150 mM data.

Furthermore, we have repeated the pull-down experiment with as the Brr6¹⁻⁵³-Brl1³⁰⁰⁻⁴⁷¹ hybrid fusion, lacking the N-terminus of Brl1 that is predicated to interact with Nic96. Without Brl1 N-terminal, the fusion protein (Brr6¹⁻⁵³-Brl1³⁰⁰⁻⁴⁷¹) could not be pulled-down with Nic96, indication stronger evidence suggesting the interaction between Brl1 N-terminal and Nic96. We have updated the results in the revised manuscript (Fig. 6F).

Furthermore, we have added in vitro interaction data suggesting binding of Nic96 to N-Brl1 (Fig. 6G and comment below).

9. Anyway, the IP experiment does not provide conclusive evidence for a direct interaction. I would suggest to produce the components/ potentially interacting domains recombinantly and to test interactions by size exclusion chromatography. If the interaction interface is as large as the figure suggests, it should be straightforward to document a homogeneous and stable complex.

Comment: Thank you for the suggestion. We have done the in-vitro binding experiment with GST-Brl1¹⁻²⁰⁰ and 6HIS-Nic96 with proper control (GST-nSec1). We have done this experiment by binding recombinant GST-Brl1¹⁻²⁰⁰ and GST-nSec1 to glutathione beads and then incubating with recombinant 6HIS-Nic96. After washing, we analyzed the eluate by immunoblotting. We can successfully show the interaction between Brl1¹⁻²⁰⁰ and Nic96. We have added the results in the manuscript (Fig. 6G).

10. The authors present new mutations and their phenotypes. From the text, however, it is sometimes difficult to tell which ones had been described before and what is new information. This should be structured in a cleaner way.

Comment: Thank you for pointing this out. We have edited the manuscript carefully as suggested.

BRR6 mutants that were characterized in this study by EM (Fig. EV4) were described in Zhang et al. J Cell Biol (2018) in the context of benzyl alcohol sensitivity. EM was not performed on these mutants by Zhang et al. J Cell Biol (2018). To clarify this, we say in result: "To investigate this, we analyzed six randomly generated conditional-lethal *brr6(ts)* mutants that had previously been examined only in the context of benzyl alcohol sensitivity (Zhang *et al*, 2018). Sequence analysis revealed that all *brr6(ts)* mutants harboured mutations in the perinuclear space region, including the conserved cysteine residues (Fig. EV4A) ".

The *brl1*^{C343Y} mutant was described in Vitale et al. MBoC 2022 in the context of overexpression. EM and immuno-EM were not performed. To clarify this, we say in results: "A herniation phenotype similar to that observed in *brr6(ts)* mutants was also detected in the conditional-lethal *brl1*^{C343Y} mutant, which had previously been examined only in pGal1-*brl1*^{C343Y} overexpression experiments (Vitale et al, 2022)."

The AαH mutant *brl1*^{F391E} was described before in Vitale et al. MBoC 2022 as cited in the manuscript.

11. There are many smaller problems, such as discrepancies between text and figures, between quantifications and images, indications for images of the same dataset being processed in different ways, etc.

Comment: Thank you for pointing this out. We have edited the manuscript carefully as suggested. We have modified the figures such that the format is comparable and text and figures match.

Referee #2 (Report for Author)

General summary and opinion about the principle significance of the study, its questions and findings:

Brl1 and Brr6 are essential, related membrane proteins in the yeast nuclear envelope that temporarily associate with nuclear pore complexes during their assembly, likely to aid in the fusion of the inner and outer nuclear membranes. However, their precise roles in this process are not fully understood. The manuscript "Multifunctional Roles of Brl1-Brr6 in Nuclear Envelope Fusion During Nuclear Pore Complex Biogenesis" by Mondal et al. reveals that Brr6 is involved in both early and late stages of nuclear pore complex assembly. Early functions are demonstrated by AαH mutants that impede nucleoporin recruitment and inner membrane deformation. Notably, the nuclear envelope phenotype ("herniation") differs from the phenotype observed in AαH mutants of Brl1 ("petal-like structures"), suggesting that Brr6 acts at an earlier stage. Mutations in conserved cysteine residues disrupt membrane fusion, while the N-terminus of Brl1 interacts with Nic96 to facilitate its recruitment to new assembly sites. Additionally, artificially lengthening specific regions of Brl1 and Brr6 or overexpressing these protein variants leads to lethality, likely due to compromised nuclear envelope integrity.

Overall, this study provides valuable insights into the mechanisms of nuclear pore complex assembly in yeast, particularly regarding Brr6's function. It suggests that Brr6 may play a role in lateral interactions with Brl1 within the inner membrane and head-to-head interactions connecting the inner and outer membranes. While the data largely support this interpretation, further experiments are needed to strengthen these conclusions.

Specific major concerns essential to be addressed to support the conclusions

1.) 1.) The authors suggest that mutations in the AαH mutants of Brr6 (and Brl1) disrupt the homo/heterodimerization, referred to as "lateral interaction," between Brr6 and Brl1 (specifically *brl1*^{F391E} and *brr6*^{L145E}). Interestingly, combining these two mutations appears to rescue the phenotype, which is proposed to restore the

interaction. It is surprising that introducing two negatively charged amino acids would reinstate a hydrophobic interaction; one would typically expect an ionic interaction between a negatively and positively charged group. Since the model is primarily supported by AlphaFold predictions, which in the figures does not clearly visualize the compensatory interaction, and hard to interpret blue native gels (Figure EV2F) conducting immunoprecipitation experiments similar to those performed by Zhang et al. (2018) or Lone et al. (2015) would enhance clarity. Additionally, while blue native gel electrophoresis is presented in Figure EV2F, it remains unclear why no 130 kDa signal is detected for the co-expression condition of *brl1*^{F391E}-*brr6*^{L145E}.

Comment: Thank you for the suggestion to perform immunoprecipitation experiments with wild-type and mutant versions of Brl1 and Brr6 for more stronger validation. We have done the co-IP experiment between the amphipathic helix mutants of Brr6 and Brl1 with proper controls. We can successfully pull-down Brl1 in similar intensities when either both wild-type copies (*BRR6* and *BRL1*) or both amphipathic helix mutants (*brr6*^{L145E} and *brl1*^{F391E}) are co-expressed. There is also a pull-down, however in much lower intensities when either *brr6*^{L145E} and *BRL1* or *BRR6* and *brl1*^{F391E} are co-expressed suggesting that the lateral interactions are reduced by the mutations. We have updated the manuscript accordingly (Fig. EV1).

This is indeed interesting how the replacement of two hydrophobic residues by two negatively charged residues reinstate the interaction. Our interpretation is that the negatively charged residues make anion- π interactions to stabilize the interaction surface (for example, Glu391 (Brl1) with Phe152 (Brr6)). Thus, co-expression of the double mutants results in cross-complementation behaving similar like wild-type.

Regarding blue native gel electrophoresis, there is a signal (130 kDa) in co-expression condition of *brl1*^{F391E}-*brr6*^{L145E} (Supplemental Fig. 1E, asterisk). Interestingly, this larger complex was only observed in Brl1-Brr6 and *brl1*^{F391E}-*brr6*^{L145E} conditions in agreement with the predication of lateral interactions between both proteins in AlphaFold.

2.) One of the most intriguing findings in this manuscript is the toxic effect observed with constructs of *brl1* and *brr6* featuring an extended perinuclear space domain. This phenomenon is interpreted as indicating that these proteins could induce uncontrolled fusion between the inner and outer nuclear membranes independent of nuclear pore complex assembly. However, the data supporting this interpretation, particularly EM figure 8h, are difficult to interpret. It raises questions as to why the authors present a nucleus from a dividing cell in this context. Additionally, quantification of this significant experiment is absent. If there were indeed a loss of nuclear envelope integrity, one would expect an efflux of nuclear proteins and an influx of cytosolic proteins; testing this with appropriate marker proteins (e.g., immunofluorescence or fluorescent protein fusions) should be considered.

Comment: We believe the phenotype is more pronounced in anaphase cells, as the nuclear envelope passes through the bud neck and is subjected to additional mechanical stress. We have mentioned this point in the revised version of our manuscript.

We have also added the EM images of interphase cells with disintegrated nuclear envelope as a result of the toxic effect (Fig. EV5D). We have done the quantifications for the fluorescence microscopy and updated the manuscript accordingly (Fig. 8G,H and Fig. EV5A,B).

We have performed the experiments (expressing either *brr6^{ePNS}* or *brl1^{ePNS}* with *GFP-NLS* in *dsRED-HDEL* cells) to show the loss of the nuclear envelope integrity, as suggested by the referee (Fig. 8J,K and Fig. EV5E,F). When either *brr6^{ePNS}* or *brl1^{ePNS}* were overexpressed, we see GFP signals all over the cells because of the loss of the nuclear envelope integrity. We have added the results and the quantifications in the revised version (Fig. 8 and Fig. EV5).

In addition, we performed the experiments (expressing either *brr6^{ePNS}* or *brl1^{ePNS}* in *Rpl25-GFP dsRED-HDEL* cells) to show the loss of the nuclear envelope integrity, as suggested by the referee. When either *brr6^{ePNS}* or *brl1^{ePNS}* overexpressed, we see influx of GFP signal into the nucleus because of the loss of the nuclear envelope integrity. We have added the results and the quantifications in the revised version (Fig.8L,M and Fig. EV5 G,H).

3.) The study suggests that besides lateral interactions, head-to-head interactions between Brr6 and Brl1 are likely crucial for facilitating membrane fusion. The authors propose that a conserved PALXE motif is necessary for this interaction. This hypothesis should be validated by introducing mutation(s) that disrupt the head to head interactions into Brr6 and Brl1; such mutations would predictably block the fusion process, resulting in both proteins exhibiting a petal-like phenotype. While cysteine mutations may affect conserved disulfide bridges and potentially influence overall folding within the perinuclear space region of Brr6 and Brl1, they do not definitively prove the proposed head-to-head interaction.

Comment: Thank you for the suggestion. We have done experiments with mutations in the PAL motifs of both Brl1 and Brr6. We found those mutations as lethal and showing dominant-negative phenotype when overexpressed on the top of the wild-type copies. Interestingly we found super-herniations phenotypes for both of the proteins, indicating strong roles in the membrane fusion process (Fig. 7 and Fig. EV3).

Minor concerns that should be addressed:

1.) Figure 5A demonstrates that the combination of *brl1^{F391E}* with *BRR6* overexpression is viable; however, it also exhibits the petal phenotype shown in Figure 5B. This raises a question about how this is possible, especially considering that the AlphaFold predictions do not indicate a restoration of the Brl1-Brr6 interaction as depicted in Figure EV1. The authors should clarify how these seemingly contradictory observations can coexist and what mechanisms might allow for viability.

Comment: In this experiment, cells co-overexpressing of *brl1^{F391E}* and *BRR6* were viable with the formation of petals. We agree that this is puzzling. We propose that viability is maintained because functional NPCs can still form in these *brl1^{F391E}* and *BRR6* overexpressing cells, whereas this is not the case for *brl1^{F391E}* overexpression. This is now mentioned in the result section: „We propose that viability is maintained because some functional NPCs can still form in these cells whereas this is not the case for *brl1^{F391E}* overexpression.“.

2.) Figures 4D and 5E present quantification of nuclear pore complexes, but it is unclear how these are defined in these instances. Are they identified through Nsp1 staining or by membrane strain? The authors should provide this information.

Comment: Now Fig. 3D and Fig. 4E. The quantification of NPCs was done based on Nsp1 signals at the nuclear pore complexes. We now mention in the legends to Fig. 3D and Fig. 4E: “The quantification of NPCs was done based on Nsp1 signals at the NPCs.”

Referee #3 (Report for Author)

In the current manuscript, Mondal et al. set out to explore the role(s) of the two integral membrane proteins Brl1 and Brr6 during NPC assembly. They provide evidence that both proteins perform multiple roles in NPC biogenesis, and that their loss of function impairs nucleoporin recruitment and inner nuclear membrane deformation on the one hand and defective fusion of the inner and outer nuclear membranes on the other hand. The authors show that Brl1 interacts with the nucleoporin Nic96 and that the interaction of Brl1-Brr6-Nic96 is essential to recruit Nic96 and its partner nucleoporins during early NPC assembly. Potential oligomerization of Brl1-Brr6 on the other hand is critical for INM-ONM fusion. I found this a very interesting paper that provides exiting new insights into the NPC assembly mechanism. The paper is well written and the data are clearly presented. However, some points should be addressed prior to publication.

1. Most of the experiments were done only twice, common sense is that experiments should be done at least three times.

Comment: Thank you for pointing this out. We now have done the experiments with at least three replicates. This is mentioned in the figure legends. For the EM analysis we have repeated the experiments multiple times, however, each time under slightly different conditions. We have performed EM and immuno-EM independently from the same cells (Fig. 7J, Fig. EV4 and Supplemental Fig. 4) or slightly different yeast strains (e.g. with and without NPC-GFP marker) or different expression or incubation times confirming the phenotypes. In any case, analysis of the phenotypes with three replicates and statistics was done by fluorescence microscopy throughout the manuscript.

2. The statistics are not always carried out to a sufficient degree. For example, error bars are missing in Figs. 3E and 5E. Fig. 4E, here it is not specified what the data present: mean, median, standard deviation, interquartile range? In cases in which ANOVA test were performed, please specify which.

Comment: Thank you for pointing this out.

We now write in Materials and methods:
“**Statistical analysis**”

PRISM v.10.6.1 software (GraphPad) was used for the statistical analysis. Unpaired two-tailed t-test and one-way ANOVA were used for the statistical significance test with a significance level of $p \leq 0.05$.”.

Figs. 3E and 5G lack error bars (now also Supplemental Fig. 6F) because the immuno-EM or EM analysis in this exact setting was only performed once (see 2. above). However, phenotype analysis and statistics was always performed based on the fluorescence microscopy with an additional evidence from EM analysis. For example, Fig. 5D,E and F,G.

Fig. 4E: “Statistical test: One-way ANOVA. The plot shows the median with quartiles. The single dots indicate the individual values of the measurements.”

3. On page 13, the authors state that "..., contacted the ONM without any evidence of membrane fusion (Fig. 6C)." Fig. 6C is a graph, not a micrograph. Please refer to the correct panel.

Comment: Thank you for pointing this out. We have corrected this.

4. Next, the authors state that they measured the distance between the INM and ONM: where exactly? Next to NPCs, away from NPCs, random? This would be important to know as the spacing in the mutant strains appears, apart from the herniations, similar as in the wild type strain. It would be also useful to include measurements/statistics from the mutant strains into Fig. 6D.

Comment: Thank you for the suggestion. We have provided a model schematics showing how we have measured the distance between the INM and ONM (Fig. EV4C. We have also added the measurements in the mutants as the referee suggested (Figure EV4D,E).

5. Page 14, "It is likely that relatively mild INM deformations were repaired upon BRR6 expression, whereas extended herniations may have reached a point of no return." What exactly is meant by "mild"? How would this herniations look like? Please specify and ideally include EM images.

Comment: We have taken this data out of our revised manuscript as it was not adding new information to our story and we have added more relevant data in the revised version. But we can readd the data if requested.

6. Fig. 6B: it would be beneficial to also show a high magnification image of the NE in wild-type cells.

Comment: Thank you for pointing this out. We have included the higher magnification image of the nuclear envelope in wild-type cells in the revised version of the manuscript (Figure EV4B).

7. Fig. 7: the pull-down experiment is insufficient. Negative controls are missing to proof the "specific" enrichment of Br1 and Nup57 with Nic96. Test for other Nups,

other membrane proteins, use Nic96 truncations to show that this leads to a loss of interaction.

Comment: Thank you for the suggestion. We have done the pull-down experiment with proper controls and also in the way suggested by the referee 1. Please refer to our response to the comments of point 8 and 9 from referee 1 for more details.

Minor comments:

1. Fig. 1 and one page 14: the mention of the *S. pombe* Brr6 comes out of the blue. Please remove or elaborate.

Comment: Thank you for pointing this out. We have edited the revised manuscript as suggested.

Fig. 1. Besides *S. pombe*, we have added additional PAL motif sequences to show conservation:

“Fig. 1(E) Sequence alignment of the conserved PAL motif across the indicated Brl1 and Brr6 homologs. ScBrl1 and ScBrr6: Brl1 and Brr6 from *S. cerevisiae*; SpBrr6: *S. pombe* Brr6; CtBrl1: *Chaetomium thermophilum* Brl1; DbBrl1: *Dictyostelium discoideum* Brl1; AnBrl1: *Aspergillus nidulans* Brl1.”

We modified the sentence by adding “for example”:

“Interestingly, homology is not limited to the perinuclear space regions of Brl1 and Brr6 (Zhang *et al.*, 2018); the N-terminus of *S. cerevisiae* Brl1 also shares sequence similarity with for example the N-terminus of *Schizosaccharomyces pombe* Brr6 (Supplemental Fig. 3A).”

We have removed the *S. pombe* part from the Discussion as suggested by the reviewer.

2. On page 21: Apq12 is mentioned out of context and was not mentioned before. Please specify or remove.

Comment: Thank you for pointing this out. We have removed this section from the Discussion as suggested by the reviewer.

3. The introduction and the discussion are a bit too long, please get more to the point.

Comment: Thank you for the suggestion. We have edited the revised manuscript as suggested. We removed one paragraph in the Introduction on KARMA (mentioned later in the text) and one at the beginning of the Discussion that overlapped with the Introduction.

4. Please check carefully that all abbreviations are sufficiently introduced.

Comment: Thank you for pointing this out.

We have added in addition: dsRED-HDEL (His-Asp-Glu-Leu endoplasmic reticulum retention signal fused to dsRED); pGal1-*BRL1* (*BRL1* expressed from the galactose inducible pGal1 promoter); pAdh1-*BRL1* (*BRL1* constitutively expressed from the pAdh1 promoter); GFP-NLS (GFP used to a nuclear localization signal); electron microscopy (EM); endoplasmic reticulum (ER).

Prof. Elmar Schiebel
Universität Heidelberg
Zentrum für Molekulare Biologie
Im Neuenheimer Feld 282
Heidelberg 69120
Germany

30th Jan 2026

Re: EMBOJ-2025-122067R
Multifunctional Roles of Br11-Brr6 in Nuclear Envelope Fusion During Nuclear Pore Complex Biogenesis

Dear Elmar,

Thank you for submitting your revised manuscript to The EMBO Journal. We have at this point received re-reviews from two of the original referees, but unfortunately not heard back from original referee 1. Both reports consider the study improved and most original issues addressed, although referee 2 continues to be concerned about the reliability of some of the structural predictions. In this light, I chose to go myself once more carefully through the original reports and your responses to them, and decided that based on the combined experimental and computational evidence, the study would now appear sufficiently compelling to warrant publication at this point. Following a final round of minor revision needed to address a number of editorial points, we shall therefore be happy to accept the study for publication:

- Please adjust the order of the manuscript sections, and also make sure to use the correct section headers: Title page with complete author information, Abstract, Introduction, Results, Discussion, Methods, Data Availability, Acknowledgements, Disclosure and Competing Interests Statement, References, Main Figure Legends, Tables, Expanded Figure Legends.
- As we are switching from a free-text author contribution statement towards a more formal statement based on Contributor Role Taxonomy (CRediT) terms, please remove the present Author Contribution section and instead specify each author's contribution(s) directly in the Author Information page of our submission system during upload of the final manuscript. See <https://casrai.org/credit/> for more information.
- Please carefully go through the reference list, which currently contains many wrongly formatted entries:
 - * some references seem to list full author names instead of last name - first name initials
 - * several citations are incomplete, missing e.g. citation year, volume, and/or page/locator numbers
 - * Please adjust the format for citation of real preprints: The citation in the text should be: "(preprint: NAME1 et al, YEAR)"; and in the reference list: "NAME1, NAME2, ... (YEAR) article title. BIORXIV doi: XXX"
- Please make sure to call out all individual main figure panels at least once in the text, preferentially in sequential order.
- Please provide suggestions for a short 'blurb' text prefacing and summing up the conceptual aspect of the study in two sentences (max. 250 characters), followed by 3-5 one-sentence 'bullet points' with brief factual statements of key results of the paper; they will form the basis of an editor-written 'Synopsis' accompanying the online version of the article. Please also upload a synopsis image, which can be used as a "visual title" for the synopsis section of your paper. The image should be in PNG or JPG format, and please make sure that it remains in the modest dimensions of (exactly) 550 pixels wide and 300-600 pixels high.
- Finally, during routine pre-acceptance checks, our data editors have raised the following queries regarding figures, data, and legends; I would appreciate if you briefly answered to them in the cover letter of your final submission, and made the requested text modifications with changes/additions highlighted via the "Track changes" option, to facilitate our final checking":
 - 1) Please note that the exact p values are not provided in the legends of figures 3D, 4E, 7H, I; 8H, K, L; EV4 E, EV5 B, F, H
 - 2) Please note that the error bars are not defined in the legend of figure EV5 B
 - 3) Please note that the pink, blue, black asterisks are not defined in the legend of figure S4. This needs to be rectified.
 - 4) Please note that the pink and violet arrows are not defined in the legend of figure S6 E. This needs to be rectified.
 - 5) In the legend of Fig. EV4, it should be explicitly stated that panel C re-displays a section of panel D solely for illustrating the principle explained in the adjacent schematic.
 - 6) In Fig 8C, the colony growth test photograph in the lower panel appears somewhat pixelated, please replace this with a slightly higher-resolution version.

I am returning the manuscript to you for a final round of minor revision, solely to allow you to make these modifications and upload the revised files. Once we will have received them, we should be ready to swiftly proceed with formal acceptance and

production of the manuscript.

With kind regards,
Hartmut

*** PLEASE NOTE: All revised manuscripts are subject to initial checks for completeness and adherence to our formatting guidelines. Revisions may be returned to the authors and delayed in their editorial re-evaluation if they fail to comply to the following requirements. As a first step please read our guidelines for revised submissions:
<https://link.springer.com/journal/44318/submission-guidelines#cms-Revised-submissions>

- 1) Every manuscript requires a Data Availability section (even if only stating that no deposited datasets are included). Primary datasets or computer code produced in the current study have to be deposited in appropriate public repositories prior to resubmission, and reviewer access details provided in case that public access is not yet allowed.
- 2) Each figure legend must specify
 - size of the scale bars that are mandatory for all micrograph panels
 - the statistical test used to generate error bars and P-values
 - the type error bars (e.g., S.E.M., S.D.)
 - the number (n) and nature (biological or technical replicate) of independent experiments underlying each data point
 - Figures may not include error bars for experiments with $n < 3$; scatter plots showing individual data points should be used instead.
- 3) Revised manuscript text (including main tables, and figure legends for main and EV figures) has to be submitted as editable text file (e.g., .docx format). We encourage highlighting of changes (e.g., via text color) for the referees' reference.
- 4) Each main and each Expanded View (EV) figure should be uploaded as individual production-quality files (preferably in .eps, .tif, .jpg formats). For suggestions on figure preparation/layout, please refer to our Figure Preparation Guidelines:
<https://media.springernature.com/original/springer-cms/rest/v1/content/27825798/data/v1>
- 5) Point-by-point response letters should include the original referee comments in full together with your detailed responses to them (and to specific editor requests if applicable), and also be uploaded as editable (e.g., .docx) text files.
- 6) Please complete our Author Checklist, and make sure that information entered into the checklist is also reflected in the manuscript; the checklist will be available to readers as part of the Review Process File.
- 7) All authors listed as (co-)corresponding need to deposit, in their respective author profiles in our submission system, a unique ORCID identifier linked to their name. Please see our Guide to Authors for detailed instructions.
- 8) Please note that supplementary information at EMBO Press has been superseded by the 'Expanded View' for inclusion of additional figures, tables, movies or datasets; with up to five EV Figures being typeset and directly accessible in the HTML version of the article.
- 9) To facilitate reproducibility and cross-laboratory adoption of methodologies, please structure the Materials & Methods section as outlined in our guide to authors, including a completed Reagents and Tools Table.
- 10) Digital image enhancement is acceptable practice, as long as it accurately represents the original data and conforms to community standards. If a figure has been subjected to significant electronic manipulation, this must be clearly noted in the figure legend and/or the 'Materials and Methods' section. The editors reserve the right to request original versions of figures and the original images that were used to assemble the figure. Finally, we generally encourage uploading of numerical as well as gel/blot image source data.

In the interest of ensuring the conceptual advance provided by the work, we recommend submitting a revision within 3 months (30th Apr 2026). Please discuss the revision progress ahead of this time with the editor if you require more time to complete the revisions. Use the link below to submit your revision:

Link Not Available

Referee #2:

In their revised manuscript titled "Multifunctional Roles of Br11-Brr6 in Nuclear Envelope Fusion During Nuclear Pore Complex Biogenesis," Mondal et al. address the functions of Br11 and Brr6 in the assembly of the nuclear pore complex (NPC) in budding yeast. While most points raised by the reviewers have been sufficiently addressed, the main concern, particularly that of Reviewer 1, remains unresolved: the reliability of AlphaFold predictions. The scores for these predictions fall below the threshold typically considered reliable. However, it is important to note that the ideas generated from these predictions are supported by the experimental data presented. In particular, the yeast genetic data provide valuable insights and are quite informative. Additionally, the pull-down experiments conducted with both yeast cells and recombinant proteins bolster their conclusions; nonetheless, including some negative controls with proteins that do not interact with the preys would have strengthened their findings further and It is regrettable that the suggestion to biochemically test complex formation has not been followed up in the manuscript.

Referee #3:

The authors have addressed all previous concerns. I have no further comments to add.

Response letter

Please adjust the order of the manuscript sections, and also make sure to use the correct section headers:

Title page with complete author information, Abstract, Introduction, Results, Discussion, Methods, Data Availability, Acknowledgements, Disclosure and Competing Interests Statement, References, Main Figure Legends, Tables, Expanded Figure Legends.

Thank you for the suggestions. We have arranged the manuscript sections in the order as requested.

- As we are switching from a free-text author contribution statement towards a more formal statement based on Contributor Role Taxonomy (CRediT) terms, please remove the present Author Contribution section and instead specify each author's contribution(s) directly in the Author Information page of our submission system during upload of the final manuscript. See <https://casrai.org/credit/> for more information.

We have now removed the author contribution section from the manuscript text and provided author's contributions in the dedicated section of the submission system.

- Please carefully go through the reference list, which currently contains many wrongly formatted entries:

* some references seem to list full author names instead of last name - first name initials

* several citations are incomplete, missing e.g. citation year, volume, and/or page/locator numbers

* Please adjust the format for citation of real preprints: The citation in the text should be: "(preprint: NAME1 et al, YEAR)"; and in the reference list: "NAME1, NAME2, ... (YEAR) article title. BIORXIV doi: XXX"

Thank you for pointing this out. We have now checked and corrected the reference formats, wherever needed.

- Please make sure to call out all individual main figure panels at least once in the text, preferentially in sequential order.

Thank you for the suggestion. We have carefully checked the same and made sure that all the main figure panels are mentioned in the manuscript text in the correct order.

- Please provide suggestions for a short 'blurb' text prefacing and summing up the conceptual aspect of the study in two sentences (max. 250 characters), followed by 3-5 one-sentence 'bullet points' with brief factual statements of key results of the paper; they will form the basis of an editor-written 'Synopsis' accompanying the online version of the article.

Please also upload a synopsis image, which can be used as a "visual title" for the synopsis section of your paper. The image should be in PNG or JPG format, and please make sure that it remains in the modest dimensions of (exactly) 550 pixels wide and 300-600 pixels high.

We have provided the text and the 'Synopsis' image as requested.

- Finally, during routine pre-acceptance checks, our data editors have raised the following queries regarding figures, data, and legends; I would appreciate if you briefly answered to them in the cover letter of your final submission, and made the requested text modifications with changes/additions highlighted via the "Track changes" option, to facilitate our final checking":

1) Please note that the exact p values are not provided in the legends of figures 3D, 4E, 7H, I; 8H, K, L; EV4 E, EV5 B, F, H

Thank you for pointing this out. We have now mentioned the exact p values, wherever needed. We made the changes via the "Track changes" option, so that it can be re-verified.

2) Please note that the error bars are not defined in the legend of figure EV5 B

Thank you for pointing out. We have mentioned and defined the error bar in the legend of Fig. EV5B. We made the changes via the "Track changes" option, so that it can be re-verified.

3) Please note that the pink, blue, black asterisks are not defined in the legend of figure S4. This needs to be rectified.

Thank you for pointing out. We have now defined pink and blue asterisks in the legend of Fig. S4 (Appendix File). There are no black asterisks in the figure S4; black dots are the gold particle signals.

4) Please note that the pink and violet arrows are not defined in the legend of figure S6 E. This needs to be rectified.

Thank you for pointing out. We have now defined the arrows in the legend of Fig. S6E (Appendix File).

5) In the legend of Fig. EV4, it should be explicitly stated that panel C re-displays a section of panel D solely for illustrating the principle explained in the adjacent schematic.

Thank you for pointing this out. We have now mentioned about the re-display of the micrograph in Fig. EV4C just for the purpose of illustration. We made the changes via the "Track changes" option, so that it can be re-verified.

6) In Fig 8C, the colony growth test photograph in the lower panel appears somewhat pixelated, please replace this with a slightly higher-resolution version.

Thank you for pointing it out. We have now replaced the lower panel with a high-resolution version. We have submitted the revised fig. 8.

Prof. Elmar Schiebel
Universität Heidelberg
Zentrum für Molekulare Biologie
Im Neuenheimer Feld 282
Heidelberg 69120
Germany

4th Feb 2026

Re: EMBOJ-2025-122067R1
Multifunctional Roles of Brl1-Brr6 in Nuclear Envelope Fusion During Nuclear Pore Complex Biogenesis

Dear Elmar,

Thank you for submitting your final revised manuscript for our consideration. I am pleased to inform you that we have now accepted it for publication in The EMBO Journal.

You may qualify for financial assistance for your publication charges - either via a Springer Nature fully open access agreement or an EMBO initiative. Check your eligibility: <https://link.springer.com/journal/44318/how-to-publish-with-us>

With best regards,

Hartmut

Please note that it is The EMBO Journal policy for the transcript of the editorial process (containing referee reports and your response letters) to be published as an online supplement to each paper. If you should prefer removal of any referee-only figures included in the point-by-point response(s), e.g. because they may still be used for future publication or because they have been reproduced from published work by others, please do let us know immediately via response email.

More information is available here: <https://link.springer.com/partners/embo-press/editorial-policies#Peer%20review>